

# Simulation Improvements of ECHAM5-NEMO3.6 and ECHAM6-NEMO3.6 Coupled Models Compared to MPI-ESM and the Corresponding Physical Mechanisms

Shu Gui [1], Ruowen Yang [1], and Jie Cao [1],

[1] Department of Atmospheric Sciences, Yunnan University, Kunming, 650091, China.

*Correspondence to*: Jie Cao (caoj@ynu.edu.cn) and Ruowen Yang (yangruowen@ynu.edu.cn)

**Abstract.** To improve the model simulation through decisive coupling mechanisms, rather than blindly updating the parameterization schemes, it is necessary to compare model performances between the CGCMs with the same atmospheric or oceanic component model. Therefore, two new CGCMs have been developed with the same oceanic component model, namely

ECHAM5-NEMO3.6 and ECHAM6-NEMO3.6. The MPI-ESM that consists of ECHAM6 and MPIOM has also been employed. Experiments are carried out with the same settings in coupler and individual component model if applicable, and the new models show substantial improvements in the simulation of SST, precipitation and ocean currents. Further analysis has made it clear that the primary cause of SST biases in ECHAM5-NEMO3.6 and ECHAM6-NEMO3.6 can be attributed to the momentum field, while oceanic dynamics and surface radiation budget are accountable for more SST deviations in the

MPI-ESM. Inter-model comparison between the coupled models with the same oceanic model suggests that cumulus convection is in the central part of simulation differences, which finally influence the SST through WES feedback mechanism. Whereas the OGCM replacement shows that latent heat of evaporation plays a predominant role in changing SST and surface radiation budget, and eventually bringing about variations in air temperature and atmospheric circulation. The mechanisms revealed in this study provide a new perspective of bias genesis during model coupling, which can be helpful for tuning other

climate models towards a more realistic simulation.

## 1 Introduction

The physical processes sensitive to the air-sea interaction and its impact on climate variabilities have been studied for decades. Research findings suggest that dynamic air–sea coupling is important for tropical cyclone prediction regarding its intensity and rate of intensification (Sandery et al., 2010; Chen et al., 2013; Lin et al., 2018). The air-sea interaction has a

substantial effect on precipitation response to ENSO teleconnections (Langenbrunner and Neelin, 2013), which is also associated with basic state climatology (Ham and Kug, 2015) in the Coupled Models Intercomparison Project phase 5 (CMIP5) models. The double intertropical convergence zone (ITCZ) problem of precipitation simulation in the coupled general circulation models (CGCMs) are closely linked with ocean-atmosphere feedbacks, including Bjerknes feedback, sea surface temperature (SST)-surface latent and surface shortwave flux feedback (Lin, 2007). The ocean-atmosphere interaction tends to



amplify trade wind biases in general circulation models (GCMs) (Li and Xie, 2014), which is also responsible for excessive cold tongue simulation in equatorial Pacific. Besides traditional understanding of coupling processes that account for SST and precipitation biases, recent studies have revealed other factors in the air-sea interaction that significantly contribute to the model bias pattern. Burls et al. (2017) find a quadratic relationship between extra-tropical Pacific albedo and equatorial SST

bias, and Pham et al. (2017) suggested that the deep cycle of cold tongue turbulence can be affected by cloud cover and rain. The air-sea kinetic energy budget is found to be linked with surface gravity waves, where wave age and friction velocity affect in the ratio between kinetic energy from the winds and underlying surface currents (Fan and Hwang, 2017).

The dynamic mechanisms that play a major role in the biases propagation in the CGCMs are also investigated from a wide range of perspectives. Double ITCZ precipitation problem is usually associated with biases in radiation budget and

surface winds (Lin, 2007). There is a close relationship between clouds and SST variation (Klein & Hartmann, 1993; Norris & Leovy, 1994), and changes of low clouds and shortwave radiation flux react on the SST and sea level pressure (SLP) (Norris et al., 1998; Mochizuki & Awaji, 2008; Bond & Cronin, 2008). Wu and Kinter III (2010) suggested that high-frequency changes in atmospheric circulation affects largely on the surface shortwave radiation, and hence its correlation with SST variability in the mid-latitude North Pacific. The weak simulation of Atlantic meridional overturning circulation (AMOC) is

found to be responsible for sea surface temperature (SST) cold biases in the northern hemisphere in 22 CMIP5 climate models (Wang et al., 2014), which poses a great impact on the North Pacific through Northern Hemisphere annular mode (NAM) and wind-evaporation-SST (WES) feedback, combining SST biases in extratropical North Atlantic (ENA) and tropical North Atlantic respectively (Zhang & Zhao, 2015). The temperature bias can also be attributed to underestimation of water vapor amount (Liu et al., 2011), radiative and non-radiative processes (Ren et al., 2015), modelling of cloud-radiation feedback (Song

et al., 2012) and gap winds (Sun & Yu, 2006). Wu and Liu (2003) suggested that the regional SST variations in central and eastern North Pacific are separately resulted from changes of Ekman advection and surface heat flux, both of which are affected by the subtropical ocean circulation, as a part of meridional overturning circulation in North Pacific (NPMOC). The NPMOC has been found to work as a bridge for mass and heat exchanges (McCreary and Yu, 1992; Liu et al., 1994), and is largely driven by sea surface wind stress and resulting in east-west sea level slope (Liu et al., 2011; Liu et al., 2013). However, there

are larger discrepancies in the estimates of drag coefficients among different computational approaches, especially due to uncertainties in velocity measurements and removal of non-wind-driven currents.

Some efforts have been made to improve the SST simulation quality in a variety of CGCMs, for example, through changing zonal filtering and advection scheme (Xiao, 2006), modifying radiation and cumulus parameterization scheme (Bao et al., 2010), including frozen precipitating hydrometeors in cloud mass (Li et al., 2014), and decreasing relative humidity

threshold for low cloud formation (Tang et al., 2016). However, these studies only focus on a limited range of processes that turn out to be important in the statistical analysis of model biases. Updates in the parameterization schemes bring in both improvements and setbacks in simulation. On behalf of oscillations in atmosphere-ocean coupling, the contribution of the individual component model to the simulation of some key variables, regarding ENSO variability, extreme precipitation and hurricanes, and climate response to anthropogenic greenhouse gases, have not yet been made clear.



This paper studies the simulation improvements by new combinations of component models, analyses differences by respectively changing each component model, and determines the key physical processes behind the simulation deviations. To compare relative contributions of the individual component model in the CGCM, we need 2 CGCMs based on the same oceanic model but different atmospheric models, and another 2 CGCMs with the same atmospheric model but different oceanic models.

For this purpose, two new CGCMs have been developed. One uses ECHAM5.4 as the atmospheric component model and NEMO3.6st as the oceanic component model, which is referred to as ECHAM5-NEMO3.6. The other uses ECHAM6.3 as the atmospheric component model and NEMO3.6st as the oceanic component model, and thus referred to as ECHAM6-NEMO3.6. The MPI-ESM developed by Max-Plank Institute for Meteorology, based on ECHAM6.3 for atmosphere and MPIOM for ocean, is also used in this study. To minimize simulation differences caused by model configurations, the three coupled models

are set to the same coupling frequency of every 4 hours, and the same horizontal resolution T63 (192 longitudes×96 latitudes) for the atmospheric component model. The ocean models, namely MPIOM and NEMO3.6st, have different model structures and thus being used with their own default configurations. The content organization of this paper is as follows: A brief description of model frameworks and experiment setup configurations are presented in section 2. The reanalysis data sets used for model assessment and bias analysis are illustrated in section 3. Model evaluation and comparison among ECHAM5-

NEMO3.6, ECHAM6-NEMO3.6 and MPI-ESM are presented in section 4. Further analysis on the cold tongue bias and opposite SST bias in North Pacific is presented in Section 5. Determination of key physical processes responsible for the differences in SST simulation after changing each component model is elaborated in Section 6. Summary and discussion part is in Section 7.

## 2 Model Description

### 2.1 OGCM

### 2.1.1 NEMO

NEMO model is a well renowned modelling system with high skills in global oceanic circulation simulation, which has been widely used for scientific research, weather forecast (Storkey et al., 2014; Megann et al., 2014), and reanalysis data assimilation (Mogensen et al., 2012a, b). Designed to serve as a flexible tool for ocean and sea ice studies, NEMO manifests

good usability interacting with other ACGMs (Gualdi et al., 2003; Luo et al., 2005; Park et al., 2009; Dunlap et al., 2014; Huang et al., 2014). The ocean component of NEMO calculates primitive equations on the ORCA2 grid, a tripolar grid of 182 (longitude)×149 (latitude) curvilinear orthogonal mesh in horizontal direction and 31 vertical levels on partial step z coordinate in current research. Turbulent kinetic energy (TKE) closure scheme has been chosen for vertical mixing with enhanced vertical diffusion for convective processes.



### 2.1.2 MPIOM

MPIOM model is formulated on Arakawa-C grid for horizontal dimension and z-grid for vertical dimension, using the hydrostatic and Boussinesq approximations in the model dynamic equations (Jungclaus et al., 2006; Jungclaus et al., 2013). Vertical mixing and diffusion are parameterized following Pacanowski and Philander (1981), with a diffusion coefficient varies

with grid spacing (Redi, 1982). Sea-ice model is included in MPIOM where sea-ice thickness is modulated by turbulent atmospheric fluxes and oceanic heat transport (Wolff et al., 1997; Marsland et al., 2003; Notz et al., 2013). The model is configured with 40 unevenly spaced vertical levels on the GR1.5 grid, a conformal mapping grid of 256 (longitude) × 220 (latitude) in the horizontal making horizontal resolution approximately 1.5°.

### 2.2 AGCM

The ECHAM atmospheric model developed by the Max Planck Institute for Meteorology has been used in many studies since its first version (ECHAM1) branched from the cycle 17 operational model at Medium Range Weather Forecasts (ECMWF) (Roeckner et al., 1989; Simmons et al., 1989). Incorporation of new features along the course of model development gradually makes ECHAM capable of reproducing meticulous characteristics in the weather system, including those in cumulus convection, moisture transport, radiation and land-surface processes (Roeckner et al., 1996, 2003; Raddatz et al., 2007; Brovkin

et al., 2009). ECHAM has also been employed in the coupled earth modelling system, from coupling with large‑scale geostrophic ocean model (LSG) (Maier‑Reimer et al., 1993) to the latest version of Earth system model MPI-ESM (Baehr et al., 2015). The ECHAM model consists of a dry spectral dynamic core, a set of parameterization schemes dealing with solar irradiance, moist convection, land-surface properties, etc. The versions in use for this paper are ECHAM5.4 (Roeckner et al., 2003) and ECHAM6.3 (Stevens et al., 2013). Major updates from ECHAM5 to ECHAM6 include improved representation of

shortwave spectrum, a new aerosol parameterization scheme, middle atmosphere and surface albedo descriptions are also enhanced. Despite the new implementations in ECHAM6, the AGCMs are set to the same configuration if applicable, to minimize the differences caused by updates of physical parameterization schemes.

### 2.3 CGCM

#### 2.3.1 ECHAM5-NEMO3.6

The schematic structure of ECHAM5-NEMO3.6 is shown in Fig. 1a. Overall, the ECHAM5-NEMO3.6 consists of the atmospheric model ECHAM5.4, oceanic and sea ice model NEMO3.6st (the stable version of NEMO3.6), and the coupler Ocean Atmosphere Sea Ice Soil (OASIS3) (Valcke, 2013). Although ECHAM5 (Roeckner et al., 2003, 2006) is an older version of the atmospheric model developed by Max Plank Institute of Meteorology compared with ECHAM6 (Stevens et al., 2013), it was employed by the previous coupled atmosphere ocean model ECHAM5/MPI-OM (Jungclaus et al., 2006), which

has been well tested and reassured of accurate surface flux transfer between the oceanic and atmospheric component models. Based on the interchange coupling structure of the ECHAM5/MPI-OM, seventeen variables are passed from ECHAM5 to



NEMO-3.6st through OASIS3 coupler, including physical quantities such as solar radiation, non-solar heat flux, wind stress and precipitation. The meridional and zonal wind stress vectors are passed to the ORCA2 V and U grid of NEMO-3.6st. In the opposite direction, six variables comprised of the SST, sea ice fraction, ocean currents and so forth are transferred from the ocean model to the atmosphere. Bilinear interpolation method is used in the exchanges of physical variables between ECHAM5

gaussian grid and NEMO-3.6st ORCA2 grid. The time steps for ocean and atmosphere models are both set to 1200 seconds, and the coupling frequency is 4 hours once (every 12-time steps).

### 2.3.2 ECHAM6-NEMO3.6

Following the coupling framework of MPI-ESM (the update version of ECHAM5/MPI-OM, version number MPI-ESM-1.2.00p4), ECHAM6-NEMO3.6 has been developed with the same atmospheric component model ECHAM-6.3, coupled with

NEMO 3.6 stable version through OASIS3-MCT (Craig et al., 2017) (Fig. 1b). Namelist settings of the ECHAM-6.3 and the coupler OASIS3-MCT are brought into correspondence with those in ECHAM5-NEMO3.6 to the utmost, for example, with the same horizontal resolution T63 on gaussian grid and the same parameterization settings for greenhouse gases. Details in experiment configuration are elaborated in section 2.4.

### 2.3.3 MPI-ESM

The MPI-ESM is comprised of the atmospheric general circulation model ECHAM6 and the oceanic circulation model MPIOM (Jungclaus et al., 2013; Stevens et al., 2013). It has been continuously developed at Max Planck Institute for Meteorology and has been successfully applied on a broad range of studies, including volcano studies (Zanchettin et al. 2013; Zhang et al. 2013), anthropogenic land cover change (Reick et al., 2013; Brovkin et al., 2013), circulation feedback sensitive to Intertropical Convergence Zone (ITCZ), and double ITCZ precipitation (Mobis and Stevens, 2012). MPI-ESM has been

used in CMIP5 and is employed in the upcoming CMIP6. Due to high computational cost, the low-resolution version MPI-ESM-LR is used in this study (version number MPI-ESM-1.2.00p4), with ECHAM6 running at T63L47 resolution (horizontal resolution about 1.875° and 47 vertical levels) and MPIOM running at GR1.5 horizontal grid (about 1.5° in grid distance) with 40 vertical levels from sea surface to the bottom. Experimental settings are migrated from piControl default configuration, and then adjusted to being in consistence with those of ECHAM5-NEMO3.6 and ECHAM6-NEMO3.6 experiments. The major

differences from default piControl experiment are the increased coupling frequency (4 hours once), and climatology recalculated from the year 1981 to 2010 to serve as the model input files (e.g. aerosol properties, ozone mole fractions and land use transitions).

### 2.4 Experimental Setup

The control experiments conducted in this study are aimed to reproduce atmospheric and oceanic circulation characteristics of present time, and then to compare with each other in order to examine model performance improvements.

The coupled control simulation is thus configured with reference to MPI-ESM piControl experiment settings. Model



initialization is started from the climatology basic state recalculated with the AMIP run input data from 1981 to 2010. The atmospheric component models (ECHAM-5.4 and ECHAM-6.3) used in this study are running on T63 gaussian grid (approximately equivalent to 1.875°× 1.875° on average), with the same coupling frequency of 4 hours to exchange momentum and heat fluxes with the ocean component model. Physical parameterization schemes relative to solar irradiance, aerosol optical

properties, cumulus convection and strong stratospheric damping are maintained the same among ECHAM5-NEMO3.6, ECHAM6-NEMO3.6, and MPI-ESM experiments. Since the timestep length of ECHAM-6.3 in coupled mode is suggested to be 450 seconds according to its user manual, only the ECHAM5-NEMO3.6 experiment uses 1200° seconds as timestep length for the atmospheric model ECHAM-5.4. The oceanic component models (MPIOM and NEMO3.6) possess different model structures and mapping technologies, making it impossible to directly migrate physical parameterization settings from one to

the other. In this regard, namelist settings of MPIOM and NEMO3.6 still follow their own default settings for control run provided by their respective official websites. Three coupled experiments, namely ECHAM5-NEMO3.6, ECHAM6-NEMO3.6, and MPI-ESM experiments, have been conducted for 200-year realizations including spin-up runs. Model results spanning the last 100 years (model year 101 to 200) should be well-equilibrated, and thus they are used to compute simulation climatology for the inter-model comparison analysis.

**3. Reanalysis Data**

The assessment of model performance with reference to each CGCM employs the monthly data from the Hadley Center (HadISST) (Rayner et al., 2003). Model precipitation evaluation uses reanalysis data of the Global Precipitation Climatology Project (GPCP) (Adler et al., 2003). Observations of the mean sea level pressure, zonal and meridional winds at 10m height, cloud cover and surface temperature use the ERA-Interim monthly reanalysis data (Simmons et al., 2006). Surface wind stress

data from the Scatterometer Climatology of Ocean Winds (SCOW) including QuikSCAT measurements (Risien & Chelton, 2008) has been chosen for its more advanced stress-measuring instrument and better sampling, which has been widely used in researches regarding oceanic circulation and dynamical processes (Kanzow et al., 2010; Roquet et al., 2011; Johnson et al., 2012) and evaluation of CGCMs and reanalysis data (Xue et al., 2011; Lee et al., 2013). To characterize the changes in ocean circulation associated with the SST bias, the observation from SODA reanalysis data (Carton & Giese, 2008) has been used

following the massive researches on ocean variability and mechanisms (Dewitte et al., 2009; Tett et al., 2014; Drenkard & Karnauskas, 2014; Vargas-Hernández et al., 2015). Finally, surface net radiation flux from CERES EBAF-Surface Ed4.0 (Kato et al., 2013) is employed for its higher accuracy by using more accurate cloud data to calculate solar radiation at the Earth's surface (Wild et al., 2013; Wild et al., 2015; Zhang et al., 2015), which is better than popular reanalysis data sets including NCEP-DOE, MERRA and ERA-Interim (Zhang et al., 2016). Because the resolution of NEMO ORCA2 grid in high latitudes

is coarser (about 2°) than tropics (about 0.5°), the SST model data has been remapped onto 1° × 1° to ensure the interpolation accuracy. The precipitation from ECHAM-5.4 and ECHAM-6.3 (T63 resolution) has been re-gridded to the GPCP data grid (2.5° × 2.5°). Other physical quantities are interpolated onto 1° × 1° grid for both model and reanalysis data. As SODA3.3



reanalysis data has more vertical levels than NEMO model output, especially in the upper ocean, the OGCM velocity output has been interpolated onto SODA3.3 vertical levels to better compare the vertical circulation. The time period of observation climatology spans from 1981 to 2010, except for SCOW with 122 months from September 1999 to October 2009 and CERES EBAF from 2000 to 2015. The differences between model and observation are defined as the anomalous fields with simulation

minus reanalysis data.

## 4. Model Evaluation and Simulation Improvements

### 4.1 Overall Performance

The simulated climatology of 9 physical variables, which are sensitive to coupling processes of each CGCM on the global domain, has been evaluated with Taylor's diagram analysis (Fig. 2). Summer and winter seasons are chosen because they

represent the opposite extreme conditions in the course of model integration. The Taylor diagram shows satisfactory model performances for most physical quantities' simulation in the opposite seasons, except for meridional currents of MPI-ESM that lie beyond 2 times the standard deviation. It is noteworthy that these errors are not caused by changes in the MPI-ESM experiment settings, which is subject to discussion later by comparing with piControl experiment results in Section 5. The best simulations among the three CGCMs in summer are topped by 2m temperature of ECHAM5-NEMO3.6 with the maximum

pattern correlation coefficient 0.995, followed by SST simulation in ECHAM6-NEMO3.6 and total radiation flux in MPI-ESM. For the rest of variables other than ocean surface currents, the standard deviations lie between 0.953 and 1.336, which represent an unremarkable deviation in spatial variability of model simulation compared to observational counterparts. The simulation differences for each variable in terms of root-mean-square deviations (RMSD) are below 0.840, with smaller values accompanied by higher correlation coefficients. The distributions of these statistical values in winter season are a little bit

different, marked by 0.996 for SST simulation in ECHAM6-NEMO3.6 as the top, and followed by 2m temperature also in ECHAM6-NEMO3.6 and total radiation flux in MPI-ESM. Standard deviations of all simulated variables except for ocean currents are between 0.865 and 1.218, and RMSD values are within 0.765, both of which show a little improvement than the summer case. Precipitation simulation in MPI-ESM experiment for both seasons resembles that at the annual scale for MPI-ESM-LR experiments in Stevens et al. (2013).

The model performances shown in Fig. 2 are reasonably good compared with previous studies evaluating CMIP5 models (Sheffield et al., 2013), with simulation improvements relative to surface temperature and winds in ECHAM6-NEMO3.6 and ECHAM5-NEMO3.6. Improvements in climatology simulation are resulted from changing component model, for example, 2m temperature simulation in ECHAM6-NEMO3.6 is better than that in MPI-ESM, because the ocean model MPIOM in MPI-ESM has been replaced by NEMO3.6st to make ECHAM6-NEMO3.6. It arouses interests in how component model

replacement affects the simulation in detail. Therefore, a few key variables in the course of coupling are selected to do inter-model comparison.



### 4.2 Sea Surface Temperature

The SST properties are among the most important factors that decides air-sea fluxes in a coupled system. Model biases of SST seasonal climatology in summer (June-July-August, JJA) and winter (December-January-February, DJF) are presented in Fig. 3. The differences between model simulations and observational counterparts are usually less than 2℃ except for polar areas. Reanalysis data itself is responsible for large biases near polar oceans, because there are also large biases in the previous research by Huang et al. (2014) using the same reanalysis dataset. Taken into account SST biases in both summer and winter, it can be seen that the spatial distribution of SST bias in the MPI-ESM experiment (Fig. 3e, f) resembles main characteristics of the annual SST bias in Jungclaus et al. (2013). In boreal summer, three CGCMs exhibits excessive cold tongue simulation with the minimum extent in ECHAM5-NEMO3.6 (Fig. 3c, d) and the maximum extent in MPI-ESM (Fig. 3e, f). There are substantial cold biases in western North Pacific and North Atlantic in ECHAM6-NEMO3.6 and MPI-ESM experiments. However, the ECHAM5-NEMO3.5 exhibits warm SST biases instead of cold biases in North Pacific, and no remarkable SST bias is found North Atlantic. The opposite SST biases in North Pacific may imply different physical mechanisms that take effects on the coupling processes, which will be discussed later. In boreal winter, the ECHAM6-NEMO3.6 only shows remarkably warm SST biases over 2 ℃ on southern tropical Atlantic, while other two CGCMs still present significant SST biases in other parts of the globe. The ECHAM5-NEMO3.6 features warm SST biases over the southern tropical ocean, with a large area of warm SST bias over the eastern Pacific. In contrast, MPI-ESM shows significantly cold SST biases over northern subtropical Pacific, mid-latitude southern Indian Ocean, and mid-latitude northern Atlantic. The Southern Ocean SST nevertheless takes on warm biases in the MPI-ESM experiment.

Model improvements of the SST simulation can be summarized as follows: ECHAM5-NEMO3.6 best decreases the scope and intensity of excessive cold tongue and presents no remarkable biases over North Atlantic in boreal summer, and ECHAM6-NEMO3.6 reproduces the best winter climatology with only tropical South Atlantic suffering from more than 2 ℃ bias. Their SST simulation qualities are substantially improved compared with that of MPI-ESM, both from Taylor's diagram results (Fig. 2) and model deviations (Fig. 3).

### 4.3 Precipitation

Inter-model comparison of precipitation climatology simulation is presented in Fig. 4. Precipitation bias pattern of MPI-ESM experiment (Fig. 4e, f) generally resembles that of MPI-ESM-HR experiment at the annual scale in Stevens et al. (2013). ECHAM5-NEMO3.6 and ECHAM6-NEMO3.6 show similar bias patterns as that of MPI-ESM with variations in detail. The main errors in precipitation simulation are inadequate rainfall along the equatorial areas, flanked by excessive precipitation in subtropics of both hemispheres. The double ITCZ problem in summer is still remarkable in both MPI-ESM and ECHAM5-NEMO3.6, but it is ameliorated in ECHAM6-NEMO3.6 over South Pacific Intertropical Convergence Zone (SPCZ). As suggested by Stevens et al. (2013), the extent of positive precipitation bias over the southern tropical Atlantic and negative bias on tropical South America seems related to each other, because the values and scales of biases in ECHAM6-NEMO3.6



and MPIESM are simultaneously larger than those in ECHAM5-NEMO3.6 (Fig. 4a, c, e). Inter-model simulation differences may thus imply an important role of atmospheric dynamics simulation, for example storm-track and ITCZ, in reducing precipitation biases. Rainfall biases in winter exhibit larger dry biases in tropical Pacific, also flanked by wet biases in subtropics with significantly overestimation in South Africa and South Indian Ocean.

Model improvements of the precipitation simulation can be summarized as follows: The ECHAM6-NEMO3.6 reproduces the best overall precipitation climatology with the highest pattern correlation coefficient (Fig. 2). ECHAM6-NEMO3.6 model ameliorates double ITCZ problem over SPCZ in boreal summer, while ECHAM5-NEMO3.6 best decreases the wet bias over tropical Atlantic.

## 4.4 Surface Wind Stress

Wind stress biases in boreal summer (Fig. 5a, c, e) exhibit similar bias patterns among the three CGCMs, with anomalous anticyclonic circulations over North Pacific, South Atlantic and South Indian Ocean. There are prevailing westerly anomalies over the Southern Ocean that reinforce the anticyclones in mid-latitudes of southern hemisphere. The major difference among the three CGCMs is in tropical Pacific, where MPI-ESM features southerly (northerly) biases but ECHAM5-NEMO3.6 and ECHAM6-NEMO3.6 presents east (west) oriented biases north (south) of the equator. It implies that the excessive cold tongue

simulation in MPI-ESM has little to do with surface wind stress anomalies, which can drive the underlying sea water to across the date line in the other two CGCMs. The anticyclonic bias over North Pacific can affect the meridional overturning circulation and subsequent heat transport. Therefore, the opposite SST bias in North Pacific between ECHAM5-NEMO3.6 and ECHAM6-NEMO3.6 can be partly attributed to differences in wind stress and radiation simulation. In boreal winter, only westerly anomalies are still prominent over the Southern Ocean among three CGCMs, whereas model biases in other parts of

the world display arbitrary patterns. ECHAM5-NEMO3.6 still suffers from anticyclonic biases over North Pacific in winter, which has been largely diminished in ECHAM6-NEMO3.6.

The three CGCMs possess similar bias patterns at the global scale, without obvious improvements in wind stress simulation after changing component models. However, it is worth noting that the common SST bias over tropical Pacific may be ascribed to different physical mechanisms for MPI-ESM, because there only exist southward oriented wind biases.

## 25   4.5 Surface Ocean Current

The ocean current biases are mainly located in tropical areas (Fig. 6), in predominantly zonal directions for ECHAM5-NEMO3.6 and ECHAM6-NEMO3.6, but meridionally distributed for MPI-ESM. Anomalous currents move westward south of the equator but turn to the opposite direction in northern tropics during boreal summer, with more intensified currents in ECHAM5-NEMO3.6 than those in ECHAM6-NEMO3.6 (Fig. 6a, c). Whereas the MPI-ESM features southward (northward)

tilted biases south (north) of the equator to a larger degree than the other two CGCMs. The direction of ocean current biases generally agrees with that of wind stress biases, where poleward deflection can be attributed to Coriolis effects. Since the poleward motion is too strong in the MPI-ESM experiment, the ocean currents in subtropical North Pacific even turn to the



east. There are northward current anomalies in North Pacific for both MPI-ESM and ECHAM5-NEMO3.6, favouring more heat transport from subtropics to higher latitudes. Yet colder SST biases still exist over large maritime space in MPI-ESM experiment, which suggests an investigation on radiation budget and the meridional overturning currents that provide a full picture of most relevant oceanic processes. Biases in winter season are diminished to some degrees in tropical oceans, with little amplification of biases outside tropics in ECHAM5-NEMO3.6 and ECHAM6-NEMO3.6 experiments. But the MPI-ESM comes up with prominent northward biases in subtropical North Pacific (Fig. 6f). It may help to explain warm SST bias around Sea of Japan (Fig. 3f) and cold SST bias in the subtropical ocean.

Although the structure of model bias differs between MPI-ESM and the other two CGCMs, the bias amplitude has been substantially reduced especially in tropics by using NEMO3.6st as the oceanic component model. The marked deficit in the simulation of ocean currents in tropical Pacific may be related to more in situ SST biases of the MPI-ESM than ECHAM5-NEMO3.6 and ECHAM6-NEMO3.6.

## 4.5 Total Radiation

Figure 7 shows the total radiation biases in each CGCM, and it turns out that three CGCMs suffer from a similar bias pattern in boreal summer, with overheated areas around the Pacific warm pool and insufficient total irradiance over the tropical eastern Pacific, subtropical southern Indian Ocean and North Atlantic. The MPI-ESM and ECHAM6-NEMO3.6 ameliorate the bias amplitude in ECHAM5-NEMO3.6, which is likely due to parameterization updates in the middle atmosphere that result in a more realistic simulation of cloud radiation forcing. Underestimation of surface net irradiance over the eastern Pacific leads to cooler SST, which may exaggerate the cold SST bias in tropical Pacific by anomalous ocean currents. Since the bias amplitude in ECHAM5-NEMO3.6 is the biggest, which is consistent with statistical analysis in Taylor's diagram (Fig. 2), the consequent SST bias should be larger than at least one of the other two CGCMs that either has the minimum deviation (MPI-ESM) or has the same oceanic component model (ECHAM6-NEMO3.6). However, the SST bias ranking is completely the opposite, especially around tropical Pacific (Fig. 3a, c, e). It implies that oceanic dynamics may be the major source for cold SST bias in the MPI-ESM, according to assessment results of wind stress, surface currents and radiation, whereas momentum fluxes may be the predominant factor for the other two CGCMs. Biases in boreal winter transform into an approximately zonal-band structure, including overestimation in tropical areas and underestimation in mid-latitudes (Fig. 7b, d, f). Cold biases still remain in limited areas of tropical Pacific and Atlantic, while considerable deviations occupy most part of subtropical and mid-latitude areas for ECHAM5-NEMO3.6.

Model assessments on SST, precipitation and surface ocean currents clearly show some simulation improvements after changing the component models. Cold tongue biases are common among the three CGCMs, but the ECHAM5-NEMO3.6 presents warm SST bias in North Pacific with the opposite sign to other two models. Simulation of total radiation doesn't differ much in tropical Pacific, while the surface stress in the ECHAM5-NEMO3.6 and ECHAM6-NEMO3.6 contains intensified eastly anomalies but in the MPI-ESM biases appear in meridional direction. It implies that the momentum anomalies are accountable for cold SST bias in tropical Pacific for coupled models using NEMO, but biases in oceanic dynamics and radiation



are accountable for the same cold bias in the MPI-ESM. Further investigation is undertaken on circulations deep into the ocean and atmosphere to get a full picture of bias genesis during the course of coupling.

## 5. Circulation Patterns Relevant to SST Biases

### 5.1 Meridional Overturning Circulation

The importance of Meridional Overturning Circulation (MOC) to SST in coupled models has been proved in previous studies (Wang et al., 2014, Liu et al., 2016). Thus, the comparison of North Pacific MOC (NPMOC) (Fig. 8) between the CGCMs with the same atmospheric or oceanic component model can help to explain bias characteristics in relation to SST deviations. The ECHAM5-NEMO3.6 and ECHAM6-NEMO3.6 possess similar bias patterns overall, with intensified tropical cell and deep tropical cell that are mentioned in Liu et al. (2011) (Fig. 8a, b). Tropical cell enhancement is more significant in

the ECHAM5-NEMO3.6, so that the upwelling in the tropics and subsequent heat transport to mid-latitudes are more than those in the ECHAM6-NEMO3.6. Bias in the MPI-ESM experiment manifests itself to a larger degree (Fig. 8c), The piControl experiment result (available in http://esgf-node.llnl.gov/) is attached (Fig. 8d) to demonstrate that the prominent biases in MPI-ESM are not caused by increasing coupling frequency from one day to 4 hours. More contour lines appear in NPMOC bias distribution in piControl than the MPI-ESM setting in this paper (Fig. 8c, d), suggesting obvious improvements after decreasing

coupling interval. This is consistent with previous studies (Bernie et al., 2008; Ge et al., 2017). Enhancement of tropical cell in MPI-ESM is unremarkable, compared with that of ECHAM5-NEMO3.6, but deep tropical cell is significantly intensified that tropical upwelling is forced to become too strong. This may explain why tropical SST bias of MPI-ESM is more than 2℃ in boreal summer, without significant radiation errors and easterly anomalies in surface stress. Excessive upwelling in tropical Pacific, induced by intensified deep tropical cell of NPMOC, cools down the local SST. Unlike the ECHAM5-NEMO3.6 case,

where the SST cooling in tropical Pacific is driven by intensified tropical cell of NPMOC that transports more heat to mid-latitudes, the MPI-ESM simulation of tropical and subtropical cells does not differ much from SODA reanalysis data. The resulting poleward heat transport carried by NPMOC is not increased in the MPI-ESM experiment. Therefore, the SST bias in North Pacific remains negative (Fig. 3e) under the impact of Northern Hemisphere annular mode (NAM) and wind-evaporation-SST (WES) feedback, when cold SST biases appear in tropical and extratropical North Atlantic (Zhang & Zhao,

2015). The cold SST bias in North Pacific and North Atlantic for ECHAM6-NEMO3.6 can also be explained with the same reason. But for the ECHAM5-NEMO3.6 case, the poleward heat transport has been enhanced by intensified tropical cell to bring up the SST in mid-latitudes (Fig. 3c).

### 5.2 Vertical Structure of Atmospheric Circulation

On behalf of the atmosphere motion that accounts for cold tongue bias and strong easterly bias in surface currents, vertical circulation bias zonally averaged over South Equatorial Current (SEC) is given in Fig. 9. In consistence with surface wind

 

stress bias (Fig. 5), there are easterly anomalies in the lower atmosphere for ECHAM5-NEMO3.6 and ECHAM6-NEMO3.6 (Fig. 9a, b). With anomalous upward (downward) motion over the western (eastern) Pacific, the easterly anomalies are maintained by the anomalous Walker circulation across tropical Pacific. The easterly anomalies in lower atmosphere decrease significantly in MPI-ESM experiment (Fig. 9c), without strong downward motion in the eastern Pacific. Inter-model

differences in the easterly biases at low levels are consistent with the amplitude of NPMOC tropical cell, because the stronger surface winds collaborating with Coriolis effect push more sea water to move west and poleward.

It can be inferred from the relationship between the overall temperature biases and the magnitude of easterly anomalies in the lower atmosphere that warmer temperature bias is correlated with stronger easterly anomalies. Recalling precipitation biases over the SEC area (Fig. 4a, c, e), where ECHAM5-NEMO3.6 shows the driest situation and MPI-ESM presents the

wettest circumstance, the latent heat absorption in the course of evaporation turns out to be responsible for the temperature bias. More precipitation requires higher specific humidity that favours cloud formation, in which more latent heat of vaporization is taken up when water vapor mixing ration is increased. Since cumulus convection modulates changes in temperature, specific humidity and atmospheric circulation, it is most likely to be the predominant factor that shapes the inter-model differences.

The analysis on oceanic and atmospheric circulation has made it clear that the SST bias is consistent with meridional overturning circulation in North Pacific, driven by surface wind stress anomalies that are maintained by anomalous Walker circulation over the tropical Pacific. Cumulus convection process is found to be a major contributor to inter-model differences. To better understand the impact by changing each component model, it is necessary to quantitively analyse their contributions to simulation differences.

**6. Contribution of Each Component Model**

A total of 12 variables that are sensitive to air-sea coupling are selected to calculate pattern correlation with SST simulation differences in boreal summer. Correlations between model differences in SST and each variable with ECHAM6-NEMO3.6 minus ECHAM5-NEMO3.6, denoted as "AGCM" column in Table 1, indicate the effects of changing atmospheric component model. Similarly, correlation results with ECHAM6-NEMO3.6 minus MPI-ESM to represent the contribution of changing

oceanic component model are named "OGCM" in Table 1.

**6.1 Effects of AGCM Replacement**

For the AGCM case, net longwave radiation is ranked top, followed by surface evaporation (latent heat flux) and net shortwave radiation. It is noteworthy that the correlation coefficients of evaporation and latent heat flux are identical with approximations to 3 decimal digits. This is probably because evaporation directly influences the latent heat release of

vaporization.  With net longwave and shortwave flux ranked in top 3, it suggests that radiation budget is closely associated with SST inter-model differences. Changing the atmospheric component model from ECHAM-6.3 to ECHAM-5.4 affects



cumulus convection processes, including temperature, specific humidity and winds, which further alter the ground radiation fluxes through cloud radiation feedback. Noting that latent heat flux has a correlation value close to that of net longwave radiation, it can be assumed that cumulus convection changes sea surface winds and then surface evaporation, leading to differences in latent heat flux. Consequently, the SST is altered in tropical oceans through wind-evaporation-SST (WES)

feedback. The negative sign of correlations for longwave and latent heat flux (Tab. 1) suggests a contrary trend with SST variations. It is easy to understand because more latent heat and longwave dissipation cools down the surface sea water. Similarly, positive correlation for shortwave flux is due to more solar irradiance that brings up the SST.

To determine the physical processes responsible for simulation differences, changes in SST, surface winds, radiation budget and vertical circulation are plotted, respectively. It can be seen from Fig. 10d that negative deviations almost occupy

the northern hemisphere, including tropical Pacific and Indian Ocean, North Pacific and North Atlantic. First, the mechanisms behind SST deviations in tropical and subtropical oceans are discussed here. Deviations of 10m wind exhibit northerly anomalies east of dateline in southern tropical Pacific (Fig. 10b), where easterly winds prevail for summer climatology (Fig. 10a), hence the evaporation and latent heat absorption over the sea surface are enhanced that makes SST deviation colder than 1℃ (Fig. 10d). There are eastward oriented wind anomalies over subtropical Indian Ocean, subtropical Atlantic and some parts

of subtropical Pacific (Fig. 10b) in the opposite direction against climatology (Fig. 10a). Their superposition results in a decreased wind speed that reduces surface evaporation and latent heat flux, which finally keeps the SST warm for those sea waters. The SST deviations thus form a positive feedback (WES feedback) with changes of surface wind and evaporation, in accordance with signs of correlation in Table 1. Since the latent heat and surface wind differences are caused by replacing the AGCM, it is advisable to compare deviations in vertical circulation that may shed some light on corresponding physical

processes. An anti-clockwise Walker circulation accompanied by negative temperature deviations occupies central and eastern Pacific over the SEC area (Fig. 11a). Low-level easterly anomalies are consistent with surface wind distribution (Fig. 10b) because northerly flows in southern tropical Pacific that contributes to more evaporation are not taken into account in meridional average. It can be assumed that changing the AGCM alters radiation budget, and tropospheric temperature becomes colder so that more air currents cool down and sink by insufficient heat. Downward motion over the SEC area connects easterly

anomalies in the middle troposphere, which forms a complete Walker cell that further enhances low-level westerly anomalies.

To verify this assumption, differences in radiation budget with respect to net shortwave and longwave fluxes, latent heat and sensible heat are drawn in Fig. 12. It turns out that deviations in shortwave flux and latent heat are more significant than those in longwave and sensible heat fluxes. Differences in shortwave radiation can be attributed to cloud radiation feedback, which is induced by changes in cumulus convection after replacing the AGCM. Whereas the deviations in latent heat flux are

derived from surface wind anomalies, also affected by cumulus convection. For example, latent heat fluxes over subtropical oceans are negative (meaning less heat loss) around the areas (Fig. 12c) where surface wind anomalies oppose background climatology (Fig. 10a, b). The deviation patterns are congruent with the correlation signs (Tab. 1). More solar irradiance corresponds to higher SST of subtropical oceans in southern hemisphere, while less shortwave flux matches cold SST in central Pacific. Latent heat and longwave fluxes show reverse variations with fewer heat releases related to higher SST. Therefore, it



can be confirmed that the AGCM replacement first alters cumulus convection that modulates temperature, specific humidity and atmospheric circulation, which in turn accommodates cloud radiation feedback to a consistent change and affects the radiation budget.

## 6.2 Impacts of OGCM Replacement

For simulation differences after changing the OGCM, latent heat flux (evaporation) holds the biggest share in pattern correlation with corresponding SST differences (Tab.1), much bigger than net longwave flux ranked at the second place. It implies a predominant impact of surface evaporation by the OGCM, much greater than any other physical processes during model coupling. Differences in sea surface evaporation affect latent heat release and low-level cloud formation, which in turn alters the blocking effect on shortwave and longwave radiation. Sensible heat flux is ranked at the third place with correlation

value close to that of longwave flux, almost twice as large as shortwave flux (also ranked third when changing the AGCM). Compared with the rankings in the AGCM case, it can be ensured that the OGCM influences the simulation with different physical mechanisms than those by replacing AGCM. The effect of Cloud radiation feedback is not prominent, while changes of latent heat flux induce variations in low-level atmospheric circulation and SST. Consequently, sensible heat varies by heat conduction to keep pace with SST changes, and through convection to diffuse the heat up into the atmosphere. SST differences

can also lead to changes in longwave flux. The negative sign of correlations for latent heat, sensible heat and longwave flux (Tab. 1) suggests a contrary trend with SST variations. This suggests a different process than that drive the SST deviations through cumulus convection by replacing the AGCM.

Contrary to the AGCM case, low-level wind deviations in tropical Pacific (Fig. 10c) are in the same direction as the background climatology (Fig. 10a), which should increase surface evaporation through stronger near surface winds and then

results in colder SST through WES feedback. Nevertheless, warmer SST appears in most parts of tropical and subtropical oceans, laying waste to the assumption trying to explain SST deviation in terms of WES mechanism. Since the top 3 variables that are most relevant to SST deviations (Tab. 1) suggest changes in radiation budget, the only choice is to start from their deviation patterns (Fig. 13). Latent heat flux shows the most obvious change among the four radiation terms. The SST deviations of the tropical oceans (Fig. 10e) are generally positive, which corresponds to less latent heat loss. West of South

Africa, the latent heat flux is positive (meaning more heat loss) and is associated with cold SST deviations. Variations of net longwave radiation and sensible heat flux are also in accordance with their signs of pattern correlation (Tab. 1). It seems to suggest that surface evaporation plays a predominant role in the SST simulation differences by OGCM replacement. Because a more latent heat release due to increase in evaporation results in higher SST, and rising SST leads to more upward longwave flux suggested by its computation formula with 4 times temperature value. The associated sensible heat is magnified through

heat conduction and convection processes.

Atmospheric circulation accommodates itself to changes of radiation budget. An anomalous Walker circulation cell appears over the SEC area (Fig. 11b), with updraft over Philippine sea where positive SST deviations indicate a net radiation surplus. The upward flow rises to the upper-level troposphere and condensation becomes evident due to temperature drops. It



then diverts toward east and orients downward over eastern Pacific, which enhances surface westerly anomalies in the lower atmosphere. This circulation pattern is consistent with anomalous easterlies over tropical Pacific (Fig. 10c). Warm temperature deviations (Fig.11b) can be viewed as the manifestation of surface radiation surplus and latent heat release of condensation. So far, the analysis has made it clear that changing the OGCM affects the SST simulation through latent heat of evaporation.

Surface evaporation has a direct impact on the amount of latent heat release, which further alters the SST and low-level circulation. Sensible heat and longwave radiation vary accordingly. Under the impact of net radiation surplus, the temperature rises with upward motion over western Pacific, which diverts east and descends over eastern Pacific. This anomalous Walker cell drives low-level winds towards the west, leading to westerly anomalies over vast areas of tropical Pacific.

## 7. Summary and Discussion

In this study, two new CGCMs have been developed based on the coupling structure of MPI-ESM, namely ECHAM5-NEMO3.6 and ECHAM6-NEMO3.6. The new CGCMs show some improvements in the simulation of SST, precipitation and ocean currents compared with MPI-ESM. The ECHAM5-NEMO3.6 presents the best SST simulation in summer with the minimum cold tongue bias in tropical Pacific and no remarkable bias in North Atlantic, while ECHAM6-NEMO3.6 reproduces the best winter climatology with most of the biases less than 1 ℃. For precipitation simulation, ECHAM6-NEMO3.6 presents

the highest pattern correlation with observational counterpart and substantially ameliorates double ITCZ problem over SPCZ in boreal summer. Biases in surface currents and meridional overturning circulation are also considerably reduced in both ECHAM5-NEMO3.6 and ECHAM6-NEMO3.6. Wind stress bias patterns are alike in most areas among the three CGCMs, but MPI-ESM shows only poleward anomalies over tropical Pacific without strong easterly biases that are common in most existing coupled models. Furthermore, the ECHAM5-NEMO3.6 has much larger biases in total radiation than MPI-ESM,

whereas the latter presents the maximum deviation in SST simulation. Besides, the ECHAM5-NEMO3.6 shows warm SST bias in North Pacific, with the opposite sign to most CGCM biases at present time. These facts constitute evidence that suggests model errors of each CGCM are caused by different physical mechanisms, depending on the mesoscale processes in atmospheric and oceanic systems.

Meridional overturning circulation in North Pacific and vertical structure of atmospheric circulation are analysed for a

comprehensive understanding of bias genesis in each CGCM. Overestimation of tropical cell in the NPMOC transfers more heat to mid-latitudes and results in warm SST bias in North Pacific of the ECHAM5-NEMO3.6. Excessively strong deep tropical cell of the MPI-ESM intensifies the tropical upwelling that leads to the largest cold SST bias among all three CGCMs. The analysis on atmospheric vertical circulation over the SEC area has confirmed that momentum field plays a major role in the SST biases of ECHAM5-NEMO3.6 and ECHAM6-NEMO3.6, while oceanic processes and radiation budget are

responsible for the same cold SST biases in tropical Pacific. Since the inter-model differences are caused by changing component models, 12 surface variables that are sensitive to coupling processes are chosen to calculate pattern correlation with the SST differences between CGCMs with the same atmospheric or oceanic component model. The top 3 variables ranked in



the case of changing the AGCM are net longwave radiation, surface evaporation (latent heat flux) and net shortwave radiation. It is noteworthy that surface evaporation directly impacts on latent heat of vaporization, so that their correlation coefficients are identical with approximation to 3 decimal digits. Differences in SST, radiation budget and atmospheric circulation are analysed, and it is confirmed that the AGCM replacement first militates in the alteration of cumulus convection including

temperature, specific humidity and atmospheric circulation, which in turn changes SST through WES feedback and affects the radiation budget through cloud radiation feedback. For the OGCM replacement, latent heat flux (evaporation) holds the maximum pattern correlation with SST variation, much larger than that of net longwave flux at the 2$^{nd}$ place and sensible heat at the 3$^{rd}$ place. Through analysis on circulation and radiation terms, it has been clear that latent heat of evaporation plays a predominant role in the SST differences after changing the OGCM. The SST and low-level circulation vary according to the

amount of latent heat release, which in turn alter the longwave radiation and sensible heat through conduction and convection processes. Attributing simulation deviations to latent heat in the OGCM case is consistent with Cao et al. (2015), which points out that amplitude and meridional variability of latent heat flux over Pacific are the most diverse in CMIP5 models.

It is noteworthy that the SST deviations by changing the OGCM and AGCM are nearly the opposite (Fig. 10d, e), suggesting that reverse transformation of model bias can be realized through mechanisms unveiled in this paper. Although the

top 3 coupling variables that are most relevant to SST deviations after changing the AGCM or OGCM are radiation terms, the physical mechanisms behind the opposite SST variations are different. AGCM replacement affects cumulus convection that eventually changes momentum field and radiation budget over the sea surface, while OGCM replacement alters sea surface evaporation that results in latent heat variations and consequently leads to readjustment of radiation budget. It implies that one can pursue simulation improvement of the cold tongue from two aspects: 1. Improve the cumulus convection scheme in the

AGCM, 2. Ameliorate errors of surface radiation budget in the OGCM, especially for latent heat. With better cumulus convection scheme, easterly anomalies over the tropical sea surface can be reduced significantly, which will cut down the latent heat absorption of more evaporation induced by stronger surface winds, and consequently break down the WES feedback that amplifies the cold SST bias. For the OGCM part, better treatment of surface evaporation and radiation budget can reduce the cold SST bias in relation to net radiation deficiency.

For strong easterly bias over eastern Pacific in ECHAM6-NEMO3.6, also common in the most CGCMs, the MPI-ESM instead shows poleward bias, which suggests that OGCM replacement can also diminish this bias through coupling processes. It is easy to see that an anomalous Walker circulation with counter-clockwise rotation will appear over the tropical Pacific, if net surface radiation is warmer in the east and colder in the west. With proper OGCM configuration, sea surface evaporation can initiate this radiation anomaly, and the resulting Walker circulation will decrease the easterly wind speed. The mechanisms

illustrated in this study provides a new vision of model bias origin, which is heuristic for model improvement researches and practices.



**Code and data availability**

The model source code is available from the authors upon request. The experimental data can be found in https://doi.org/10.5281/zenodo.1306338 (Gui et al., 2018).

The reanalysis data used in this study can be downloaded from the following websites:

1.  The Hadley Centre SST data is downloaded from https://www.metoffice.gov.uk/hadobs/hadisst/.

2.  The GPCP precipitation data is downloaded from  https://precip.gsfc.nasa.gov/.

3.  The ERA-Interim monthly reanalysis data is available at http://apps.ecmwf.int/datasets/.

4.  The SCOW wind stress data is available at http://cioss.coas.oregonstate.edu/scow.

5.  Surface radiation reanalysis data of CERES EBAF-Surface Ed4.0 is downloaded from https://ceres.larc.nasa.gov.

6.  The SODA3.3 ocean current reanalysis data is downloaded from: http://www.atmos.umd.edu.

The MPI-ESM piControl experiment data is available at http://esgf-node.llnl.gov.





**Acknowledgements**

This work was supported by the National Key Research and Development Program of China (2016YFA0601600), and the National Natural Science Foundation of China (U1502233 and 41565002). It is also co-supported by Yunnan University's Research Innovation Fund for Graduate Students (YDY17019).



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




**Table 1: Pattern Correlations of surface variables and SST deviation**

| Model Replacement<br>Variables | AGCM | OGCM |
|---|---|---|
| Surface zonal currents | 0.006 | -0.002 |
| Surface meridional currents | 0.0158 | -0.018* |
| Sensible heat flux | -0.068* | -0.269* |
| Latent heat flux | -0.262* | **-0.393**\* |
| Mean sea level pressure | -0.136* | -0.009 |
| 10m zonal wind | 0.018* | -0.010 |
| 10m meridional wind | -0.025* | 0.045* |
| Surface albedo | -0.101* | -0.022* |
| Net shortwave flux | 0.192* | 0.138* |
| Net longwave flux | **-0.269**\* | -0.283* |
| Evaporation | -0.262* | **-0.393**\* |
| Precipitation | 0.100* | 0.154* |

Asterisk (*) denotes the correlation coefficients passing the significance test above 99.9% confidence level. A larger number of grid points are involved that makes the threshold value relatively small. Numbers in boldface denotes the maximum
5   absolute correlation value for the AGCM or OGCM replacement.





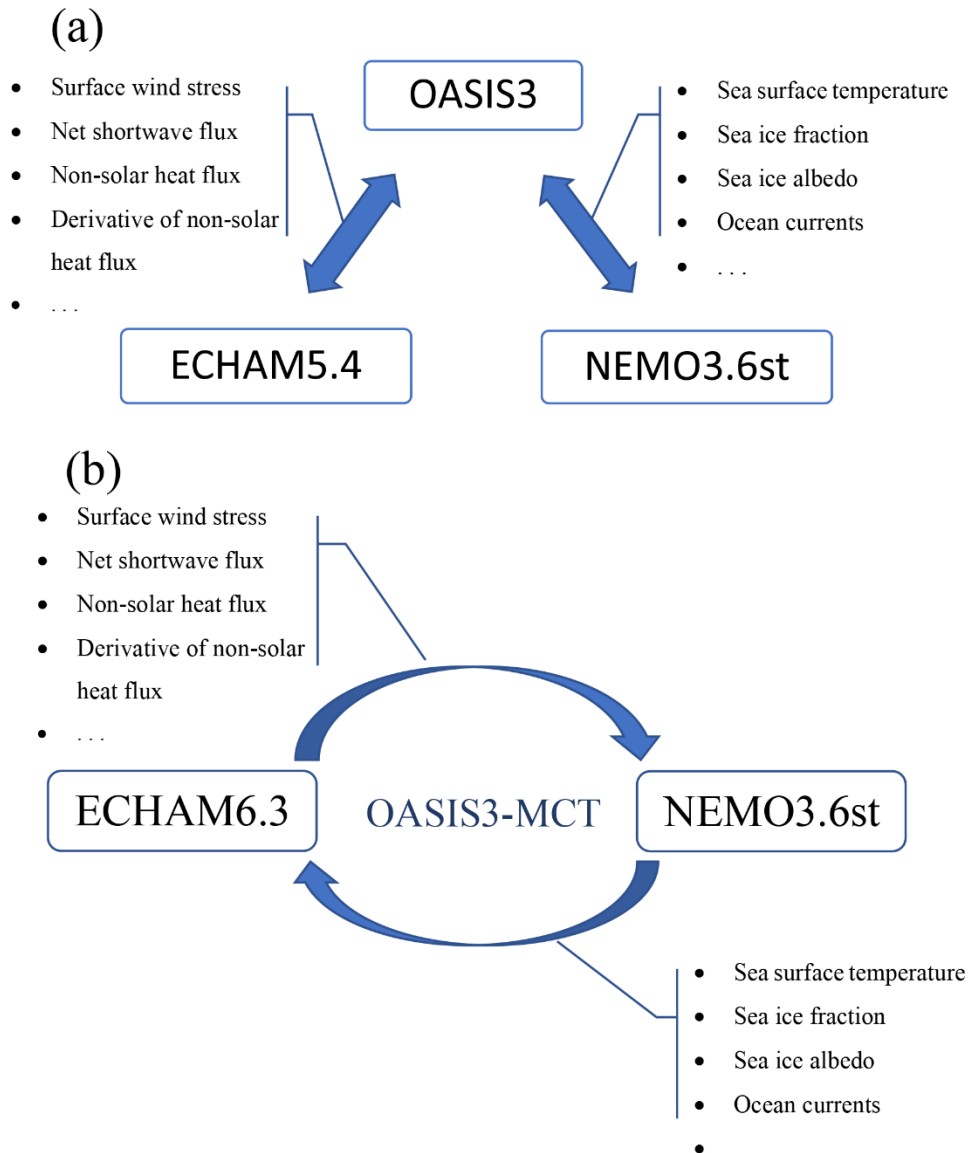

**Figure 1: Schematic structure of ECHAM5-NEMO3.6 (a), and ECHAM6-NEMO3.6 (b).**





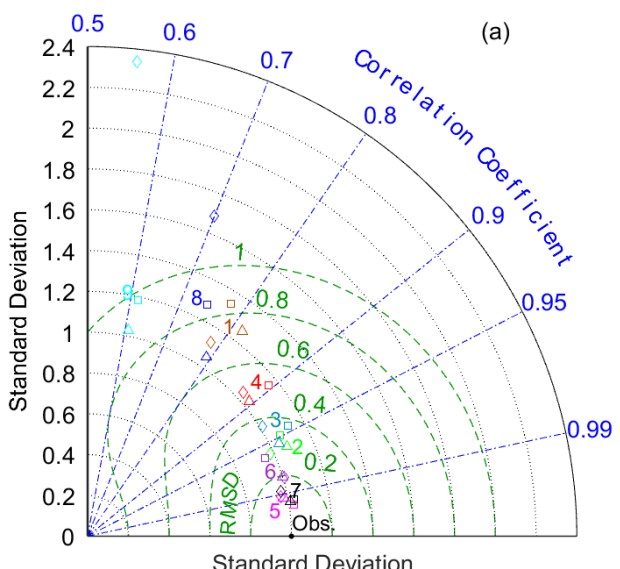 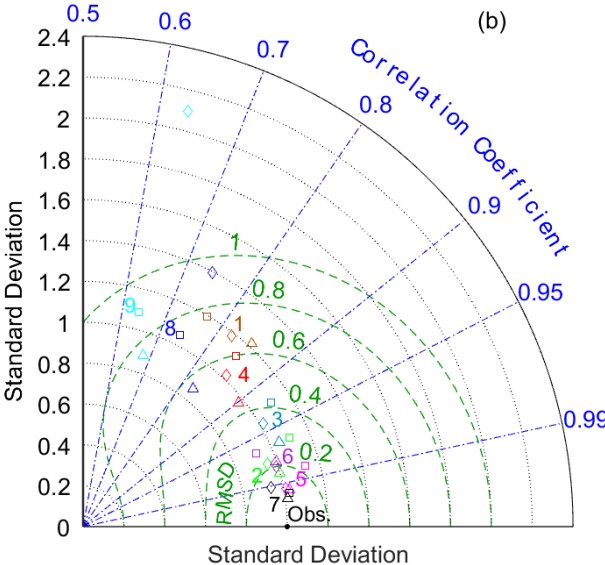

**Figure 2: Taylor diagram that exhibits a statistical comparison between the simulations and observations of nine selected variables in summer (a) and winter (b). Each number represents one variable: (1) precipitation, (2) mean sea level pressure, (3) zonal winds at 10m height, (4) meridional winds at 10m height, (5) 2m temperature, (6) total radiation flux (net shortwave plus net longwave), (7) SST, (8) sea surface zonal currents, (9) sea surface meridional currents. Upward-pointing triangles, squares and diamonds, respectively, represent the ECHAM6-NEMO3.6, ECHAM5-NEMO3.6, and the MPI-ESM results.**





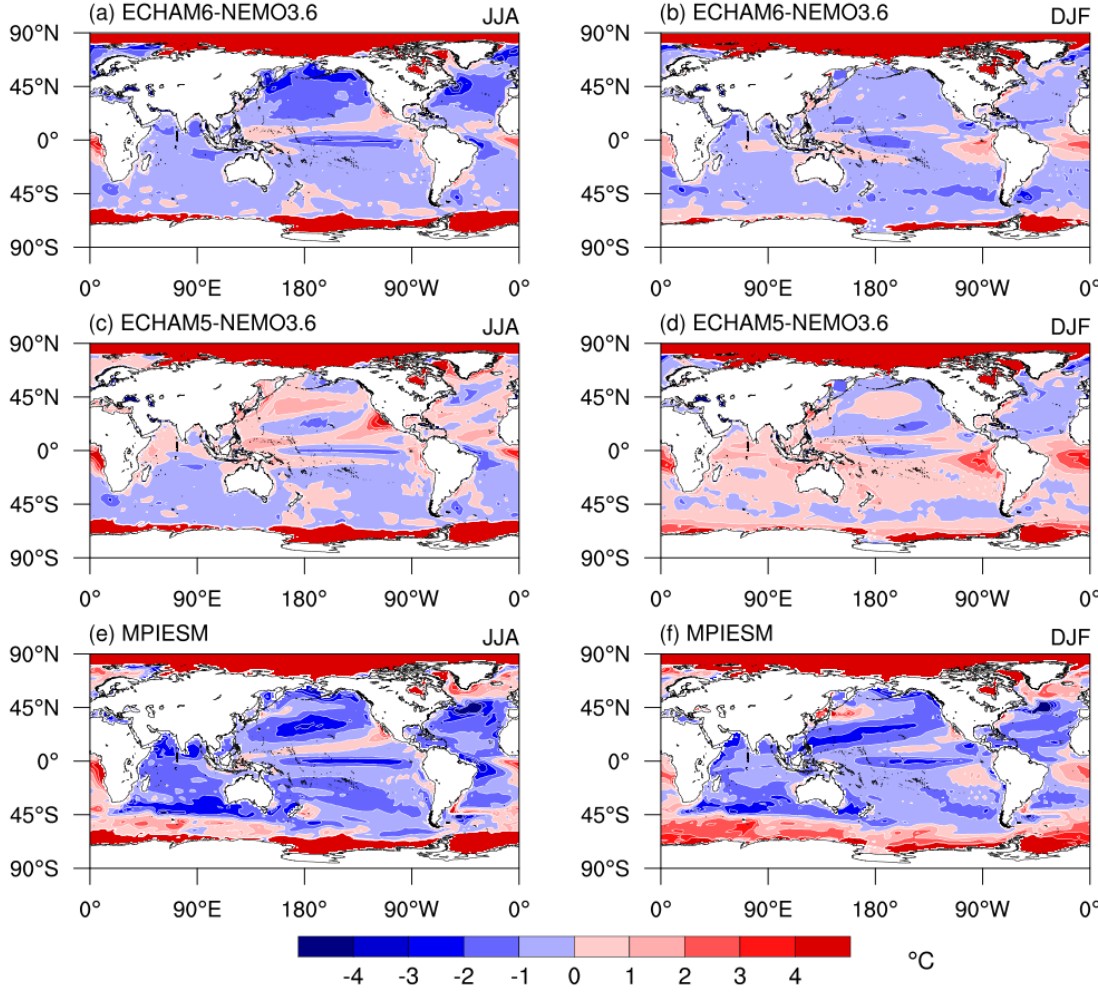

**Figure 3: Biases of the SST simulation in summer (left column) and winter (right column) corresponding to each CGCM: (a, b) ECHAM6-NEMO3.6, (c, d) ECHAM5-NEMO3.6, (e, f) MPI-ESM.**





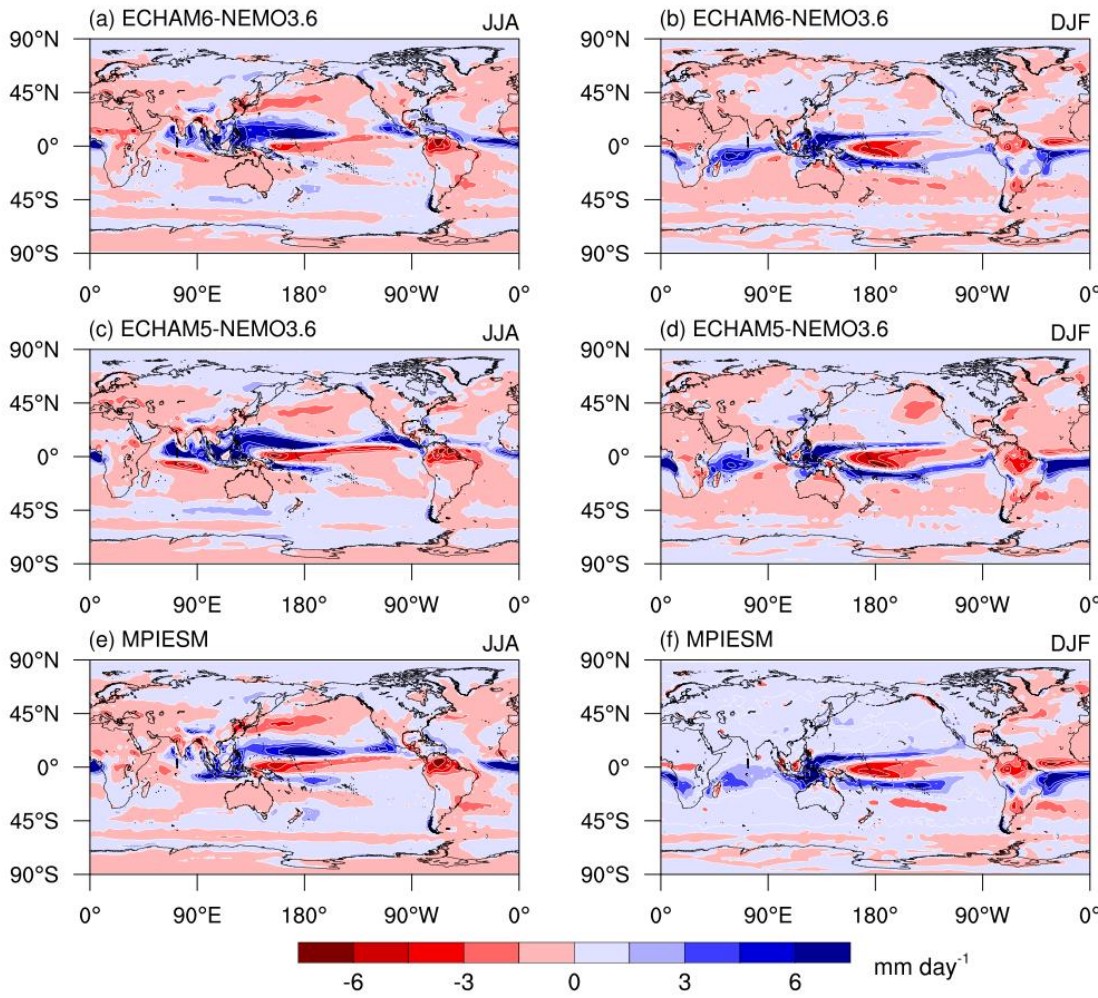

**Figure 4: The same as Fig. 3 but for simulated precipitation climatology.**





**Figure 5: The same as Fig. 3 but for simulated surface wind stress.**



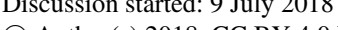



**Figure 6: The same as Fig. 3 but for simulated surface currents.**







**Figure 7: The same as Fig. 3 but for simulated total radiation.**

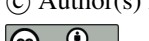



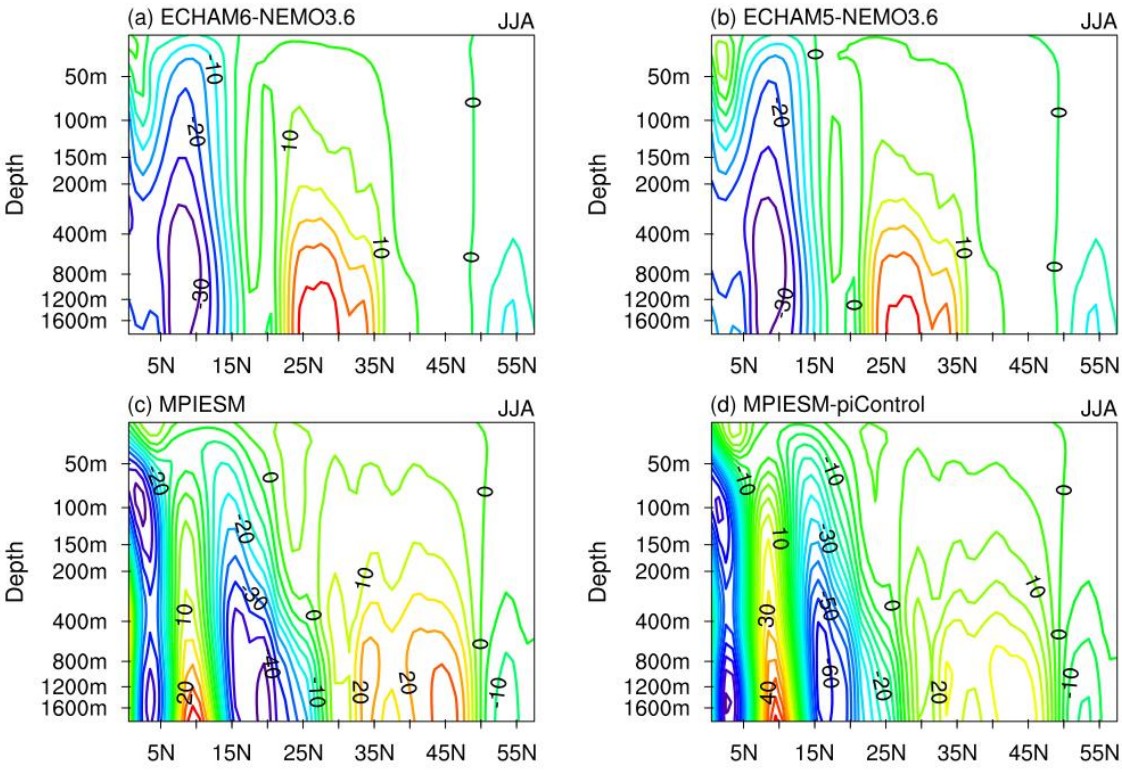

**Figure 8: Model biases of summer climatology of meridional overturning circulation simulation in North Pacific of (a) ECHAM6-NEMO3.6, (b) ECHAM5-NEMO3.6, (c) MPI-ESM, (d) MPI-ESM piControl.**



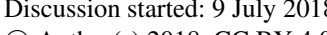


**Figure 9: Model biases of the vertical structure of atmospheric circulation (vector) and the temperature (contour) over SEC area in summer for (a) ECHAM6-NEMO3.6, (b) ECHAM5-NEMO3.6, (c) MPI-ESM**





**Figure 10: Summer climatology of 10m wind for (a) observation, (b) model differences between ECHAM6-NEMO3.6 and ECHAM5-NEMO3.6, (c) model differences between ECHAM6-NEMO3.6 and MPI-ESM. SST simulation differences for (d) between ECHAM6-NEMO3.6 and ECHAM5-NEMO3.6, and (e) between ECHAM6-NEMO3.6 and MPI-ESM.**

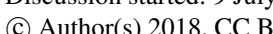



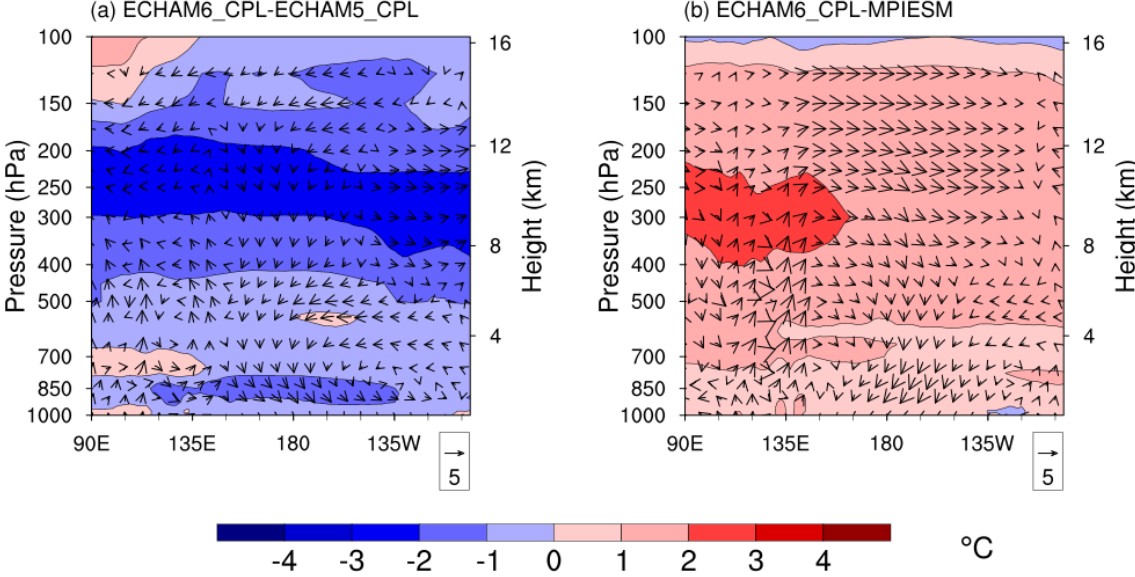

**Figure 11: Simulation differences in the vertical structure of atmospheric circulation and the temperature over the SEC area: (a) between ECHAM6-NEMO3.6 and ECHAM5-NEMO3.6, (b) between ECHAM6-NEMO3.6 and MPI-ESM.**

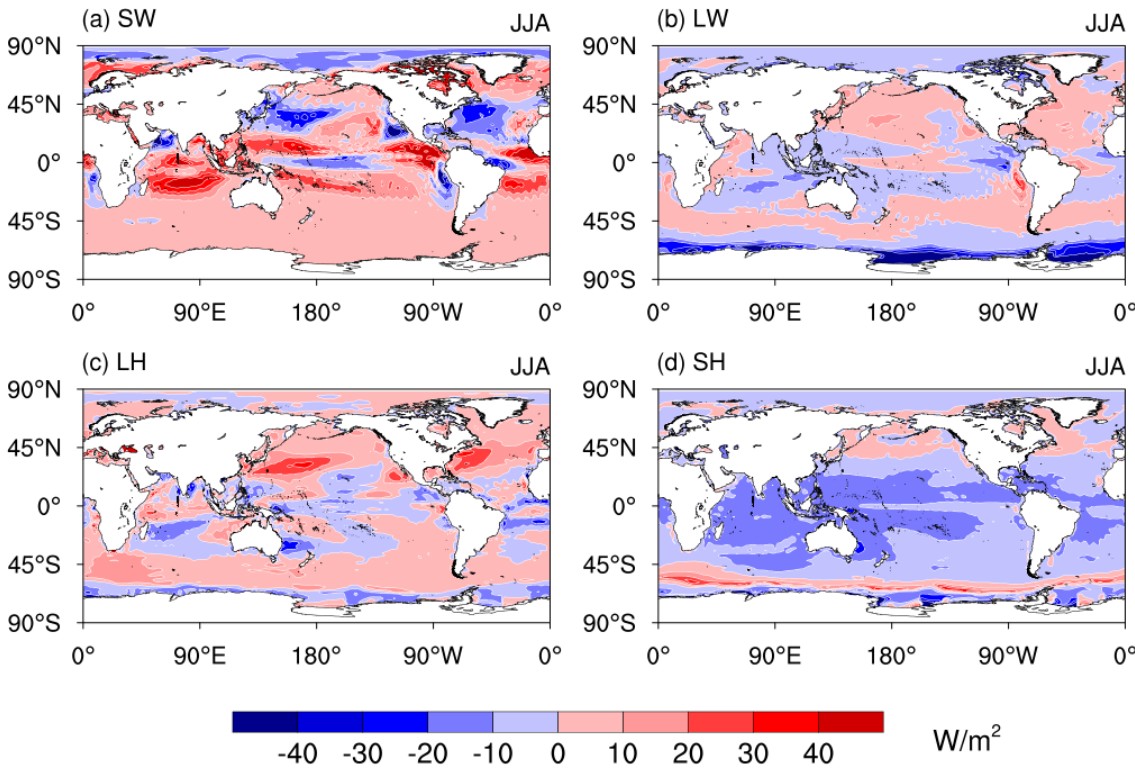

**Figure 12: Simulation differences in radiation budget between ECHAM6-NEMO3.6 and ECHAM5-NEMO3.6.**




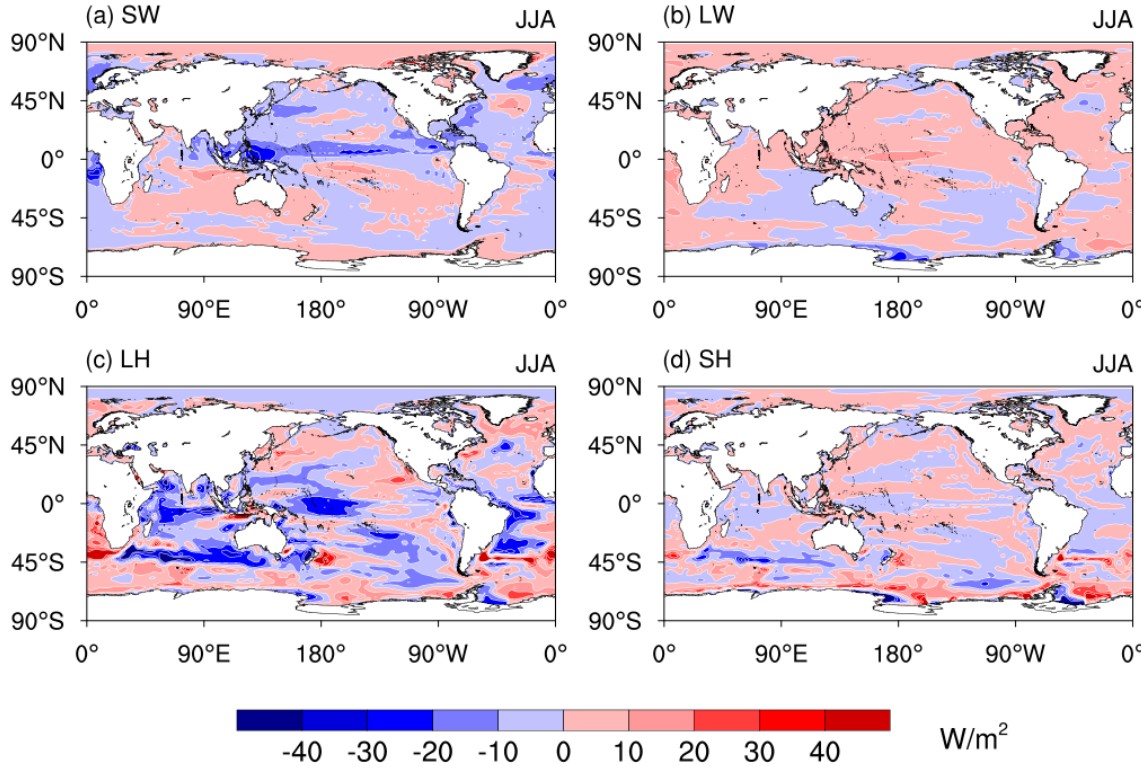

**Figure 13: Simulation differences in radiation budget between ECHAM6-NEMO3.6 and MPI-ESM.**