# Peer review of "Simulation Improvements of ECHAM5-NEMO3.6 and ECHAM6-NEMO3.6 Coupled Models Compared to MPI-ESM and the Corresponding Physical Mechanisms"

_Geoscientific Model Development, 2018_

## Short Comment (SC1) · 9 Jul 2018

Dear authors,

As explained in https://www.geoscientific-model-development.net/about/manuscript_types.html. GMD is encouraging authors to upload the program code of models (including relevant data sets) as supplement or make the code and data of the exact model version described in the paper accessible through a DOI (digital object identifier). In case your institution

does not provide the possibility to make electronic data accessible through a DOI you may consider other providers (eg. zenodo.org of CERN) to create a DOI. Please note that in the code availability section you can still point the reader to how to obtain the newest version. If for some reason the code and/or data cannot be made available in this form (e.g. only via e-mail contact) the "Code Availability" section need to clearly state the reasons for why access is restricted (e.g. licensing reasons).

Yours,

Astrid Kerkweg

---

## Author Comment (AC1) · 9 Jul 2018

Dear Astrid Kerkweg,

Thank you very much for reminding us about the required contents in the "Code Availability" section. We will add an explanation of restricted access to model source code.

Sincerely,

Gui Shu

---

## Short Comment (SC2) · 23 Jul 2018

I discovered two minor errors in the text:

1. Section number of "Total Radiation" should be 4.6.

2. Figure numbers of Figure 7 is wrong.

I have fixed the problems in the manuscript, and the figure 7 with correct figure numbers is uploaded.

[Figure]

**Fig. 1.**

---

## Short Comment (SC3) · 31 Jul 2018

The ECHAM5-NEMO3.6 presents a better SST simulation than that of the ECHAM6-NEMO3.6 and the MPI-ESM. Is it possible that the NEMO OGCM is intrinsically more compatible with the ECHAM5.4 than with the ECHAM6.3? Since the MPIOM is not so famous compared with NEMO, it seems to suggest that better OGCM reproduces better SST simulation, which has little to do with the air-sea coupling processes.

---

## Author Comment (AC2) · 1 Aug 2018

Dear Prof. Wang:

Thank you very for your post. It is certainly possible that the NEMO3.6 is more suitable for ECHAM5.4, even though the surface flux biases are larger than those of the ECHAM6.3 in the coupled experiment. However, the SST simulation doesn't all depends on the OGCM. It has been confirmed that the OGCM does not exhibit excessive cold tongue bias in the standalone historical runs. Apart from other studies that probe

into the SST variations with different coupling systems, our test results also suggest that the air-sea interaction reshape the bias pattern to a large extent. We have tested the standalone NEMO performance with historical input files and the SST bias becomes much smaller in tropical areas, but significant cold SST biases appear in the North Pacific and North Atlantic, similar to that of the ECHAM6-NEMO3.6 (See the following figure). It serves as an evidence to prove that air-sea coupling with substantial amount of heat, momentum, and condensation fluxes can redirect the model performance to a new equilibrium state that can be out of the original perspective.

Sincerely,

Shu Gui and coauthors
* * *
[Figure]

[Figure]

(a) Model — JJA

(b) Model — DJF

(c) Obs. (HadISST) — JJA

(d) Obs. (HadISST) — DJF

°C

-4  0  4  8  12  16  20  24  28

(e) Diff. — JJA

(f) Diff. — DJF

°C

-4  -3  -2  -1  0  1  2  3  4

**Fig. 1.**

---

## Short Comment (SC4) · 2 Aug 2018

The paper is novel, interesting and creative. The authors tried to reveal the causes of the improvements of the latest GCGM (i.e., ECHAM6-NEMO3.6) in simulating the SST, precipitation, and ocean currents by going beyond simply parameterization schemes, thus the results have great scientific implications. My question is that how to explain the dominant role of the latent heat of evaporation in changing SST when regarding to the OGCM replacement, since our conventional opinion mainly focus on the differences in ocean dynamics and thermodynamical structure when we use discrepant ocean

models.

---

## Author Comment (AC3) · 3 Aug 2018

Dear Dr. Ying Jun:

Thank you for your thoughtful comment. We agree that the SST and oceanic dynamics are usually considerred in the air-sea coupling systems for climate response to an external forcing. The atmosphere affects the oceanic dynamics through momentum and radiative fluxes, while the ocean transfers SST, surface currents and sea-ice fraction to the air-sea interface. However, the air-sea interaction is based on the non-linear

systems of atmospheric and oceanic dynamics in each component model, which can go beyond normal understanding of the key coupling processes within.

As we have confirmed through correlation analysis, the latent heat flux (evaporation) holds the maximum pattern correlation with SST variation. This can be explained with newtonian cooling effects. After replacing the oceanic component model, the ocean velocity differences induce the ocean temperature changes, which in turn modulates the sea surface evaporation through newtonian cooling. Due to time limitation, we haven't further probed into the topic in this regard, i.e. which part of the changes in the oceanic model accounts for the discrepancies in latent heat flux. We shall throughly discuss it in another paper.

King Regards,

Gui Shu and co-authors

---

## Referee Comment (RC1) · Anonymous Referee #1 · 28 Aug 2018

The paper describes the effect of changing both the atmospheric and oceanographic components of a coupled general circulation model by trying to disentangle the different effects of upgrading the ocean model and upgrading/changing the atmosphere model.

I really like the paper. It is well-written and the underlying analysis of upgrading the components contribute to modify the coupled solutions is interesting and innovative. So, I am happy to recommend this for publications with minor revisions. Some of my comments below really more suggestions than required revisions.

[Figure]

**1 General questions/comments:**

The authors should probably comment on how ocean (atmosphere) models might initially be tuned by running OMIP (AMIP) runs with fixed forcing fields (SST/sea-ice) to minimize biases etc in uncoupled mode. An existing coupled model could have been tuned to give a good coupled performance, so upgrading a component might require a retuning of the remaining component to get the best possible coupled performance. It could be good to discuss this briefly in the paper.

The authors briefly mention the sea-ice model in MPIOM but this is not mentioning of the sea-ice model in NEMO. Since there are large differences in biases between the different model combinations in high latitude on Figure 3 then it might be appropriate to at least mention those differences.

While I am aware that it is common in the climate modelling community to talk about reanalysis data as observations then it is not true. Both atmospheric and oceanographic reanalyses have errors due both lack of "real" observations until recent times and deficiencies in the data assimilation techniques used.

**2 Detailed comments:**

Page 3 line 24: The "Mogensen et al 2012b" reference is not relevant for data assimilation.

Page 3 line 26: ORCA2 is just one of many NEMO configurations, so it should be mentioned as a choice made by the authors and not all something all NEMO based system uses.

Page 3 line 27: The 31 levels of ORCA2 is unevenly distributed, so this should be mentioned for consistency with the MPIOM remark on page 4.

Page 5 line 1 to 3 (and Figure): The authors should be more precise on all fields exchanged between ECHAM5/6 and NEMO.

Page 5 line 5: The 1200 seconds time step for NEMO quite short compared to the reference time step 5760 second time for ORCA2. Any reason for choosing this ocean time step?

Page 5 line 22: The grid specifications for MPIOM was already mentioned in section 2.1.1 on page 4 line 6, so this should not be repeated.

Page 6 line 18+24: Reanalyses are not observations as mentioned above.

Page 6 line 29: ORCA2 only has refinement in latitude from 2.0 to 0.5 degree, but not in longitude. The text seems to imply that ORCA2 has 0.5 degree resolution in both latitude and longitude which is not true.

Page 8 line 11: ECHAM5-NEMO3.5 should be ECHAM5-NEMO3.6.

Page 10 line 9-11: Would the fact that NEMO3.6st and MPIOM uses different grids contribute to the differences in SST biases?

Section 5.1+Figure 8: I assume that the MOC is compared with the SODA reanalysis? Maybe this should be I am aware of several inter-comparisons of reanalysis of the AMOC which show quite different results compared to the Rapid array, so I would expect quite some uncertainty in the SODAY reanalysis. Maybe this could be discussed briefly?

Section 5.2+Figure 9: Again, mention that this is against ERAI reanalysis and not observations. Also it might worth being very specific on which area the authors define as the SEC. Maybe a map on Figure 9 with a box could be useful.

Section 6: Unless I missed something then this section focus exclusively on JJA with no mentioning of DJF. There are some differences in SST biases on Figure 3 for all models, so maybe it would be worth mentioning if the authors would expect the conclusions

to also be true for the boreal winter?

Page 14 line 12: "Cloud" should be "cloud".

Figure 1: Please add all fields exchanged between the atmosphere and ocean model.

Figure 2: I am not sure that a Taylor diagram is the best way to present the data. Some of the data points are very small and difficult to see. Maybe the authors could think of alternatives for presenting the information?

All maps on Figure 3,4,5,6,7,9,10,13: I suggest moving the longitudes from 0 to 360 degrees to 20 to 380 degrees since it will mostly avoid cutting the Atlantic Ocean in two.

Figure 5+6+10: I suggest moving the legends with the arrow outside the plots.

Figure 8: Adding a colour legend might be useful since the minus sign on the contours are a bit hard to see.
* * *

---

## Referee Comment (RC2) · Anonymous Referee #2 · 3 Sep 2018

Recommendation: Accept pending minor revisions

General comments: The authors compared two new GCMs (ECHAM5-NEMO3.6 and ECHAM6-NEMO3.6) to MPI-ESM (ECHAM6-MPIOM), showing improvements and differences in biases in each model.

The paper in well written and only has a few issues that I would hope the authors would address before publication.

1) The authors compared mostly the Pacific ocean and concentrated on tropical Pacific.

One of the stark improvements lie in the North Atlantic, especially where the cold bias in the subpolar region seen in MPI-ESM has been ameliorated. It would be extremely beneficial for the community to discuss this too, and show how the AMOC may be different between these different CGCMs.

2) Changes in mean state from changing component model provide simple but necessary information. However, it would benefit readers and the community if more information such as variability of the system is assessed, such as variance of variables that the authors have covered, and maybe even power spectrum of ENSO and SAM.

3) Section 6.2: Please be aware that correlation does not indicate intensity or magnitude of relationship. Fig. 13 shows that shortwave radiation has a much large magnitude and can thus possibly have a greater effect on SST even though it has a smaller correlation value compared to sensible heat and longwave radiation. This does not change the proposed mechanism, but SW radiation should be incorporated into the explanation.

4) Along the lines of pattern correlation being used to rank the importance of contributing factors. Since the region of focus is the Pacific, the pattern correlation should does be performed over the Pacific rather than globally. This should not change the concluding results, but it may provide some differences in the correlation and add robustness to your estimates.

5) Please state significance test for pattern correlation, particularly how the effective degree of freedom is computed or decorrelation length scale used. Having a significance at 99.9% level for a correlation at 0.018 is hard to reckon with.

6) Page 6, line 13: Which year of external forcing (like $CO_2$, greenhouse gases, aerosol, etc.) was used for the piControl run to obtain equilibrated state? Is this the same (similar) forcing as seen during observation period? If the piControl uses forcing that is quite different from observations (e.g. excess of 100ppm of $CO_2$), then one would have to account for these differences when computing biases with respect to

observations.

7) page 12, line 7-8: Rather than inferring the relationship, the authors can easily compute the pattern correlation and quantify the correlation.

Minor comments: 1) page 6, line 11: please provide the actual websites.

2) page 8, line 31-32: Been staring at Fig. 4 but am not able to see "positive precipitation bias over the southern tropical Atlantic and negative bias on tropical South America". Maybe you meant "negative bias over southern tropical Atlantic and positive bias on tropical South America"? Please indicate which specific figures you are referring to. 3) page 8, line 32: "... seems related to each other, because the values and scales of biases in ... " Please explain, it is not obvious how this is the case.

4) Fig. 5: Please provide magnitude difference in surface wind stress in colour. This would make it easier for readers to see extent of anomalies. 5) page 9, line 15: Change "sea water to across" to "sea water across"

6) page 10, line 13: Change "it turns out that three CGCMs" to "it turns out that all three CGCMs"

7) page 11, line 11: Change "," to "." 8) page 11, line 31: Change "In consistence" to "Consistent"

9) page 13, line 3: Change "it can be assumed" to "it would suggest"

10) page 15, line 22: None of the CGCMs used on this paper resolved mesoscale processes, so the subsentence "depending on mesoscale processes in the atmospheric and oceanic systems" cannot be concluded from this study and should thus be omitted.

11) Please state units for Fig. 5, 6, 8, 9, 10, 11.

---

## Short Comment (SC5) · 3 Sep 2018

Comments on behalf of the FOCI development team at GEOMAR

General comments

The authors try to disentangle the oceanic and atmospheric impact on the SST bias in a coupled GCM by replacing the ocean and atmosphere components separately. While the idea seems very innovative at a first glance, the paper unfortunately lacks a discussion on the difficulties and side effects of replacing a model component in a

coupled GCM. The large effect of coupled ocean-atmosphere feedbacks onto the development of the SST bias is not discussed properly. The replacement of a component in a coupled GCM is not as straightforward as the paper suggests in its current form, and the conclusions drawn are hence questionable.

Some more information on the experimental setup would be desirable. How is the ocean initialized, e.g. are World Ocean Atlas ('Levitus') data used? This is in particular of interest since the authors claim that their model ocean is in equilibrium after only 100 years of spin up whereas other modeling groups perform multi-century (Delworth et al., 2006) or even multi-millennial (Müller et al., 2018) spin-up runs to significantly reduce the temperature drift in the ocean where clearly a drift is still visible after 300 or 500 years (Delworth et al., 2012, Fig. 1; Delworth et al., 2006, Fig. 3). Such a drift is best visible in timeseries of the global mean temperature for the surface but also deeper ocean layers, which unfortunately are not provided by the authors and should be added. It is essential for other modelling centres to provide at least a number or even better a timeseries of the TOA radiation (im)balance.

Other major concerns are:

1. For both the Atmosphere (ECHAM5 and ECHAM6) and the ocean models (NEMO, MPI-OM) very little information is provided on the technical details except for the configuration of the coupler (e.g., which parametrizations are active, which model options are switched on or off, how are the model components initialized, is nudging or restoring used in the ocean or atmosphere, model tuning)

2. The most prominent feature of no North Atlantic cold SST bias in a 2° ocean model coupled to ECHAM5 in their ECHAM5-NEMO3.6st configuration is not properly discussed. This bias has been around for decades in coupled climate models at the given resolution, and numerous papers discuss it. None of this work is mentioned or compared to. See also our detailed comments on Figure 3 below.

3. No figures or information on the stability of the control simulation (e.g., timeseries of

surface air temperature, TOA radiation budget, etc.) are provided which are crucial to evaluate coupled GCM performance.

4. A new coupled model system is presented and key ocean parameters such as the Atlantic Meridional Overturning circulation (MOC) or important coupled atmosphere ocean variability patterns (e.g., ENSO, NAO), their difference amongst the different GCM configurations and their possible impact on the SST bias are not discussed and should be added to the paper.

5. In our opinion, the pattern correlation method (table 1, with pattern correlations always below 0.4) cannot be used to explain the inter-model differences as it completely ignores both the physical dependencies of the parameters used in the correlation as well as the impact of ocean dynamics and coupled ocean-atmosphere feedbacks onto the SST bias in a GCM. In addition the presented pattern correlation values are very low.

6. Unfortunately, no information about the setup of the land component in the new coupled GCM is provided. We assume it is using JSBACH, the new land model component within ECHAM6. Is JSBACH running interactively? Why are the pattern correlations for albedo that weak? The interpretation of simulated precipitation is questionable, as differences in the extra tropics are not really visible (scale inappropriate). Additionally, the paper lacks also information about 2m temperatures (also referred to as SAT - surface air temperature) simulated over land.

7. No information on sea ice in the different GCM configuration is provided. To be able to judge the SST differences between the different model configurations properly, some information such as sea ice extent and sea ice thickness should be added to the paper.

8. Some of the presented model configurations (ECHAM5 coupled to NEMO) have been developed almost 10 years ago (Park et al., 2009), and have been used extensively during the last 10 years including work on the SST bias (Wahl et al., 2009,

Harlass et al., 2015). Unfortunately, none of this work is mentioned in the introduction or in the discussion.

Detailed major comments:

1. Page 5, line 31: The model experiments are performed using the piControl standard scenario of the MPI-ESM (p. 5, line 31). Most of the observational or reanalysis products used to compute model biases cover more recent periods than pre-industrial (1850 or 1870). Which reference period is used to compare the model runs to?

2. Page 6, line 1: "Model initialization is started from the climatology basic state recalculated with the AMIP run input data from 1981 to 2010." This suggests that the initial conditions in the atmosphere are based on a climatology calculated from AMIP simulations. Due to the chaotic nature of the atmospheric circulation, the choice of initial conditions of the atmosphere are not crucial for the performance of a coupled GCM. Hence we would strongly suggest to provide more information on the ocean initial conditions (see also our general comment above).

3. Figure 3c/d clearly shows the absence of a North Atlantic (NA) cold bias in a 2° ocean model coupled to a coarse resolution atmosphere (ECHAM5/6-NEMO3.6st configuration) which is a very striking result. The NA cold bias has been around for decades in coupled climate models at the given resolution, and numerous papers discuss it (e.g. Zhang and Zhao, 2015). None of this work is mentioned or compared to. Please provide details (e.g. namelists for both ocean and atmosphere model) to make it possible for other modelling centres to understand how you were able to achieve this immense improvement in the North Atlantic.

4. Figure 3: The authors note that the SST bias is largest in the polar regions exceeding 4degC as shown on Figure 3. This large bias clearly coincides with sea-ice coverage. There, the HadISST data set, which is used as a reference here, provides temperatures near the freezing point of sea water (∼-1.9degC). While the authors are right that HadISST and other reanalysis products have deficiencies in high latitudes

due to the lack of observations, it is quite astonishing how the model bias can exceed 4degC where the sea water should be at or near the freezing point. Large biases can be expected at the sea-ice edge, which position may quite differ among coupled models.

5. Page 9, section 4.4 on ocean currents: The section on ocean currents is confusing as the differences in ocean currents are not related to the underlying ocean currents. When discussing differences in ocean currents please term the ocean currents that are enhanced/weakened, for example "enhanced/weakened Kuroshio transport is present in Model A compared to observations."

6. Page 11, lines 11-15: In the first sentence you claim that the differences in NPMOC in the two MPI-ESM model simulations are not caused by the increased coupling frequency while in the second sentence you argue that "suggesting obvious improvements after decreasing coupling interval" are present. Please clarify. Additionally please provide more information on the MPI-ESM model data (e.g. MPI-ESM model version and a reference paper) cited as "The piControl experiment result (available in http://esgf-node.llnl.gov/)". To our knowledge, the publicly available MPI-ESM output available at http://esgf-node.llnl.gov is based on the CMIP5 version of MPI-ESM which implements older versions of both ECHAM6 and MPIOM. It means that the two models differ by far more than just the coupling frequency.

7. page 12, line 15: "The analysis on oceanic and atmospheric circulation has made it clear that the SST bias is consistent with meridional overturning circulation in North Pacific, driven by surface wind stress anomalies that are maintained by anomalous Walker circulation over the tropical Pacific. Cumulus convection process is found to be a major contributor to inter-model differences". The authors should explain in more detail why they assume that cumulus convection is the key in the chain of arguments provided.

8. page 13, line 11: This sentence is confusing. Your statement that enhanced

northerly winds (which we cannot find on Figure 10b as you indicate in the text) in ECHAM6-NEMO3.6 compared to ECHAM5-NEMO3.6 in a region dominated by easterly trade winds are responsible for stronger evaporative cooling of SSTs in ECHAM6-NEMO3.6 compared to ECHAM5-NEMO3.6 is unclear. Please clarify.

9. page 13, line 18: "Since the latent heat and surface wind differences are caused by replacing the AGCM,..." The statement is challenging, as for example the latent heat flux between ocean and atmosphere is a coupled process (see also 12. below).

10. Page 13, line 27: "It turns out that deviations in shortwave flux and latent heat are more significant than those in longwave and sensible heat fluxes." Additional information on the physics behind this statement would be helpful. In its current form it completely ignores the fact that there are large differences in the regional importance of the different fluxes.

11. Page 14 first lines: "...can be confirmed that the AGCM replacement first alters cumulus convection that modulates temperature, specific humidity and atmospheric circulation, which in turn accommodates cloud radiation feedback to a consistent change and affects the radiation budget". Some references that underpin the postulated process chain should be added.

12. Page 16, line 8: "Through analysis on circulation and radiation terms, it has been clear that latent heat of evaporation plays a predominant role in the SST differences after changing the OGCM." This statement does not take into account that LH flux is a coupled process. LH heat flux may impact SSTs in the tropics where atmospheric temperature is high and hence strong evaporation is possible, but is mainly dominated by stability of the atmospheric stratification, windspeed, moisture and temperature in the lowest atmospheric level. It's not as simple as the sentence suggests.

13. Page 16, line 10: What are "conduction processes" and how do they affect sensible heat flux? Additionally, this sentence indicates that the authors don't take into account that e.g. sensible heat flux is a coupled ocean-atmosphere process that mainly depends on the ocean atmosphere temperature difference. The importance of sensible heat flux for SST depends on the region. The authors should also provide a reference if and to what extend the surface flux parameterizations have changed in ECHAM6 with respect to ECHAM5.

14. page 16: line 11: "Attributing simulation deviations to latent heat in the OGCM case is consistent with Cao et al. (2015), which points out that amplitude and meridional variability of latent heat flux over Pacific are the most diverse in CMIP5 models." From our understanding, Cao et al. summarize that LH flux is very diverse amongst coupled models due to the large differences in simulated SST but not that the LH flux differences between the models can explain the bias (From the abstract of Cao et al., 2015: "Regression analysis indicates that the inter-model diversity [in LH flux] may come from the diversity of simulated SST and near-surface atmospheric specific humidity").

15. Page 16, line 14: Please explain what the term "reverse transformation of model bias" means.

16. Page 16, line 25: More details on the "coupling processes" (line 26) that you claim to be responsible for the differences in the wind field in the different GCMs should be added.

17. Page 16, line 28: Please explain what you mean by "net surface radiation is warmer in the east and colder in the west."

Minor comments:

1. Fig. 8: Unit of color contours missing.

2. Fig. 8: Why does the MOC plot stop at 1600m depth?

3. Page 8, Line 6: Details for the reference Huang et al. (2014) are missing.

4. Page 5, Line 29: It is not clear whether the control experiments for the

ECHAM5/NEMO3.6st and ECHAM6/NEMO3.6st setup use present day or piControl external forcing. Please clarify.

5. Page 11, line 5: The Wang et al., 2014 paper cited focuses on the Atlantic MOC and not on the Pacific MOC, hence the citation in this context is not appropriate.

6. Page 11, line 15: The Ge et al., 2017 reference is not appropriate in the context, as Ge et al., 2017 focus on the impact vertical resolution on the SST bias in a ocean model (MOM5) driven by reanalysis data and not the impact of coupling frequency.

7. Page 11, line 31 and Figure 9: Please provide the coordinates you use to determine the SEC region. Figure 9 does not show zonal averages (longitudes on the x axis) as indicated in the text. Please correct.

8. Page 12, line 28: LH flux and evaporation describe the same physical process, so there is no need to discuss the two separately.

References:

Delworth, TL, and co-authors, 2006: GFDL's CM2 Global Coupled Climate Models. Part I: Formulation and Simulation Characteristics. J. Climate, 19, 643–674, https://doi.org/10.1175/JCLI3629.1

Delworth, T.L., A. Rosati, W. Anderson, A.J. Adcroft, V. Balaji, R. Benson, K. Dixon, S.M. Griffies, H. Lee, R.C. Pacanowski, G.A. Vecchi, A.T. Wittenberg, F. Zeng, and R. Zhang, 2012: Simulated Climate and Climate Change in the GFDL CM2.5 High-Resolution Coupled Climate Model. J. Climate, 25, 2755–2781, https://doi.org/10.1175/JCLI-D-11-00316.1

Harlass, J., Latif, M. and Park, W. (2015) Improving Climate Model Simulation of Tropical Atlantic Sea Surface Temperature: The Importance of Enhanced Vertical Atmosphere Model Resolution. Open Access Geophysical Research Letters, 42 (7). pp. 2401-2408. DOI 10.1002/2015GL063310.

Müller, W. A., Jungclaus, J. H., Mauritsen, T., Baehr, J., Bittner, M., Budich, R., et al., 2018: A higher‐resolution version of the Max Planck Institute Earth System Model (MPI‐ESM1.2‐HR). Journal of Advances in Modeling Earth Systems, 10, 1383–1413. https://doi.org/10.1029/2017MS001217

Park, W. et al.: Tropical Pacific Climate and Its Response to Global Warming in the Kiel Climate Model. Journal of Climate, http://doi.org/10.1175/2008JCLI2261.1

Wahl, S., Latif, M., Park, W., & Keenlyside, N. (2009). On the Tropical Atlantic SST warm bias in the Kiel Climate Model. Climate Dynamics, 36(5–6), 891–906. http://doi.org/10.1007/s00382-009-0690-9

Zhang, L., & Zhao, C. (2015). Processes and mechanisms for the model SST biases in the North Atlantic and North Pacific: A link with the Atlantic meridional overturning circulation. Journal of Advances in Modeling Earth Systems, 7(2), 739–758. http://doi.org/10.1002/2014MS000415
* * *

---

## Short Comment (SC6) · 4 Sep 2018

In this comment, I do not directly answer each question or prove the validity of the research findings. In fact, I have integrated the CGCMs for 1000-year realizations that exhibit no sign of climate drift. There are some other figures that have been omitted in the manuscript due to time limitation of writing this paper. Under the supervision of my mentor, I do the research carefully and discuss the findings with teachers and classmates in our research team. Comments like "No information is provided" or "lack of concrete evidence" are likely due to my writing habits that only focus on the innovative points. Currently, I am busy with my graduation paper so that the reponse to these comments have to be postponed. I will probably have enough time two weeks later to answer each question and revise the manuscript.

—————————————————

---

## Author Comment (AC4) · 24 Sep 2018

Response to anonymous referee #1

1. General questions/comments:

*1.1 The authors should probably comment on how ocean (atmosphere) models might initially be tuned by running OMIP (AMIP) runs with fixed forcing fields (SST/sea-ice) to minimize biases etc in uncoupled mode. An existing coupled model could have been tuned to give a good coupled performance, so upgrading a component might require a retuning of the remaining component to get the best possible coupled performance. It could be good to discuss this briefly in the paper.*

Reply:

Thank you for these comments. We would like to tune each component model for a better performance of the coupling system. However, due to time limitation of the research project and high computational cost of coupled model integration, we are currently following the standard control run settings of each component model based on the guidance of model manuals and the configuration of ICM coupled model developed by Huang et al. (2014). Since the CGCM experiments show good results in reproducing the climatology, the SST interannual variability and the ENSO air-sea feedback (relevant information has been supplemented in the context), and model errors are within the tolerance, we believe that the model configuration is sufficiently good for further analysis. Relevant information can be found in section 2.3.1 and 2.3.2.

**'2.3.1 ECHAM5-NEMO3.6**

The schematic structure of ECHAM5-NEMO3.6 is shown in Fig. 1a. … At present, the component models are integrated with the default parameter settings suggested in the user manual, model retuning will be scheduled in further research.

**2.3.2 ECHAM6-NEMO3.6**

Following the coupling framework of MPI-ESM (the update version of ECHAM5/MPI-OM, version number MPI-ESM-1.2.00p4), ECHAM6-NEMO3.6 has been developed with the same atmospheric component model ECHAM-6.3, coupled with NEMO 3.6 stable version through OASIS3-MCT (Craig et al., 2017) (Fig. 1b). Model retuning of the ECHAM-6.3 and NEMO3.6 is left for further studies due to time limitation and

high computational cost. ….'

*1.2 The authors briefly mention the sea-ice model in MPIOM but this is not mentioning of the sea-ice model in NEMO. Since there are large differences in biases between the different model combinations in high latitude on Figure 3 then it might be appropriate to at least mention those differences.*

Reply:

Thank you for your suggestions. We have added the relevant information in section 2.1.1.

'**2.1.1 NEMO**

NEMO model is a well renowned modelling system with high skills in global oceanic circulation simulation, which has been widely used for scientific research, weather forecast (Storkey et al., 2014; Megann et al., 2014), and reanalysis data assimilation (Mogensen et al., 2012a, b). Designed to serve as a flexible tool for ocean and sea ice studies, NEMO manifests good usability interacting with other ACGMs (Gualdi et al., 2003; Luo et al., 2005; Park et al., 2009; Dunlap et al., 2014; Huang et al., 2014). The NEMO stable version 3.6 has been employed in this study, whose ocean component calculates primitive equations on the ORCA2 grid, a tripolar grid of 182 (longitude)×149 (latitude) curvilinear orthogonal mesh in horizontal direction and 31 vertical levels on partial step z coordinate in current research. Turbulent kinetic energy (TKE) closure scheme has been chosen for vertical mixing with enhanced vertical diffusion for convective processes. The Louvain-la-Neuve sea-ice model (LIM3), originally developed by Fichefet and Morales-Maqueda (1997), has been incorporated in NEMO3.6 to represent the sub-grid-scale dynamics and their impact on sea ice thickness and ice-ocean salt exchanges. Main differences between LIM3 and other ice models are related to the physical parameterization of open boundary conditions and sea-ice interactions, with the C-grid formulation of elastic-viscous-plastic rheology (Bouillon et al., 2013).'

*1.3 While I am aware that it is common in the climate modelling community to talk*

*about reanalysis data as observations then it is not true. Both atmospheric and oceanographic reanalyses have errors due both lack of "real" observations until recent times and deficiencies in the data assimilation techniques used.*

Reply:

Thank you for your comments. We have revised the expressions of "observation" with "reanalysis data" or something else equivalent in the article. The following modifications have been made as suggested in section 3, section 4, section 7 and figure captions.

'Evaluation of the mean sea level pressure, zonal and meridional winds at 10m height, cloud cover and surface temperature use the ERA-Interim monthly reanalysis data (Simmons et al., 2006).'

'To characterize the changes in ocean circulation associated with the SST bias, the SODA reanalysis data (Carton & Giese, 2008) has been used following the massive researches on ocean variability and mechanisms (Dewitte et al., 2009; Tett et al., 2014; Drenkard & Karnauskas, 2014).'

'The time period of reanalysis climatology spans from 1981 to 2010, ….'

'The differences between model and reanalysis data are defined as the anomalous fields with simulation minus reanalysis counterpart.'

'For the rest of variables other than ocean surface currents, the standard deviations lie between 0.953 and 1.336, which represent an unremarkable deviation in spatial variability of model simulation compared to reanalysis counterparts.'

'The differences between model simulations and reanalysis counterparts are usually less than 2ºC except for polar areas.'

'For precipitation simulation, ECHAM6-NEMO3.6 presents the highest pattern correlation with reanalysis counterpart and substantially ameliorates double ITCZ problem over SPCZ in boreal summer.'

'Taylor diagram that exhibits a statistical comparison between the simulations and reanalysis data of nine selected variables in summer (a) and winter (b).'

'Figure 10: Summer climatology of 10m wind for (a) reanalysis data, (b) model differences between ECHAM6-NEMO3.6 and ECHAM5-NEMO3.6, (c) model

differences between ECHAM6-NEMO3.6 and MPI-ESM.'

2.  Detailed comments:

*2.1 Page 3 line 24: The "Mogensen et al 2012b" reference is not relevant for data assimilation.*

Reply:

Thank you for your comments. The citation and bibliography regarding *"Mogensen et al 2012b"* have been deleted from the context. We are sorry for the mistake.

*2.2 Page 3 line 26: ORCA2 is just one of many NEMO configurations, so it should be mentioned as a choice made by the authors and not all something all NEMO based system uses.*

Reply:

Thank you for picking up the inaccurate expression. We have modified the sentence as follows:

'The NEMO stable version 3.6 has been employed in this study, whose ocean component is configured to calculate primitive equations on the ORCA2 grid, a tripolar grid of 182 (longitude)×149 (latitude) curvilinear orthogonal mesh ….'

*2.3 Page 3 line 27: The 31 levels of ORCA2 is unevenly distributed, so this should be mentioned for consistency with the MPIOM remark on page 4.*

Reply:

Thank you for pointing it out to us. Modifications in the context have been made as suggested.

'… a tripolar grid of 182 (longitude)×149 (latitude) curvilinear orthogonal mesh in horizontal direction and 31 vertical levels unevenly distributed on partial step z coordinate in current research.'

*2.4 Page 5 line 1 to 3 (and Figure): The authors should be more precise on all fields*

*exchanged between ECHAM5/6 and NEMO.*

Reply:

Thank you for the comments. We have explicitly listed all the coupling fields in section 2.3.1 and 2.3.2, and have modified Figure 1 accordingly.

**'2.3.1 ECHAM5-NEMO3.6**

… Based on the interchange coupling structure of the ECHAM5/MPI-OM, seventeen variables are passed from ECHAM5 to NEMO-3.6st through OASIS3 coupler, including solar radiation, non-solar heat flux and its derivative with respect to temperature, zonal and meridional wind stress, evaporation minus precipitation, and sublimation. The meridional and zonal wind stress vectors are passed to the ORCA2 U and V grid of NEMO-3.6st, while other variables are passed to the T grid. Both ocean and ice regions are considered in the coupling processes. In the opposite direction, six variables comprised of the SST, sea ice temperature, sea ice fraction, sea ice albedo and surface ocean currents are transferred from the ocean model to the atmosphere model. ….

**2.3.2 ECHAM6-NEMO3.6**

… Namelist settings of the ECHAM-6.3 and the coupler OASIS3-MCT are brought into correspondence with those in ECHAM5-NEMO3.6 to the utmost, for example, with the same horizontal resolution T63 on gaussian grid and the same parameterization settings for greenhouse gases. The oceanic component model NEMO3.6 still uses the same configuration as that in ECHAM5-NEMO3.6. Coupling variables of the ECHAM6-NEMO3.6 are the same as those in ECHAM5-NEMO3.6. ….

[Figure]

**Figure 1: Schematic structure of ECHAM5-NEMO3.6 (a), and ECHAM6-NEMO3.6 (b).'**

*2.5 Page 5 line 5: The 1200 seconds time step for NEMO quite short compared to the reference time step 5760 second time for ORCA2. Any reason for choosing this ocean time step?*

Reply:

Thank you for your comments. We chose the short time step length following Huang et al. (2014), who made the ICM coupled model based on ECHAM-5.4 and NEMO2.3. Through personal contact with their development team, we were told that they set uniform time step length (1200s) for both atmospheric and oceanic component models.

They explained that the atmosphere model couldn't sustain large time step length as that in the ocean model. After increasing the coupling frequency to every 4 hours, the boundary conditions of the ocean model change rapidly so that the oceanic variables fluctuate more drastically. It is better to synchronize the ocean model with the same time step length to mitigate the violent oscillation, which may improve the numerical stability in model integration.

*2.6 Page 5 line 22: The grid specifications for MPIOM was already mentioned in section 2.1.1 on page 4 line 6, so this should not be repeated.*

Reply:

We're sorry for the mistake. Redundant content has been deleted from section 2.3.3.

'Due to high computational cost, the low-resolution version MPI-ESM-LR is used in this study (version number MPI-ESM-1.2.00p4), with ECHAM6 running at T63L47 resolution (horizontal resolution about 1.875° and 47 vertical levels).'

*2.7 Page 6 line 18+24: Reanalyses are not observations as mentioned above.*

Reply:

Thanks for your comment. We have rewritten the sentences as listed in the above response to major comment 1.3.

*2.8 Page 6 line 29: ORCA2 only has refinement in latitude from 2.0 to 0.5 degree, but not in longitude. The text seems to imply that ORCA2 has 0.5 degree resolution in both latitude and longitude which is not true.*

Reply:

Thank you for these comments. We have corrected the expression in section 3.

'Because the local mesh refinements of ORCA2 grid in NEMO3.6 make meridional resolution finer in tropics (about 0.5°) than that of higher latitudes (about 2°), the SST model data has been remapped onto 1° × 1° as a trade-off.'

*2.9 Page 8 line 11: ECHAM5-NEMO3.5 should be ECHAM5-NEMO3.6.*

Reply:

Thank you for pointing out the typo. We have corrected the expression.

'However, the ECHAM5-NEMO3.6 exhibits warm SST biases instead of cold biases in North Pacific, and no remarkable SST bias is found North Atlantic.'

*2.10 Page 10 line 9-11: Would the fact that NEMO3.6st and MPIOM uses different grids contribute to the differences in SST biases?*

Reply:

Thank you for asking the question. We fully agree that grid differences also affect the SST simulation. As suggested in previous studies, the model resolution has a non-negligible impact on the simulation results, let alone different projection grids. Comparison among different grid systems is an interesting topic to probe into. In this paper, we would like to focus on the underlying physical mechanisms that account for model biases after changing component models. In this regard, an additional explanation has been added in section 4.5.

'The SST biases are also attributed to different projection grids of NEMO3.6 and MPIOM, which however is beyond the scope of this paper.'

*2.11 Section 5.1+Figure 8: I assume that the MOC is compared with the SODA reanalysis? Maybe this should be I am aware of several inter-comparisons of reanalysis of the AMOC which show quite different results compared to the Rapid array, so I would expect quite some uncertainty in the SODAY reanalysis. Maybe this could be discussed briefly?*

Reply:

Thank you for these comments. Yes, the North Pacific MOC is compared with the SODA reanalysis in Figure 8. Choosing different reanalysis datasets can influence the MOC bias pattern, but the uncertainties of climatology above bathypelagic zone are less than those of deeper levels. Previous studies also suggest that the interannual variability of oceanic variables tend to suffer from large discrepancies among the ocean reanalyses with weak signal-to-noise ratio. Nevertheless, we haven't gone so far to

explore the interannual variations in this paper. Since the SODA reanalysis data has been employed in many ocean studies, including the North Pacific MOC that is mostly within the thermocline layer, it is reliable to use the SODA reanalysis data to analyze model biases related to the MOC. The following illustration has been added to section 5.1.

'Model bias pattern can be influenced by the choice of ocean reanalysis dataset. Uncertainties in the ocean reanalyses (ORAs) include the estimation of sea ice thickness, interannual variability of salinity, surface heat flux and mixed layer depth (Balmaseda et al., 2015; Toyoda et al., 2017). There are substantive discrepancies in temperature, salinity and density in the deep ocean and the Southern Ocean, where observations are sparse especially before the year 2000. Ocean heat content at deep levels varies largely among the ORAs, with majority of spread originating in the Southern Hemisphere due to lack of observation for data assimilation (Palmer et al., 2017). Discrepancies among the ORAs are evident in the strength and structures of the AMOC, with distinctive differences in the depth of equatorward return flow and the depth of the maximum AMOC in northern high latitudes (Karspeck et al., 2017). However, the uncertainties of climatology above bathypelagic zone are less than those of deeper levels. Since the SODA reanalysis data has been employed in many ocean studies, including the North Pacific MOC that is mostly within the thermocline layer, it is reliable to use the SODA reanalysis data to analyze model biases related to the MOC.'

*2.12 Section 5.2+Figure 9: Again, mention that this is against ERAI reanalysis and not observations. Also it might worth being very specific on which area the authors define as the SEC. Maybe a map on Figure 9 with a box could be useful.*

Reply:

Thank you for these comments. We have modified the 1$^{st}$ sentence in section 5.2 to explicitly illustrate that the model bias is against the ERA-Interim reanalysis. The south equatorial current (SEC) is defined as the ocean currents between 15$^o$ S~4$^o$ N in tropical Pacific. A new subplot with a box that encompasses the region has been attached to Figure 9. The content modifications are as follows:

'On behalf of the atmosphere motion that accounts for cold tongue bias and strong easterly bias in surface currents, vertical circulation bias zonally averaged over South Equatorial Current (SEC) against the ERA-Interim reanalysis is given in Fig. 9.'
'

**Figure 9: Model biases of the vertical structure of atmospheric circulation (vector) and the temperature (contour) over SEC area in summer for (a) ECHAM6-NEMO3.6, (b) ECHAM5-NEMO3.6, (c) MPI-ESM. The SEC region is marked as the blue box in (d).'**

*2.13 Section 6: Unless I missed something then this section focus exclusively on JJA with no mentioning of DJF. There are some differences in SST biases on Figure 3 for all models, so maybe it would be worth mentioning if the authors would expect the conclusions to also be true for the boreal winter?*

Reply:

Thank you for your suggestion. We have mentioned it in section 7 as follows:

'Although the analysis in section 5 and 6 only focuses on the physical mechanisms

behind simulation differences in summer, the atmospheric and oceanic circulations have a similar impact on radiation budget and surface heat transport in other seasons. It implies that the effects by replacing each component model may be the same, and the conclusions drawn above are also valid for SST biases in winter.'

*2.14 Page 14 line 12: "Cloud" should be "cloud".*

Reply:

We are sorry for the careless mistake. The capitalization has been dropped.

'The effect of cloud radiation feedback is not prominent, ….'

*2.15 Figure 1: Please add all fields exchanged between the atmosphere and ocean model.*

Reply: Thank you for the suggestion. Every coupling field is explicitly annotated in the figure.

'

[Figure]

**Figure 1: Schematic structure of ECHAM5-NEMO3.6 (a), and ECHAM6-NEMO3.6 (b).**'

*2.16 Figure 2: I am not sure that a Taylor diagram is the best way to present the data. Some of the data points are very small and difficult to see. Maybe the authors could think of alternatives for presenting the information?*

Reply:

Thank you for these comments. There are nine variables in each Taylor diagram that makes some points close to each other and hard to discern. Alternatives to show the statistical information may include bar plots and tables, which nevertheless lack some spatial relationships from the visual display. Since the Taylor diagram is widely used in model assessment, and not all points shown in the figure need to be distinguished to comprehend the inference drawn in section 3, we are afraid that the Taylor diagram is

the best choice. The size of Figure 2 has been enlarged to show a clearer picture with subplots arranged in the vertical direction.

'

[Figure]

[Figure]

**Figure 2: Taylor diagram that exhibits a statistical comparison between the simulations and reanalysis data of nine selected variables in summer (a) and winter (b). Each number represents one variable: (1) precipitation, (2) mean sea level pressure, (3) zonal winds at 10m height, (4) meridional winds at 10m height, (5) 2m temperature, (6) total radiation flux (net shortwave plus net longwave), (7) SST, (8) sea surface zonal currents, (9) sea surface meridional currents. Upward-pointing triangles, squares and diamonds, respectively, represent the ECHAM6-NEMO3.6, ECHAM5-NEMO3.6, and the MPI-ESM results.'**

*2.17 All maps on Figure 3,4,5,6,7,9,10,13: I suggest moving the longitudes from 0 to 360 degrees to 20 to 380 degrees since it will mostly avoid cutting the Atlantic Ocean in two.*

Reply:

Thank you for your suggestion. We have redrawn the figures with center longitude located at 200° E. Figure 9 only concerns the vertical circulation over the south equatorial current of tropical Pacific, and thus it is left unchanged. The modified figures are attached below:

[Figure]

[Figure]

**Figure 3: Biases of the SST simulation in summer (left column) and winter (right column) corresponding to each CGCM: (a, b) ECHAM6-NEMO3.6, (c, d) ECHAM5-NEMO3.6, (e, f) MPI-ESM.**

[Figure]

**Figure 4: The same as Fig. 3 but for simulated precipitation climatology.**

[Figure]

**Figure 5: The same as Fig. 3 but for simulated surface wind stress (unit: N/m².**

[Figure]

**Figure 6: The same as Fig. 3 but for simulated surface currents (unit: m/s).**

[Figure]

**Figure 7: The same as Fig. 3 but for simulated total radiation.**

[Figure]

**Figure 10: Summer climatology of 10m wind for (a) reanalysis data, (b) model differences between ECHAM6-NEMO3.6 and ECHAM5-NEMO3.6, (c) model differences between ECHAM6-NEMO3.6 and MPI-ESM. SST simulation differences for (d) between ECHAM6-NEMO3.6 and ECHAM5-NEMO3.6, and (e) between ECHAM6-NEMO3.6 and MPI-ESM.**

[Figure]

**Figure 12: Simulation differences in radiation budget between ECHAM6-NEMO3.6 and ECHAM5-NEMO3.6.**

[Figure]

**Figure 13: Simulation differences in radiation budget between ECHAM6-NEMO3.6 and MPI-ESM.**

,

*2.18 Figure 5+6+10: I suggest moving the legends with the arrow outside the plots.*

Reply:

Thanks for your suggestion. We have moved the arrow annotation outside the plots in these vector plots. Figure 5, 6 and 10 have been attached in the above response to question 2.17.

*2.19 Figure 8: Adding a colour legend might be useful since the minus sign on the contours are a bit hard to see.*

Reply:

Thanks for your suggestion. We have added a color legend in Figure 8.

'

[revised manuscript text omitted]
 NEMO stable version 3.6 has been employed in this study, whose ocean component is configured to calculate primitive equations on the ORCA2 grid, a tripolar grid of 182 (longitude)×149 (latitude) curvilinear orthogonal mesh in horizontal direction and 31 vertical levels unevenly distributed on partial step z coordinate in current research. Turbulent kinetic energy (TKE) closure scheme has been chosen for vertical mixing with enhanced vertical diffusion for convective processes. The Louvain-la-Neuve sea-ice model (LIM3), originally developed by Fichefet and Morales-Maqueda (1997), has been incorporated in NEMO3.6 to represent the sub-grid-scale dynamics and their impact on sea ice thickness and ice-ocean salt exchanges. Main differences between LIM3 and other ice models are related to the physical parameterization of open boundary conditions and sea-ice interactions, with the C-grid formulation of elastic-viscous-plastic rheology (Bouillon et al., 2013).

**2.1.2 MPIOM**

MPIOM model is formulated on Arakawa-C grid for horizontal dimension and z-grid for vertical dimension, using the hydrostatic and Boussinesq approximations in the model dynamic equations (Jungclaus et al., 2006; Jungclaus et al., 2013). Vertical mixing and diffusion are parameterized following Pacanowski and Philander (1981), with a diffusion coefficient varies with grid spacing (Redi, 1982). Sea-ice model is included in MPIOM where sea-ice thickness is modulated by turbulent atmospheric fluxes and oceanic heat transport (Wolff et al., 1997; Marsland et al., 2003; Notz et al., 2013). The model is configured with 40 unevenly spaced vertical levels on the GR1.5 grid, a conformal mapping grid of 256 (longitude) × 220 (latitude) in the horizontal making horizontal resolution approximately 1.5°.

**2.2 AGCM**

The ECHAM atmospheric model developed by the Max Planck Institute for Meteorology has been used in many studies since its first version (ECHAM1) branched from the cycle 17 operational model at Medium Range Weather Forecasts (ECMWF) (Roeckner et al., 1989; Simmons et al., 1989). Incorporation of new features along the course of model development gradually makes ECHAM capable of reproducing meticulous characteristics in the weather system, including those in cumulus convection, moisture transport, radiation and land-surface processes (Roeckner et al., 1996, 2003; Raddatz et al., 2007; Brovkin et al., 2009). ECHAM has also been employed in the coupled earth modelling system, from coupling with large‐scale geostrophic ocean model (LSG) (Maier‐Reimer et al., 1993) to the latest version of Earth system model MPI-ESM (Baehr et al., 2015). The ECHAM model consists of a dry spectral dynamic core, a set of parameterization schemes dealing with solar irradiance, moist convection, land-surface properties, etc. The versions in use for this paper are ECHAM5.4 (Roeckner et al., 2003) and ECHAM6.3 (Stevens et al., 2013). Major updates from ECHAM5 to ECHAM6 include improved representation of shortwave spectrum, a new aerosol parameterization scheme, middle atmosphere and surface albedo descriptions are also enhanced. Despite the new implementations in ECHAM6, the AGCMs are set to the same configuration if applicable, to minimize the differences caused by updates of physical parameterization schemes.

**2.3 CGCM**

**2.3.1 ECHAM5-NEMO3.6**

The schematic structure of ECHAM5-NEMO3.6 is shown in Fig. 1a. Overall, the ECHAM5-NEMO3.6 consists of the atmospheric model ECHAM5.4, oceanic and sea ice model NEMO3.6st (the stable version of NEMO3.6), and the coupler Ocean Atmosphere Sea Ice Soil (OASIS3) (Valcke, 2013). Although ECHAM5 (Roeckner et al., 2003, 2006) is an older version of the atmospheric model developed by Max Plank Institute of Meteorology compared with ECHAM6 (Stevens et al., 2013), it was employed by the previous coupled atmosphere ocean model ECHAM5/MPI-OM (Jungclaus et al., 2006), which has been well tested and reassured of accurate surface flux transfer between the oceanic and atmospheric component models. Based on the interchange coupling structure of the ECHAM5/MPI-OM, seventeen variables are passed from ECHAM5 to NEMO-3.6st through OASIS3 coupler, including solar radiation, non-solar heat flux and its derivative with respect to temperature, zonal and meridional wind stress, evaporation minus precipitation, and sublimation. The meridional and zonal wind stress vectors are passed to the ORCA2 U and V grid of NEMO-3.6st, while other variables are passed to the T grid. Both ocean and ice regions are considered in the coupling processes. In the opposite direction, six variables comprised of the SST, sea ice temperature, sea ice fraction, sea ice albedo and surface ocean currents are transferred from the ocean model to the atmosphere model. Bilinear interpolation method is used in the exchanges of physical variables between ECHAM5 gaussian grid and NEMO-3.6st ORCA2 grid. The time steps for ocean and atmosphere models are both set to 1200 seconds, and the coupling frequency is 4 hours once (every 12-time steps). At present, the component models are integrated with the default parameter settings suggested in the user manual, model retuning will be scheduled in further research.

**2.3.2 ECHAM6-NEMO3.6**

Following the coupling framework of MPI-ESM (the update version of ECHAM5/MPI-OM, version number MPI-ESM-1.2.00p4), ECHAM6-NEMO3.6 has been developed with the same atmospheric component model ECHAM-6.3, coupled with NEMO 3.6 stable version through OASIS3-MCT (Craig et al., 2017) (Fig. 1b). Model retuning of the ECHAM-6.3 and NEMO3.6 is left for further studies due to time limitation and high computational cost. Namelist settings of the ECHAM-6.3 and the coupler OASIS3-MCT are brought into correspondence with those in ECHAM5-NEMO3.6 to the utmost,

for example, with the same horizontal resolution T63 on gaussian grid and the same parameterization settings for greenhouse gases. The oceanic component model NEMO3.6 still uses the same configuration as that in ECHAM5-NEMO3.6. Coupling variables of the ECHAM6-NEMO3.6 are the same as those in ECHAM5-NEMO3.6. Details in experiment configuration are elaborated in section 2.4.

**2.3.3 MPI-ESM**

[revised manuscript text omitted]

**5. Circulation Patterns Relevant to SST Biases**

**5.1 Meridional Overturning Circulation**

The importance of Meridional Overturning Circulation (MOC) to SST in coupled models has been proved in previous studies (Wang et al., 2014, Liu et al., 2016). Thus, the comparison of North Pacific MOC (NPMOC) (Fig. 8) between the CGCMs with the same atmospheric or oceanic component model can help to explain bias characteristics in relation to SST deviations. Model bias pattern can be influenced by the choice of ocean reanalysis dataset. Uncertainties in the ocean reanalyses (ORAs) include the estimation of sea ice thickness, interannual variability of salinity, surface heat flux and mixed layer depth (Balmaseda et al., 2015; Toyoda et al., 2017). There are substantive discrepancies in temperature, salinity and density in the deep ocean and the Southern Ocean, where observations are sparse especially before the year 2000. Ocean heat content at deep levels varies largely among the ORAs, with majority of spread originating in the Southern Hemisphere due to lack of observation for data

assimilation (Palmer et al., 2017). Discrepancies among the ORAs are evident in the strength and structures of the AMOC, with distinctive differences in the depth of equatorward return flow and the depth of the maximum AMOC in northern high latitudes (Karspeck et al., 2017). However, the uncertainties of climatology above bathypelagic zone are less than those of deeper levels. Since the SODA reanalysis data has been employed in many ocean studies, including the North Pacific MOC that is mostly within the thermocline layer, it is reliable to use the SODA reanalysis data to analyse model biases related to the MOC.

[revised manuscript text omitted]
. Although the analysis in section 5 and 6 only focuses on the physical mechanisms behind simulation differences in summer, the atmospheric and oceanic circulations have a similar impact on radiation budget and surface heat transport in other seasons. It implies that the effects by replacing each component model may be the same, and the conclusions drawn above are also valid for SST biases in winter. The mechanisms illustrated in this study provides a new vision of model bias origin, which is heuristic for model improvement researches and practices.

**Code and data availability**

The model source code is available from the authors upon request. Restrictions on source code license will be imposed unless for academic and non-commercial use. The experimental data can be found in https://doi.org/10.5281/zenodo.1306338 (Gui et al., 2018).

[revised manuscript text omitted]

Palmer, M. D., Roberts, C. D., Balmaseda, M., Chang, Y.-S., Chepurin, G., Ferry, N., Fujii, Y., Good, S. A., Guinehut, S., Haines, K., Hernandez, F., Köhl, A., Lee, T., Martin, M. J., Masina, S., Masuda, S., Peterson, K. A., Storto, A., Toyoda, T., Valdivieso, M., Vernieres, G., Wang, O., and Xue, Y.,

Ocean heat content variability and change in an ensemble of ocean reanalyses, Climate Dynamics, 49: 909-930, doi:10.1007/s00382-015-2801-0, 2017.

Park, W., Keenlyside, N., Latif, M., Stroh, A., Redler, R., Roeckner, E., and Madec, G.: Tropical Pacific climate and its response to global warming in the Kiel climate model, J. Climate, 22, 71–92, 2009.

[revised manuscript text omitted]

---

## Author Comment (AC5) · 30 Sep 2018

Response to anonymous referee #2

Major Comments:

*1. The authors compared mostly the Pacific Ocean and concentrated on tropical Pacific. One of the stark improvements lie in the North Atlantic, especially where the cold bias in the subpolar region seen in MPI-ESM has been ameliorated. It would be extremely beneficial for the community to discuss this too, and show how the AMOC may be different between these different CGCMs.*

Reply:

Thank you for these comments. We have compared the AMOC simulation results of the three CGCMs. The inter-model differences imply that cold SST in North Atlantic is partially attributed to weak AMOC, which is consistent with previous studies. The experiment results have been introduced in section 5.1. The MOC bias figures have been redrawn with the same color settings that are easier to recognize.

'Since the upper cell of Atlantic meridional overturning circulation (AMOC) plays a significant role in delaying warming signals from anthropogenic greenhouse gases and responding to climate change (Marshall et al., 2014; Buckley and Marshall, 2016), model bias analysis is still focused on the upper ocean levels. The overall magnitude of AMOC bias is less than that of NPMOC with significantly reduced biases near the sea surface (Fig. 10), which is consistent with those of surface currents among the three CGCMs. The ECHAM6-NEMO3.6 shows exiguous bias near the ocean surface, but presents strong biases in the mesopelagic zone of subtropical areas, bringing more heat to higher latitudes (Fig. 10a). Likewise, the ECHAM5-NEMO3.6 exhibits strong circulation biases rotating clockwise in the thermocline that intensifies poleward heat transport (Fig. 10b). With similar bias patterns of the AMOC, the ECHAM5-NEMO3.6 and the ECHAM6-NEMO3.6 have opposite SST biases in North Atlantic (Figs. 3a and 3c), which implies that the air-sea feedback including WES feedback and NAM as suggested by Zhang & Zhao (2015) takes the responsibility. The MPI-ESM experiment shows negative biases in tropical Atlantic from the sea surface to the bathypelagic zone, indicating that the overturning circulation has been restrained. There is a narrow positive bias in the subtropical Atlantic, but its strength has been limited by the negative biases nearby. One consequence of the weak AMOC is the decrease of SST in North Atlantic due to less heat supply from the tropics (Fig. 3e). The overturning circulation is enhanced in the middle latitudes with one centre located north of 35ºN and another centre around 55ºN at the depth of 1200m. It still promotes the poleward heat transport and results in warm SST biases in subpolar region (Fig. 3e). The AMOC biases in the MPI-ESM piControl experiment are similar as those in the MPI-ESM experiment, with more negative biases in tropical Atlantic. Comparing the AMOC biases between the MPI-ESM and the ECHAM5-NEMO3.6, it can be seen that the SST cold biases in North Atlantic are partially attributed to decreased MOC in the thermocline of tropical and extra-tropical oceans. However, the air-sea interaction also takes account of the SST variations in consideration of the SST

differences between the ECHAM5-NEMO3.6 and the ECHAM6-NEMO3.6. Zhang & Zhao (2015) suggested that the cold SST bias in Atlantic caused the same cold bias in North Pacific through different mechanisms originating in tropical and extra-tropical Atlantic. Because the differences of NPMOC are bigger than those of AMOC between these two newly developed CGCMs, it suggests an inverse cause-and-effect relationship between the cold SST biases in North Pacific and North Atlantic where the former takes the lead.

[Figure]

**Figure 10: Model biases of AMOC in summer, (a) ECHAM6-NEMO3.6, (b) ECHAM5-NEMO3.6, (c) MPI-ESM, (d) MPI-ESM piControl, Unit: Sv.᾿**

*2. Changes in mean state from changing component model provide simple but necessary information. However, it would benefit readers and the community if more information such as variability of the system is assessed, such as variance of variables that the authors have covered, and maybe even power spectrum of ENSO and SAM.*
Reply:

    Thanks for your comments. We have added assessment of the system variability, including power spectrum analysis of Niño3.4 index, SOI and SAM index. Because the model data in use spans 100-year realizations, much longer than that of the ERA-Interim reanalysis, the ERA-20c has been used instead to evaluate the model variability. The SST reanalysis data from Hadley center (HadISST) is still used for SST variability assessment.

Both HadISST and ERA-20c reanalyses have 100-year time span from 1911 to 2010. Model variability evaluation has been provided in section 4.7.

**'4.7 Model variability of ENSO and SAM**

In the coupled ocean-atmosphere system, global climate variability has been driven by the El Niño-Southern Oscillation (ENSO), the southern annular mode (SAM, also called the Antarctic Oscillation) and the Indian Ocean dipole (IOD) (Philander, 1990; Wallace and Thompson, 2002; Saji et al., 1999). It is therefore necessary to examine the model variability by applying spectra analysis on relevant indices. The CGCM simulations of the three indices are generally consistent with the theoretical red noise (Markov) spectrum (figure omitted). The Niño3.4 index is defined as the SST anomalies averaged over the NINO34 region (5°N-5°S,170°W -120°W). It shows high variance in 2-7 years' period that documents the ENSO peaks in the HadISST reanalysis (Fig. 8a). All of the CGCMs reproduce similar variations of the Niño3.4 power spectra. The ECHAM6-NEMO3.6 presents weak variabilities at the interannual and interdecadal scale, whose periodic peaks are about one year less than the reanalysis counterpart. The ECHAM5-NEMO3.6 shows a better spectral distribution that best coincides with the reanalysis at the interannual scale. However, it still suffers a weak variability at the interdecadal scale and the periodic peak is even half a year less than that of the ECHAM6-NEMO3.6. The MPI-ESM instead takes on an intensified interannual variability, which stays strong at the interdecadal scale. The Southern Oscillation Index (SOI) is calculated based on the differences in sea level pressure anomalies between Tahiti and Darwin in Australia. In comparison to the Niño3.4 spectra, the SOI exhibits similar peaks at the interannual scale in the ERA-20c reanalysis (Fig. 8b). Nevertheless, all the CGCMs reproduce weak variabilities at the interannual and interdecadal scales. The ECHAM6-NEMO3.6 presents the best simulation with a significant increase in variance around 4 years' period, while the ECHAM5-NEMO3.6 shows the weakest variability at the interannual scale. It implies that the AGCM replacement has an opposite effect on the Niño3.4 and SOI variabilities. The MPI-ESM also show reduced variance from the annual scale and above, quite the opposite to that in the Niño3.4 case. Since the model biases of ENSO variability may be attributed to thermocline feedback and zonal wind variations (Borlace et al., 2013), the reversed changes in the variabilities of Niño3.4 and SOI can be caused by the related oceanic and atmospheric processes. The SAM index is calculated following Gong and Wang (1999) by the differences of normalized monthly zonal mean sea level pressure at 40°S and 65°S. Variations of SAM tend to be more flattened than those of SOI in the ERA-20c reanalysis (Fig. 8c), with prominent fluctuations from biannual to interannual scales. Compared with the reanalysis counterpart, all CGCMs show more power at interannual time scales that represents a robust modulation of the SAM, which is possibly attributed to the semi-annual oscillation (SAO) (Hurrell and van Loon, 1994) and circulation anomalies over Antarctica (Thompson and Solomon, 2002). The ECHAM6-NEMO3.6 presents stronger decadal variability than that of the reanalysis data, while the ECHAM5-NEMO3.6 exhibits weaker low-frequency variability. Since the high variance in low-frequency band represents the upward trend of SAM index at decadal scale (Raphael and

Holland, 2006), updating the AGCM can result in a drastic change of long-term climate variability in southern hemisphere. In contrast, the MPI-ESM shows the SAM variability very close to the reanalysis counterpart from 1 year and above, indicating that the OGCM feedback to the atmosphere can lead to a better representation of the inter-decadal variability.

[Figure]

**Figure 8: Power spectra of (a) Niño3.4 index, (b) SOI, (c) SAM index. Solid line denotes the calculation results of reanalysis data, green dotted line denotes the ECHAM5-NEMO3.6 simulation, red dotted line denotes the ECHAM6-NEMO3.6 simulation, and blue dotted line denotes the MPI-ESM simulation.'**

*3. Section 6.2: Please be aware that correlation does not indicate intensity or magnitude of relationship. Fig. 13 shows that shortwave radiation has a much large magnitude and can thus possibly have a greater effect on SST even though it has a smaller correlation value compared to sensible heat and longwave radiation. This does not change the proposed mechanism, but SW radiation should be incorporated into the explanation.*

Reply:

Thank you for these comments. We have rewritten this section to explain the model differences with a full picture of the air-sea interactions. Relevant content in section 7 has also been revised. Major modifications are pasted below.

'… Since the top 3 variables that are most relevant to SST deviations (Tab. 1) suggest changes in the surface heat budget, involving the momentum and temperature exchanges between ocean and atmosphere, it implies that the joint effects of atmosphere and ocean models lead to the model deviation patterns (Fig. 15). Simulation of ocean dynamics is different after replacing the OGCM. With changes in ocean advection, the SST and surface currents are altered which modulates the surface evaporation, convection and heat conduction. Subsequently the latent heat and sensible heat fluxes vary over the sea surface. The thermal and moisture perturbations from the ocean are passed to the atmosphere during coupling processes (Fig. 15c, d). Variations of low-level atmospheric circulation and humidity take effects on cloud formation and cloud liquid water path that changes precipitation and cloud radiative forcing. The net shortwave and longwave radiations are influenced and make a difference to the atmospheric circulation (Fig. 12c) and the heat budget over the sea surface (Fig. 15 a, b). Then, the perturbation signal is transferred back to the ocean that changes the SST and surface currents. This air-sea feedback finally reaches a quasi-equilibrium with marked SST warming over vast maritime spaces over the globe. The associated physical processes represented by each radiation term in Figure 15 are in accordance with their signs of pattern correlation (Tab. 1). It seems to suggest that surface evaporation plays a predominant role in the SST differences, because the latent heat flux holds the biggest correlation coefficient and its deviation pattern is the most obvious among the four radiation terms. However, it is more likely a manifestation of the large-scale ocean dynamical effect on the inter-model differences as suggest by Ying and Huang (2016).

… So far, the analysis has made it clear that changing the OGCM affects the SST simulation through large-scale air-sea feedback that mainly involves surface evaporation, heat conduction, atmospheric convection and cloud masking of incoming and outgoing radiative fluxes. Under the impact of net radiation surplus over western Pacific, the temperature rises with upward motion which forms a warm advection heading east and descending over eastern Pacific…..'

*4. Along the lines of pattern correlation being used to rank the importance of contributing factors. Since the region of focus is the Pacific, the pattern correlation should does be performed over the Pacific rather than globally. This should not change the concluding results, but it may provide some differences in the correlation and add robustness to your estimates.*

Reply:

Thank you for these comments. We have added the correlation analysis over the Pacific to facilitate the discussion of related physical processes in section 6. As you suggested before, the assessment of SAM index variability and AMOC have been added to the paper. The focus of the paper has thus been expanded to other oceans, and we still keep the previous correlation analysis performed at the global scale in the manuscript. Major modifications are as follows:

'… Because the analysis in previous sections is mainly focused on the Pacific, the pattern correlation over the area is also provided for comparison (Table 2).

… When the study area is narrowed down to the Pacific, the top 3 ranking variables are the same, with the net shortwave radiation takes the 1st place (Tab. 2). In both cases, the net shortwave and net longwave fluxes are ranked in top 3, which suggests that radiation budget is closely associated with SST inter-model differences. The pattern correlations of the 3 variables are higher within the Pacific than that at the global scale, indicating an increased robustness of the variable estimates.

… For simulation differences after changing the OGCM, latent heat flux holds the biggest share in pattern correlation with corresponding SST differences (Tab. 1), much bigger than that of net longwave flux at the second place and sensible heat flux at the third place. When the study area is confined within the Pacific, the three leading variables are the same but the sensible heat flux is ranked top (Tab. 2). Compared with the rankings in the AGCM case, it can be seen that the OGCM influences the simulation with different physical mechanisms than those by replacing AGCM. The effect of cloud radiation feedback is not prominent, while the physical processes associated with latent heat and sensible heat, including heat conduction, evaporation and convection, take a bigger share in the SST inter-model differences. SST differences can also lead to changes in longwave flux.'

*5. Please state significance test for pattern correlation, particularly how the effective degree of freedom is computed or decorrelation length scale used. Having a significance at 99.9% level for a correlation at 0.018 is hard to reckon with.*

Reply:

Thank you for pointing out the problem. We applied the Student-t test on the correlation coefficients, and the degree of freedom is taken as n-2 (n is the number of grid points over the globe). Because the model data has been interpolated onto the 1x1 grid, which makes the sample

points too large even without land points. The threshold value for 99.9% is 0.0174. We have added information of significance test in the table caption as follows:

'Asterisk (*) denotes the correlation coefficients passing the Student-t test above 99.9% confidence level. A larger number of grid points are involved that makes the threshold value relatively small. Numbers in boldface denotes the maximum absolute correlation value for the AGCM or OGCM replacement.'

*6. Page 6, line 13: Which year of external forcing (like CO2, greenhouse gases, aerosol, etc.) was used for the piControl run to obtain equilibrated state? Is this the same (similar) forcing as seen during observation period? If the piControl uses forcing that is quite different from observations (e.g. excess of 100ppm of CO2), then one would have to account for these differences when computing biases with respect to observations.*

Reply:

Thank you for these comments. The MPI-ESM experiment is configured based on the piControl run but not restricted to every parameter value. The external forcing is consistent with the present time climatology. The $CO_2$ value is set to default 353.9 ppm in the user manual. Other greenhouse gases like $NO_2$ also follows the default present time setting so that they are consistent with each other. The aerosol settings use the climatology compiled by S. Kinne without any complementation of volcanic aerosols. Model initialization is started from the climatology basic state recalculated with the AMIP run input data from 1981 to 2010. The piControl run data used in this study has the same time span as that of the reanalysis data from the year 1981 to 2010, which should be able to represent the model abilities in reproducing the climatology of the same period. Relevant information has been added to section 2.4.

'**2.4 Experimental Setup**

The control experiments conducted in this study are aimed to reproduce atmospheric and oceanic circulation characteristics of present time, and then to compare with each other in order to examine model performance improvements. The coupled control simulation is thus configured with reference to MPI-ESM piControl experiment settings. Model initialization is started from the climatology basic state recalculated with the AMIP run input data from 1981 to 2010. The $CO_2$ value is set to default 353.9 ppm in the user manual. Other greenhouse gases like $NO_2$ also follows the default present time setting so that they are consistent with each

other. The aerosol settings use the climatology compiled by S. Kinne without any complementation of volcanic aerosols....'

*7. page 12, line 7-8: Rather than inferring the relationship, the authors can easily compute the pattern correlation and quantify the correlation.*

Reply:

Thanks for your suggestion. We have calculated the pattern correlation between temperature biases and easterly anomalies in the three CGCM experiments. Because the easterly biases are most remarkable below 800hPa, the levels taken into consideration are between 800hPa and 1000hPa. The pattern correlation coefficients between temperature biases and zonal wind biases are 0.315 for the ECHAM6-NEMO3.6 experiment, 0.586 for the ECHAM5-NEMO3.6 experiment, and 0.411 for the MPI-ESM experiment. All of them have passed the 99% Student-t significance test. The ECHAM5-NEMO3.6 has the most significant easterly biases and holds the largest correlation coefficient, much bigger than that of the other two CGCMs, which indicates a strong correlation between the warmer temperature and stronger easterly anomalies. The analysis result has been added to section 5.2.

'It can be seen from Figure 11 that cold temperature biases in the MPI-ESM transform into warm temperature biases in the ECHAM5-NEMO3.6 accompanied by increasing easterly anomalies in lower atmosphere, which suggests warmer temperature bias is correlated with stronger easterly anomalies. The pattern correlation between temperature biases and easterly anomalies in the three CGCM experiments has been calculated respectively. Because the easterly biases are most remarkable below 800hPa, the levels taken into consideration are between 800hPa and 1000hPa. The pattern correlation coefficients between temperature biases and zonal wind biases are 0.315 for the ECHAM6-NEMO3.6 experiment, 0.586 for the ECHAM5-NEMO3.6 experiment, and 0.411 for the MPI-ESM experiment. All of them have passed the 99% Student-t significance test. The ECHAM5-NEMO3.6 has the most significant easterly biases and holds the largest correlation coefficient, much bigger than that of the other two CGCMs, which confirms the correlation between the warmer temperature and stronger easterly anomalies.'

Minor comments
*1. page 6, line 11: please provide the actual websites*
Reply:
We have added the URLs of the websites for MPIOM and NEMO.

'…namelist settings of MPIOM and NEMO3.6 still follow their own default settings for control run provided by their respective official websites (http://www.mpimet.mpg.de/en/science/models/mpi-esm/mpiom and https://www.nemo-ocean.eu).'

2. *page 8, line 31-32: Been staring at Fig. 4 but am not able to see "positive precipitation bias over the southern tropical Atlantic and negative bias on tropical South America". Maybe you meant "negative bias over southern tropical Atlantic and positive bias on tropical South America"? Please indicate which specific figures you are referring to.*

Reply:

Thank you for pointing out the problem. We try to describe the variational trends of inter-model differences of precipitation that are consistent with previous research finding. It refers to the model biases in Figures 4a, 4c, 4e. We have corrected some imprecise expressions in the context. The focus areas have been outlined in the figure below.

'…As suggested by Stevens et al. (2013), the extent of precipitation bias over the tropical Atlantic and South America seems related to each other, because the ECHAM5-NEMO3.6 shows less biases than those of the other two CGCMs with less blue contours over the tropical Atlantic and lighter red contours over Colombia and Venezuela (Fig. 4a, c, e) ….

[Figure]

3. *page 8, line 32: "… seems related to each other, because the values and scales of biases in … "Please explain, it is not obvious how this is the case.*

Reply:

As response to the above question, the simulated precipitation exhibits inter-model differences within the boxes. The ECHAM5-NEMO3.6 shows less biases than those of the other two CGCMs, with less blue contours over the tropical Atlantic and lighter red contours over Colombia and Venezuela. The simultaneous reduction of precipitation biases coincides with the research findings of Stevens et al. (2013), who point out that the precipitation biases over the two areas are possibly related. We have further modified the paper content to clarify the issue.

'…As suggested by Stevens et al. (2013), the extent of precipitation bias over the tropical Atlantic and South America seems related to each other, because the ECHAM5-NEMO3.6 shows less biases than those of the other two CGCMs with less blue contours over the tropical Atlantic and lighter red contours over Colombia and Venezuela (Fig. 4a, c, e) ….'

4.  *Fig. 5: Please provide magnitude difference in surface wind stress in colour. This would make it easier for readers to see extent of anomalies.*

Reply:

We have shaded the vectors in Figure 5 as you suggest.

'

[Figure]

**Figure 5: The same as Fig. 3 but for simulated surface wind stress (unit: N/m².).**'

5.  *page 9, line 15: Change "sea water to across" to "sea water across"*

Reply:

Thank you for pointing out the grammar mistake. We have revised the sentence.

'…which can drive the underlying sea water across the date line in the other two CGCMs.'

6.  *page 10, line 13: Change "it turns out that three CGCMs" to "it turns out that all three CGCMs"*

Reply:

Thank you for carefully reading. We have revised the sentence as you suggest.

'…and it turns out that all three CGCMs suffer from a similar bias pattern in boreal summer ….'

*7. page 11, line 11: Change "," to "."*

Reply:

Thank you for pointing out the problem. We have corrected the mistake.

'Bias in the MPI-ESM experiment manifests itself to a larger degree (Fig. 8c).'

*8. page 11, line 31: Change "In consistence" to "Consistent"*

Reply:

Thanks for your advice. We have revised the sentence as you suggest.

'Consistent with surface wind stress bias (Fig. 5), ….'

*9. page 13, line 3: Change "it can be assumed" to "it would suggest"*

Reply:

Thanks for your suggestion. We have revised the sentence as you suggest.

'Noting that latent heat flux has a correlation value close to that of net longwave radiation, it would suggest that cumulus convection changes sea surface winds and then surface evaporation, ….'

*10. page 15, line 22: None of the CGCMs used on this paper resolved mesoscale processes, so the subsentence "depending on mesoscale processes in the atmospheric and oceanic systems" cannot be concluded from this study and should thus be omitted.*

Reply:

Thank you for your advice. We have deleted the improper content from the sentence.

'…model errors of each CGCM are caused by different physical mechanisms.'

*11. Please state units for Fig. 5, 6, 8, 9, 10, 11.*

Reply:

We have added the units in figure captions.

'

[Figure]

Figure 5: The same as Fig. 3 but for simulated surface wind stress (unit: N/m²).

[Figure]

**Figure 6: The same as Fig. 3 but for simulated surface currents (unit: m/s).**

[Figure]

**Figure 8: Model biases of summer climatology of meridional overturning circulation simulation in North Pacific, (a) ECHAM6-NEMO3.6, (b) ECHAM5-NEMO3.6, (c) MPI-ESM, (d) MPI-ESM piControl, Units: Sv.**

[Figure]

**Figure 9: Model biases of the vertical structure of atmospheric circulation (vector) and the temperature (contour) over SEC area in summer for (a) ECHAM6-NEMO3.6, (b) ECHAM5-NEMO3.6, (c) MPI-ESM. The SEC region is marked as the blue box in (d). Vector units: m/s, contour units: °C.**

[Figure]

**Figure 10: Summer climatology of 10m wind for (a) reanalysis data, (b) model differences between ECHAM6-NEMO3.6 and ECHAM5-NEMO3.6, (c) model differences between ECHAM6-NEMO3.6 and MPI-ESM. SST simulation differences for (d) between ECHAM6-NEMO3.6 and ECHAM5-NEMO3.6, and (e) between ECHAM6-NEMO3.6 and MPI-ESM. Vector units: m/s, contour units: °C.**

[Figure]

**Figure 11: Simulation differences in the vertical structure of atmospheric circulation and the temperature over the SEC area: (a) between ECHAM6-NEMO3.6 and ECHAM5-NEMO3.6, (b) between ECHAM6-NEMO3.6 and MPI-ESM. Vector units: m/s, contour unit: °C.**

,

The marked-up manuscript version is pasted below with yellow highlights on the changes:

[revised manuscript text omitted]
 NEMO stable version 3.6 has been employed in this study, whose ocean component is configured to calculate primitive equations on the ORCA2 grid, a tripolar grid of 182 (longitude)×149 (latitude) curvilinear orthogonal mesh in horizontal direction and 31 vertical levels unevenly distributed on partial step z coordinate in current research. Turbulent kinetic energy (TKE) closure scheme has been chosen for vertical mixing with enhanced vertical diffusion for convective processes. The Louvain-la-Neuve sea-ice model (LIM3), originally developed by

Fichefet and Morales-Maqueda (1997), has been incorporated in NEMO3.6 to represent the sub-grid-scale dynamics and their impact on sea ice thickness and ice-ocean salt exchanges. Main differences between LIM3 and other ice models are related to the physical parameterization of open boundary conditions and sea-ice interactions, with the C-grid formulation of elastic-viscous-plastic rheology (Bouillon et al., 2013).

**2.1.2 MPIOM**

MPIOM model is formulated on Arakawa-C grid for horizontal dimension and z-grid for vertical dimension, using the hydrostatic and Boussinesq approximations in the model dynamic equations (Jungclaus et al., 2006; Jungclaus et al., 2013). Vertical mixing and diffusion are parameterized following Pacanowski and Philander (1981), with a diffusion coefficient varies with grid spacing (Redi, 1982). Sea-ice model is included in MPIOM where sea-ice thickness is modulated by turbulent atmospheric fluxes and oceanic heat transport (Wolff et al., 1997; Marsland et al., 2003; Notz et al., 2013). The model is configured with 40 unevenly spaced vertical levels on the GR1.5 grid, a conformal mapping grid of 256 (longitude) × 220 (latitude) in the horizontal making horizontal resolution approximately 1.5°.

**2.2 AGCM**

The ECHAM atmospheric model developed by the Max Planck Institute for Meteorology has been used in many studies since its first version (ECHAM1) branched from the cycle 17 operational model at Medium Range Weather Forecasts (ECMWF) (Roeckner et al., 1989; Simmons et al., 1989). Incorporation of new features along the course of model development gradually makes ECHAM capable of reproducing meticulous characteristics in the weather system, including those in cumulus convection, moisture transport, radiation and land-surface processes (Roeckner et al., 1996, 2003; Raddatz et al., 2007; Brovkin et al., 2009). ECHAM has also been employed in the coupled earth modelling system, from coupling with large‑scale geostrophic ocean model (LSG) (Maier‑Reimer et al., 1993) to the latest version of Earth system model MPI-ESM (Baehr et al., 2015). The ECHAM model consists of a dry spectral dynamic core, a set of parameterization schemes dealing with solar irradiance, moist convection, land-surface properties, etc. The versions in use for this paper are ECHAM5.4 (Roeckner et al., 2003) and ECHAM6.3 (Stevens et al., 2013). Major updates from ECHAM5 to ECHAM6 include improved representation of shortwave spectrum, a new aerosol parameterization scheme, middle atmosphere and surface albedo descriptions are also enhanced. Despite the new implementations in ECHAM6, the AGCMs are set to the same configuration if applicable, to minimize the differences caused by updates of physical parameterization schemes.

**2.3 CGCM**

**2.3.1 ECHAM5-NEMO3.6**

The schematic structure of ECHAM5-NEMO3.6 is shown in Fig. 1a. Overall, the ECHAM5-NEMO3.6 consists of the atmospheric model ECHAM5.4, oceanic and sea ice model NEMO3.6st (the stable version of NEMO3.6), and the coupler Ocean Atmosphere Sea Ice Soil (OASIS3) (Valcke, 2013). Although ECHAM5

(Roeckner et al., 2003, 2006) is an older version of the atmospheric model developed by Max Plank Institute of Meteorology compared with ECHAM6 (Stevens et al., 2013), it was employed by the previous coupled atmosphere ocean model ECHAM5/MPI-OM (Jungclaus et al., 2006), which has been well tested and reassured of accurate surface flux transfer between the oceanic and atmospheric component models. Based on the interchange coupling structure of the ECHAM5/MPI-OM, seventeen variables are passed from ECHAM5 to NEMO-3.6st through OASIS3 coupler, including solar radiation, non-solar heat flux and its derivative with respect to temperature, zonal and meridional wind stress, evaporation minus precipitation, and sublimation. The meridional and zonal wind stress vectors are passed to the ORCA2 U and V grid of NEMO-3.6st, while other variables are passed to the T grid. Both ocean and ice regions are considered in the coupling processes. In the opposite direction, six variables comprised of the SST, sea ice temperature, sea ice fraction, sea ice albedo and surface ocean currents are transferred from the ocean model to the atmosphere model. Bilinear interpolation method is used in the exchanges of physical variables between ECHAM5 gaussian grid and NEMO-3.6st ORCA2 grid. The time steps for ocean and atmosphere models are both set to 1200 seconds, and the coupling frequency is 4 hours once (every 12-time steps). At present, the component models are integrated with the default parameter settings suggested in the user manual, model retuning will be scheduled in further research.

**2.3.2 ECHAM6-NEMO3.6**

Following the coupling framework of MPI-ESM (the update version of ECHAM5/MPI-OM, version number MPI-ESM-1.2.00p4), ECHAM6-NEMO3.6 has been developed with the same atmospheric component model ECHAM-6.3, coupled with NEMO 3.6 stable version through OASIS3-MCT (Craig et al., 2017) (Fig. 1b). Model retuning of the ECHAM-6.3 and NEMO3.6 is left for further studies due to time limitation and high computational cost. Namelist settings of the ECHAM-6.3 and the coupler OASIS3-MCT are brought into correspondence with those in ECHAM5-NEMO3.6 to the utmost, for example, with the same horizontal resolution T63 on gaussian grid and the same parameterization settings for greenhouse gases. The oceanic component model NEMO3.6 still uses the same configuration as that in ECHAM5-NEMO3.6. Coupling variables of the ECHAM6-NEMO3.6 are the same as those in ECHAM5-NEMO3.6. Details in experiment configuration are elaborated in section 2.4.

**2.3.3 MPI-ESM**

[revised manuscript text omitted]

**4.7 Model variability of ENSO and SAM**

In the coupled ocean-atmosphere system, global climate variability has been driven by the El Niño-Southern Oscillation (ENSO), the southern annular mode (SAM, also called the Antarctic Oscillation) and the Indian Ocean dipole (IOD) (Philander, 1990; Wallace and Thompson, 2002; Saji et al., 1999). It is therefore necessary to examine the model variability by applying spectra analysis on relevant indices. The CGCM simulations of the three indices are generally consistent with the theoretical red noise (Markov) spectrum (figure omitted). The Niño3.4 index is defined as the SST anomalies averaged over the NINO34 region (5°N-5°S,170°W -120°W). It shows high variance in 2-7 years' period that documents the ENSO peaks in the HadISST reanalysis (Fig. 8a). All of the CGCMs reproduce similar variations of the Niño3.4

power spectra. The ECHAM6-NEMO3.6 presents weak variabilities at the interannual and interdecadal scale, whose periodic peaks are about one year less than the reanalysis counterpart. The ECHAM5-NEMO3.6 shows a better spectral distribution that best coincides with the reanalysis at the interannual scale. However, it still suffers a weak variability at the interdecadal scale and the periodic peak is even half a year less than that of the ECHAM6-NEMO3.6. The MPI-ESM instead takes on an intensified interannual variability, which stays strong at the interdecadal scale. The Southern Oscillation Index (SOI) is calculated based on the differences in sea level pressure anomalies between Tahiti and Darwin in Australia. In comparison to the Niño3.4 spectra, the SOI exhibits similar peaks at the interannual scale in the ERA-20c reanalysis (Fig. 8b). Nevertheless, all the CGCMs reproduce weak variabilities at the interannual and interdecadal scales. The ECHAM6-NEMO3.6 presents the best simulation with a significant increase in variance around 4 years' period, while the ECHAM5-NEMO3.6 shows the weakest variability at the interannual scale. It implies that the AGCM replacement has an opposite effect on the Niño3.4 and SOI variabilities. The MPI-ESM also show reduced variance from the annual scale and above, quite the opposite to that in the Niño3.4 case. Since the model biases of ENSO variability may be attributed to thermocline feedback and zonal wind variations (Borlace et al., 2013), the reversed changes in the variabilities of Niño3.4 and SOI can be caused by the related oceanic and atmospheric processes. The SAM index is calculated following Gong and Wang (1999) by the differences of normalized monthly zonal mean sea level pressure at 40°S and 65°S. Variations of SAM tend to be more flattened than those of SOI in the ERA-20c reanalysis (Fig. 8c), with prominent fluctuations from biannual to interannual scales. Compared with the reanalysis counterpart, all CGCMs show more power at interannual time scales that represents a robust modulation of the SAM, which is possibly attributed to the semi-annual oscillation (SAO) (Hurrell and van Loon, 1994) and circulation anomalies over Antarctica (Thompson and Solomon, 2002). The ECHAM6-NEMO3.6 presents stronger decadal variability than that of the reanalysis data, while the ECHAM5-NEMO3.6 exhibits weaker low-frequency variability. Since the high variance in low-frequency band represents the upward trend of SAM index at decadal scale (Raphael and Holland, 2006), updating the AGCM can result in a drastic change of long-term climate variability in southern hemisphere. In contrast, the MPI-ESM shows the SAM variability very close to the reanalysis counterpart from 1 year and above, indicating that the OGCM feedback to the atmosphere can lead to a better representation of the inter-decadal variability.

**5. Circulation Patterns Relevant to SST Biases**

The importance of Meridional Overturning Circulation (MOC) to SST in coupled models has been proved in previous studies (Wang et al., 2014, Liu et al., 2016). Thus, the comparison of North Pacific MOC (NPMOC) (Fig. 9) between the CGCMs with the same atmospheric or oceanic component model can help to explain bias characteristics in relation to SST deviations. Model bias pattern can be influenced by the choice of ocean reanalysis dataset. Uncertainties in the ocean reanalyses (ORAs) include the estimation of sea ice thickness, interannual variability of salinity, surface heat flux and mixed layer depth (Balmaseda et al., 2015; Toyoda et al., 2017). There are substantive discrepancies in temperature, salinity and density in the deep

ocean and the Southern Ocean, where observations are sparse especially before the year 2000. Ocean heat content at deep levels varies largely among the ORAs, with majority of spread originating in the Southern Hemisphere due to lack of observation for data assimilation (Palmer et al., 2017). Discrepancies among the ORAs are evident in the strength and structures of the AMOC, with distinctive differences in the depth of equatorward return flow and the depth of the maximum AMOC in northern high latitudes (Karspeck et al., 2017). However, the uncertainties of climatology above bathypelagic zone are less than those of deeper levels. Since the SODA reanalysis data has been employed in many ocean studies, including the North Pacific MOC that is mostly within the thermocline layer, it is reliable to use the SODA reanalysis data to analyse model biases related to the MOC.

**5.1 Meridional Overturning Circulation**

The ECHAM5-NEMO3.6 possesses two prominent improvements of the SST simulation, one in the Pacific cold tongue region, and the other in North Atlantic. It also exhibits warm SST biases in the North Pacific that is the opposite to most CMIP CGCMs. These peculiar characteristics require further investigation on the meridional overturning circulation (MOC) in North Pacific and North Atlantic. For the MOC in North Pacific, the ECHAM5-NEMO3.6 and ECHAM6-NEMO3.6 possess similar bias patterns overall, with intensified tropical cell and deep tropical cell that are mentioned in Liu et al. (2011) (Fig. 9a, b). Tropical cell enhancement is more significant in the ECHAM5-NEMO3.6, so that the upwelling in the tropics and subsequent heat transport to mid-latitudes are more than those in the ECHAM6-NEMO3.6. Bias in the MPI-ESM experiment manifests itself to a larger degree (Fig. 9c). The piControl experiment result (available in http://esgf-node.llnl.gov/) is attached (Fig. 9d) to demonstrate that the prominent biases in MPI-ESM are not caused by increasing coupling frequency from one day to 4 hours. The piControl run data used in this study has the same time span as that of the reanalysis data from the year 1981 to 2010, which should be able to represent the model abilities in reproducing the climatology of the same period. More contour lines appear in NPMOC bias distribution in piControl than the MPI-ESM setting in this paper (Fig. 9c, d), suggesting obvious improvements after decreasing coupling interval. This is consistent with previous studies (Bernie et al., 2008; Ge et al., 2017). Enhancement of tropical cell in MPI-ESM is unremarkable, compared with that of ECHAM5-NEMO3.6, but deep tropical cell is significantly intensified that tropical upwelling is forced to become too strong. This may explain why tropical SST bias of MPI-ESM is more than 2℃ in boreal summer, without significant radiation errors and easterly anomalies in surface stress. Excessive upwelling in tropical Pacific, induced by intensified deep tropical cell of NPMOC, cools down the local SST. Unlike the ECHAM5-NEMO3.6 case, where the SST cooling in tropical Pacific is driven by intensified tropical cell of NPMOC that transports more heat to mid-latitudes, the MPI-ESM simulation of tropical and subtropical cells does not differ much from SODA reanalysis data. The resulting poleward heat transport carried by NPMOC is not increased in the MPI-ESM experiment. Therefore, the SST bias in North Pacific remains negative (Fig. 3e) under the impact of Northern Hemisphere annular mode (NAM) and wind-evaporation-SST (WES) feedback, when cold SST biases appear in tropical and extratropical North Atlantic (Zhang & Zhao, 2015).

The cold SST bias in North Pacific and North Atlantic for ECHAM6-NEMO3.6 can also be explained with the same reason. But for the ECHAM5-NEMO3.6 case, the poleward heat transport has been enhanced by intensified tropical cell to bring up the SST in mid-latitudes (Fig. 3c).

Since the upper cell of Atlantic meridional overturning circulation (AMOC) plays a significant role in delaying warming signals from anthropogenic greenhouse gases and responding to climate change (Marshall et al., 2014; Buckley and Marshall, 2016), model bias analysis is still focused on the upper ocean levels. The overall magnitude of AMOC bias is less than that of NPMOC with significantly reduced biases near the sea surface (Fig. 10), which is consistent with those of surface currents among the three CGCMs. The ECHAM6-NEMO3.6 shows exiguous bias near the ocean surface, but presents strong biases in the mesopelagic zone of subtropical areas, bringing more heat to higher latitudes (Fig. 10a). Likewise, the ECHAM5-NEMO3.6 exhibits strong circulation biases rotating clockwise in the thermocline that intensifies poleward heat transport (Fig. 10b). With similar bias patterns of the AMOC, the ECHAM5-NEMO3.6 and the ECHAM6-NEMO3.6 have opposite SST biases in North Atlantic (Figs. 3a and 3c), which implies that the air-sea feedback including WES feedback and NAM as suggested by Zhang & Zhao (2015) takes the responsibility. The MPI-ESM experiment shows negative biases in tropical Atlantic from the sea surface to the bathypelagic zone, indicating that the overturning circulation has been restrained. There is a narrow positive bias in the subtropical Atlantic, but its strength has been limited by the negative biases nearby. One consequence of the weak AMOC is the decrease of SST in North Atlantic due to less heat supply from the tropics (Fig. 3e). The overturning circulation is enhanced in the middle latitudes with one centre located north of $35^{o}$ N and another centre around $55^{o}$ N at the depth of 1200m. It still promotes the poleward heat transport and results in warm SST biases in subpolar region (Fig. 3e). The AMOC biases in the MPI-ESM piControl experiment are similar as those in the MPI-ESM experiment, with more negative biases in tropical Atlantic. Comparing the AMOC biases between the MPI-ESM and the ECHAM5-NEMO3.6, it can be seen that the SST cold biases in North Atlantic are partially attributed to decreased MOC in the thermocline of tropical and extra-tropical oceans. However, the air-sea interaction also takes account of the SST variations in consideration of the SST differences between the ECHAM5-NEMO3.6 and the ECHAM6-NEMO3.6. Zhang & Zhao (2015) suggested that the cold SST bias in Atlantic caused the same cold bias in North Pacific through different mechanisms originating in tropical and extra-tropical Atlantic. Because the differences of NPMOC are bigger than those of AMOC between these two newly developed CGCMs, it suggests an inverse cause-and-effect relationship between the cold SST biases in North Pacific and North Atlantic where the former takes the lead.

**5.2 Vertical Structure of Atmospheric Circulation**

On behalf of the atmosphere motion that accounts for cold tongue bias and strong easterly bias in surface currents, vertical circulation bias zonally averaged over South Equatorial Current (SEC) against the ERA-Interim reanalysis is given in Fig. 9. Consistent with surface wind stress bias (Fig. 5), there are easterly anomalies in the lower atmosphere for ECHAM5-NEMO3.6 and ECHAM6-NEMO3.6 (Fig. 11a, b). With

anomalous upward (downward) motion over the western (eastern) Pacific, the easterly anomalies are maintained by the anomalous Walker circulation across tropical Pacific. The easterly anomalies in lower atmosphere decrease significantly in MPI-ESM experiment (Fig. 11c), without strong downward motion in the eastern Pacific. Inter-model differences in the easterly biases at low levels are consistent with the amplitude of NPMOC tropical cell, because the stronger surface winds collaborating with Coriolis effect push more sea water to move west and poleward.

It can be seen from Figure 11 that cold temperature biases in the MPI-ESM transform into warm temperature biases in the ECHAM5-NEMO3.6 accompanied by increasing easterly anomalies in lower atmosphere, which suggests warmer temperature bias is correlated with stronger easterly anomalies. The pattern correlation between temperature biases and easterly anomalies in the three CGCM experiments has been calculated respectively. Because the easterly biases are most remarkable below 800hPa, the levels taken into consideration are between 800hPa and 1000hPa. The pattern correlation coefficients between temperature biases and zonal wind biases are 0.315 for the ECHAM6-NEMO3.6 experiment, 0.586 for the ECHAM5-NEMO3.6 experiment, and 0.411 for the MPI-ESM experiment. All of them have passed the 99% Student-t significance test. The ECHAM5-NEMO3.6 has the most significant easterly biases and holds the largest correlation coefficient, much bigger than that of the other two CGCMs, which confirms the correlation between the warmer temperature and stronger easterly anomalies. Recalling precipitation biases over the SEC area (Fig. 4a, c, e), where ECHAM5-NEMO3.6 shows the driest situation and MPI-ESM presents the wettest circumstance, the latent heat absorption in the course of evaporation turns out to be responsible for the temperature bias. More precipitation requires higher specific humidity that favours cloud formation, in which more latent heat of vaporization is taken up when water vapor mixing ration is increased. Since cumulus convection modulates changes in temperature, specific humidity and atmospheric circulation, it is most likely to be the predominant factor that shapes the inter-model differences.

[revised manuscript text omitted]

**6.2 Impacts of OGCM Replacement**

For simulation differences after changing the OGCM, latent heat flux holds the biggest share in pattern correlation with corresponding SST differences (Tab. 1), much bigger than that of net longwave flux at the second place and sensible heat flux at the third place. When the study area is confined within the Pacific, the three leading variables are the same but the sensible heat flux is ranked top (Tab. 2). Compared with the rankings in the AGCM case, it can be seen that the OGCM influences the simulation with different physical mechanisms than those by replacing AGCM. The effect of cloud radiation feedback is not prominent, while the physical processes associated with latent heat and sensible heat, including heat conduction, evaporation and convection, take a bigger share in the SST inter-model differences. SST differences can also lead to changes in longwave flux. The negative sign of correlations for latent heat, sensible heat and longwave flux (Tab. 1 and 2) suggests a reverse trend to SST variations.

Contrary to the AGCM case, low-level wind deviations in tropical Pacific (Fig. 12c) are in the same direction as the background climatology (Fig. 12a), which should increase surface evaporation through stronger near surface winds and then results in colder SST through WES feedback. Nevertheless, warmer SST appears in most parts of tropical and subtropical oceans, laying waste to the assumption trying to explain SST deviation in terms of WES mechanism. Since the top 3 variables that are most relevant to SST deviations (Tab. 1) suggest changes in the surface heat budget, involving the momentum and temperature exchanges between ocean and atmosphere, it implies that the joint effects of atmosphere and ocean models lead to the model deviation patterns (Fig. 15). Simulation of ocean dynamics is different after replacing the OGCM. With changes in ocean advection, the SST and surface currents are altered which modulates the surface evaporation, convection and heat conduction. Subsequently the latent heat and sensible heat fluxes vary over the sea surface. The thermal and moisture perturbations from the ocean are passed to the atmosphere during coupling processes (Fig. 15c, d). Variations of low-level atmospheric circulation and humidity take effects

on cloud formation and cloud liquid water path that changes precipitation and cloud radiative forcing. The net shortwave and longwave radiations are influenced and make a difference to the atmospheric circulation (Fig. 12c) and the heat budget over the sea surface (Fig. 15 a, b). Then, the perturbation signal is transferred back to the ocean that changes the SST and surface currents. This air-sea feedback finally reaches a quasi-equilibrium with marked SST warming over vast maritime spaces over the globe. The associated physical processes represented by each radiation term in Figure 15 are in accordance with their signs of pattern correlation (Tab. 1). It seems to suggest that surface evaporation plays a predominant role in the SST differences, because the latent heat flux holds the biggest correlation coefficient and its deviation pattern is the most obvious among the four radiation terms. However, it is more likely a manifestation of the large-scale ocean dynamical effect on the inter-model differences as suggest by Ying and Huang (2016).

Atmospheric circulation accommodates itself to changes of radiation budget. An anomalous Walker circulation cell appears over the SEC area (Fig. 13b), with updraft over Philippine sea where positive SST deviations indicate a net radiation surplus. The upward flow rises to the upper-level troposphere and condensation becomes evident due to temperature drops. It then diverts toward east and orients downward over eastern Pacific, which enhances surface westerly anomalies in the lower atmosphere. This circulation pattern is consistent with anomalous easterlies over tropical Pacific (Fig. 12c). Warm temperature deviations (Fig.13b) can be viewed as the manifestation of surface radiation surplus and latent heat release of condensation. So far, the analysis has made it clear that changing the OGCM affects the SST simulation through large-scale air-sea feedback that mainly involves surface evaporation, heat conduction, atmospheric convection and cloud masking of incoming and outgoing radiative fluxes. Under the impact of net radiation surplus over western Pacific, the temperature rises with upward motion which forms a warm advection heading east and descending over eastern Pacific. This anomalous Walker cell drives low-level winds towards the west, leading to westerly anomalies over vast areas of tropical Pacific.

**7. Summary and Discussion**

[revised manuscript text omitted]

For strong easterly bias over eastern Pacific in ECHAM6-NEMO3.6, also common in the most CGCMs, the MPI-ESM instead shows poleward bias, which suggests that OGCM replacement can also diminish this bias through coupling processes. It is easy to see that an anomalous Walker circulation with counter-clockwise rotation will appear over the tropical Pacific, if net surface radiation is warmer in the east and colder in the west. With proper OGCM configuration, sea surface evaporation can initiate this radiation anomaly, and the resulting Walker circulation will decrease the easterly wind speed. Although the analysis in section 5 and 6 only focuses on the physical mechanisms behind simulation differences in summer, the atmospheric and oceanic circulations have a similar impact on radiation budget and surface heat transport in other seasons. It implies that the effects by replacing each component model may be the same, and the conclusions drawn above are also valid for SST biases in winter. The mechanisms illustrated in this study provides a new vision of model bias origin, which is heuristic for model improvement researches and practices.

**Code and data availability**

The model source code is available from the authors upon request. Restrictions on source code license will be imposed unless for academic and non-commercial use. The experimental data can be found in https://doi.org/10.5281/zenodo.1306338 (Gui et al., 2018).

The reanalysis data used in this study can be downloaded from the following websites:

1. The Hadley Centre SST data is downloaded from https://www.metoffice.gov.uk/hadobs/hadisst/.

2. The GPCP precipitation data is downloaded from  https://precip.gsfc.nasa.gov/.

3. The ERA-Interim and ERA-20c monthly reanalysis data is available at http://apps.ecmwf.int/datasets/.

4. The SCOW wind stress data is available at http://cioss.coas.oregonstate.edu/scow.

5. Surface radiation reanalysis data of CERES EBAF-Surface Ed4.0 is downloaded from https://ceres.larc.nasa.gov.

6. The SODA3.3 ocean current reanalysis data is downloaded from: http://www.atmos.umd.edu.

The MPI-ESM piControl experiment data is available at http://esgf-node.llnl.gov.

**Acknowledgements**

This work was supported by the National Key Research and Development Program of China (2016YFA0601600), and the National Natural Science Foundation of China (U1502233 and 41565002). It is also co-supported by Yunnan University's Research Innovation Fund for Graduate Students (YDY17019).

**Table 1: Pattern Correlations of surface variables and SST deviation**

| Model Replacement / Variables | AGCM | OGCM |
|---|---|---|
| Surface zonal currents | 0.006 | -0.002 |
| Surface meridional currents | 0.0158 | -0.018* |
| Sensible heat flux | -0.068* | -0.269* |
| Latent heat flux | -0.262* | **-0.393**\* |
| Mean sea level pressure | -0.136* | -0.009 |
| 10m zonal wind | 0.018* | -0.010 |
| 10m meridional wind | -0.025* | 0.045* |
| Surface albedo | -0.101* | -0.022* |
| Net shortwave flux | 0.192* | 0.138* |
| Net longwave flux | **-0.269**\* | -0.283* |
| Precipitation | 0.100* | 0.154* |

Asterisk (*) denotes the correlation coefficients passing the Student-t test above 99.9% confidence level. A larger number of grid points are involved that makes the threshold value relatively small. Numbers in boldface denotes the maximum absolute correlation value for the AGCM or OGCM replacement.

**Table 2: Pattern Correlations of surface variables and SST deviation over the Pacific**

| Model Replacement  / Variables | AGCM | OGCM |
|---|---|---|
| Surface zonal currents | 0.069* | -0.033* |
| Surface meridional currents | 0.055* | -0.136* |
| Sensible heat flux | -0.286* | **-0.318**-* |
| Latent heat flux | -0.302* | -0.315* |
| Mean sea level pressure | -0.246* | 0.014 |
| 10m zonal wind | 0.070* | 0.039* |
| 10m meridional wind | 0.022 | -0.005 |
| Surface albedo | 0.274* | 0.034* |
| Net shortwave flux | **0.375**-* | 0.102* |
| Net longwave flux | -0.303**-* | -0.229* |
| Evaporation | -0.302* | -0.315* |
| Precipitation | 0.103* | 0.124* |

Asterisk (*) denotes the correlation coefficients passing the Student-t test above 99.9% confidence level. A larger number of grid points are involved that makes the threshold value relatively small. Numbers in boldface denotes the maximum absolute correlation value for the AGCM or OGCM replacement.

[Figure]

**Figure 1: Schematic structure of ECHAM5-NEMO3.6 (a), and ECHAM6-NEMO3.6 (b).**

[Figure]

[Figure]

**Figure 2: Taylor diagram that exhibits a statistical comparison between the simulations and reanalysis data of nine selected variables in summer (a) and winter (b). Each number represents one variable: (1) precipitation, (2) mean sea level pressure, (3) zonal winds at 10m height, (4) meridional winds at 10m height, (5) 2m temperature, (6) total radiation flux (net shortwave plus net longwave), (7) SST, (8) sea surface zonal currents, (9) sea surface meridional currents. Upward-pointing triangles, squares and diamonds, respectively, represent the ECHAM6-NEMO3.6, ECHAM5-NEMO3.6, and the MPI-ESM results.**

[Figure]

**Figure 3: Biases of the SST simulation in summer (left column) and winter (right column) corresponding to each CGCM: (a, b) ECHAM6-NEMO3.6, (c, d) ECHAM5-NEMO3.6, (e, f) MPI-ESM.**

[Figure]

**Figure 4: The same as Fig. 3 but for simulated precipitation climatology.**

[Figure]

**Figure 5: The same as Fig. 3 but for simulated surface wind stress (unit: N/m².).**

[Figure]

**Figure 6: The same as Fig. 3 but for simulated surface currents (unit: m/s).**

[Figure]

**Figure 7: The same as Fig. 3 but for simulated total radiation.**

[Figure]

**Figure 8: Power spectra of (a) Niño3.4 index, (b) SOI, (c) SAM index. Solid line denotes the calculation results of reanalysis data, green dotted line denotes the ECHAM5-NEMO3.6 simulation, red dotted line denotes the ECHAM6-NEMO3.6 simulation, and blue dotted line denotes the MPI-ESM simulation.**

[Figure]

**Figure 9: Model biases of summer climatology of meridional overturning circulation simulation in North Pacific, (a) ECHAM6-NEMO3.6, (b) ECHAM5-NEMO3.6, (c) MPI-ESM, (d) MPI-ESM piControl, Unit: Sv.**

[Figure]

**Figure 10: Model biases of AMOC in summer, (a) ECHAM6-NEMO3.6, (b) ECHAM5-NEMO3.6, (c) MPI-ESM, (d) MPI-ESM piControl, Unit: Sv.**

[Figure]

**Figure 11: Model biases of the vertical structure of atmospheric circulation (vector) and the temperature (contour) over SEC area in summer for (a) ECHAM6-NEMO3.6, (b) ECHAM5-NEMO3.6, (c) MPI-ESM. The SEC region is marked as the blue box in (d). Vector units: m/s, contour unit: °C.**

[Figure]

**Figure 12: Summer climatology of 10m wind for (a) reanalysis data, (b) model differences between ECHAM6-NEMO3.6 and ECHAM5-NEMO3.6, (c) model differences between ECHAM6-NEMO3.6 and MPI-ESM. SST simulation differences for (d) between ECHAM6-NEMO3.6 and ECHAM5-NEMO3.6, and (e) between ECHAM6-NEMO3.6 and MPI-ESM. Vector units: m/s, contour unit: °C.**

[Figure]

**Figure 13: Simulation differences in the vertical structure of atmospheric circulation and the temperature over the SEC area: (a) between ECHAM6-NEMO3.6 and ECHAM5-NEMO3.6, (b) between ECHAM6-NEMO3.6 and MPI-ESM. Vector units: m/s, contour unit: °C.**

[Figure]

**Figure 14: Simulation differences in radiation budget between ECHAM6-NEMO3.6 and ECHAM5-NEMO3.6.**

[Figure]

**Figure 15: Simulation differences in radiation budget between ECHAM6-NEMO3.6 and MPI-ESM.**

---

## Author Comment (AC6) · 30 Sep 2018

Dear professor,

In the previous response, I forgot to revise the abstract and conclusion parts to be consistent with other modifications in the manuscript. I'm really sorry for the mistake. The attachment in this response contains the same authors' response and the correct manuscript. We followed your advice to revise the explanation of the inter-model differences in the OGCM replacement case, and we realized that the surface evaporation

is one part of the changes induced by ocean advection. Previous studies also support this conclusion.

Thank you very much for your constructive comments.

Best regards

Gui Shu

Please also note the supplement to this comment:
https://www.geosci-model-dev-discuss.net/gmd-2018-130/gmd-2018-130-AC6-supplement.pdf

**Supplement:**

Response to anonymous referee #2

Major Comments:

*1. The authors compared mostly the Pacific Ocean and concentrated on tropical Pacific. One of the stark improvements lie in the North Atlantic, especially where the cold bias in the subpolar region seen in MPI-ESM has been ameliorated. It would be extremely beneficial for the community to discuss this too, and show how the AMOC may be different between these different CGCMs.*

Reply:

Thank you for these comments. We have compared the AMOC simulation results of the three CGCMs. The inter-model differences imply that cold SST in North Atlantic is partially attributed to weak AMOC, which is consistent with previous studies. The experiment results have been introduced in section 5.1. The MOC bias figures have been redrawn with the same color settings that are easier to recognize.

'Since the upper cell of Atlantic meridional overturning circulation (AMOC) plays a significant role in delaying warming signals from anthropogenic greenhouse gases and responding to climate change (Marshall et al., 2014; Buckley and Marshall, 2016), model bias analysis is still focused on the upper ocean levels. The overall magnitude of AMOC bias is less than that of NPMOC with significantly reduced biases near the sea surface (Fig. 10), which is consistent with those of surface currents among the three CGCMs. The ECHAM6-NEMO3.6 shows exiguous bias near the ocean surface, but presents strong biases in the mesopelagic zone of subtropical areas, bringing more heat to higher latitudes (Fig. 10a). Likewise, the ECHAM5-NEMO3.6 exhibits strong circulation biases rotating clockwise in the thermocline that intensifies poleward heat transport (Fig. 10b). With similar bias patterns of the AMOC, the ECHAM5-NEMO3.6 and the ECHAM6-NEMO3.6 have opposite SST biases in North Atlantic (Figs. 3a and 3c), which implies that the air-sea feedback including WES feedback and NAM as suggested by Zhang & Zhao (2015) takes the responsibility. The MPI-ESM experiment shows negative biases in tropical Atlantic from the sea surface to the bathypelagic zone, indicating that the overturning circulation has been restrained. There is a narrow positive bias in the subtropical Atlantic, but its strength has been limited by the negative biases nearby. One consequence of the weak AMOC is the decrease of SST in North Atlantic due to less heat supply from the tropics (Fig. 3e). The overturning circulation is enhanced in the middle latitudes with one centre located north of 35ºN and another centre around 55ºN at the depth of 1200m. It still promotes the poleward heat transport and results in warm SST biases in subpolar region (Fig. 3e). The AMOC biases in the MPI-ESM piControl experiment are similar as those in the MPI-ESM experiment, with more negative biases in tropical Atlantic. Comparing the AMOC biases between the MPI-ESM and the ECHAM5-NEMO3.6, it can be seen that the SST cold biases in North Atlantic are partially attributed to decreased MOC in the thermocline of tropical and extra-tropical oceans. However, the air-sea interaction also takes account of the SST variations in consideration of the SST

differences between the ECHAM5-NEMO3.6 and the ECHAM6-NEMO3.6. Zhang & Zhao (2015) suggested that the cold SST bias in Atlantic caused the same cold bias in North Pacific through different mechanisms originating in tropical and extra-tropical Atlantic. Because the differences of NPMOC are bigger than those of AMOC between these two newly developed CGCMs, it suggests an inverse cause-and-effect relationship between the cold SST biases in North Pacific and North Atlantic where the former takes the lead.

[Figure]

**Figure 10: Model biases of AMOC in summer, (a) ECHAM6-NEMO3.6, (b) ECHAM5-NEMO3.6, (c) MPI-ESM, (d) MPI-ESM piControl, Unit: Sv.'**

*2. Changes in mean state from changing component model provide simple but necessary information. However, it would benefit readers and the community if more information such as variability of the system is assessed, such as variance of variables that the authors have covered, and maybe even power spectrum of ENSO and SAM.*

Reply:

Thanks for your comments. We have added assessment of the system variability, including power spectrum analysis of Niño3.4 index, SOI and SAM index. Because the model data in use spans 100-year realizations, much longer than that of the ERA-Interim reanalysis, the ERA-20c has been used instead to evaluate the model variability. The SST reanalysis data from Hadley center (HadISST) is still used for SST variability assessment.

Both HadISST and ERA-20c reanalyses have 100-year time span from 1911 to 2010. Model variability evaluation has been provided in section 4.7.

**'4.7 Model variability of ENSO and SAM**

In the coupled ocean-atmosphere system, global climate variability has been driven by the El Niño-Southern Oscillation (ENSO), the southern annular mode (SAM, also called the Antarctic Oscillation) and the Indian Ocean dipole (IOD) (Philander, 1990; Wallace and Thompson, 2002; Saji et al., 1999). It is therefore necessary to examine the model variability by applying spectra analysis on relevant indices. The CGCM simulations of the three indices are generally consistent with the theoretical red noise (Markov) spectrum (figure omitted). The Niño3.4 index is defined as the SST anomalies averaged over the NINO34 region (5°N-5°S,170°W -120°W). It shows high variance in 2-7 years' period that documents the ENSO peaks in the HadISST reanalysis (Fig. 8a). All of the CGCMs reproduce similar variations of the Niño3.4 power spectra. The ECHAM6-NEMO3.6 presents weak variabilities at the interannual and interdecadal scale, whose periodic peaks are about one year less than the reanalysis counterpart. The ECHAM5-NEMO3.6 shows a better spectral distribution that best coincides with the reanalysis at the interannual scale. However, it still suffers a weak variability at the interdecadal scale and the periodic peak is even half a year less than that of the ECHAM6-NEMO3.6. The MPI-ESM instead takes on an intensified interannual variability, which stays strong at the interdecadal scale. The Southern Oscillation Index (SOI) is calculated based on the differences in sea level pressure anomalies between Tahiti and Darwin in Australia. In comparison to the Niño3.4 spectra, the SOI exhibits similar peaks at the interannual scale in the ERA-20c reanalysis (Fig. 8b). Nevertheless, all the CGCMs reproduce weak variabilities at the interannual and interdecadal scales. The ECHAM6-NEMO3.6 presents the best simulation with a significant increase in variance around 4 years' period, while the ECHAM5-NEMO3.6 shows the weakest variability at the interannual scale. It implies that the AGCM replacement has an opposite effect on the Niño3.4 and SOI variabilities. The MPI-ESM also show reduced variance from the annual scale and above, quite the opposite to that in the Niño3.4 case. Since the model biases of ENSO variability may be attributed to thermocline feedback and zonal wind variations (Borlace et al., 2013), the reversed changes in the variabilities of Niño3.4 and SOI can be caused by the related oceanic and atmospheric processes. The SAM index is calculated following Gong and Wang (1999) by the differences of normalized monthly zonal mean sea level pressure at 40°S and 65°S. Variations of SAM tend to be more flattened than those of SOI in the ERA-20c reanalysis (Fig. 8c), with prominent fluctuations from biannual to interannual scales. Compared with the reanalysis counterpart, all CGCMs show more power at interannual time scales that represents a robust modulation of the SAM, which is possibly attributed to the semi-annual oscillation (SAO) (Hurrell and van Loon, 1994) and circulation anomalies over Antarctica (Thompson and Solomon, 2002). The ECHAM6-NEMO3.6 presents stronger decadal variability than that of the reanalysis data, while the ECHAM5-NEMO3.6 exhibits weaker low-frequency variability. Since the high variance in low-frequency band represents the upward trend of SAM index at decadal scale (Raphael and

Holland, 2006), updating the AGCM can result in a drastic change of long-term climate variability in southern hemisphere. In contrast, the MPI-ESM shows the SAM variability very close to the reanalysis counterpart from 1 year and above, indicating that the OGCM feedback to the atmosphere can lead to a better representation of the inter-decadal variability.

[Figure]

**Figure 8: Power spectra of (a) Niño3.4 index, (b) SOI, (c) SAM index. Solid line denotes the calculation results of reanalysis data, green dotted line denotes the ECHAM5-NEMO3.6 simulation, red dotted line denotes the ECHAM6-NEMO3.6 simulation, and blue dotted line denotes the MPI-ESM simulation.'**

*3. Section 6.2: Please be aware that correlation does not indicate intensity or magnitude of relationship. Fig. 13 shows that shortwave radiation has a much large magnitude and can thus possibly have a greater effect on SST even though it has a smaller correlation value compared to sensible heat and longwave radiation. This does not change the proposed mechanism, but SW radiation should be incorporated into the explanation.*

Reply:

Thank you for these comments. We have rewritten this section to explain the model differences with a full picture of the air-sea interactions. Relevant content in section 7 has also been revised. Major modifications are pasted below.

'… Since the top 3 variables that are most relevant to SST deviations (Tab. 1) suggest changes in the surface heat budget, involving the momentum and temperature exchanges between ocean and atmosphere, it implies that the joint effects of atmosphere and ocean models lead to the model deviation patterns (Fig. 15). Simulation of ocean dynamics is different after replacing the OGCM. With changes in ocean advection, the SST and surface currents are altered which modulates the surface evaporation, convection and heat conduction. Subsequently the latent heat and sensible heat fluxes vary over the sea surface. The thermal and moisture perturbations from the ocean are passed to the atmosphere during coupling processes (Fig. 15c, d). Variations of low-level atmospheric circulation and humidity take effects on cloud formation and cloud liquid water path that changes precipitation and cloud radiative forcing. The net shortwave and longwave radiations are influenced and make a difference to the atmospheric circulation (Fig. 12c) and the heat budget over the sea surface (Fig. 15 a, b). Then, the perturbation signal is transferred back to the ocean that changes the SST and surface currents. This air-sea feedback finally reaches a quasi-equilibrium with marked SST warming over vast maritime spaces over the globe. The associated physical processes represented by each radiation term in Figure 15 are in accordance with their signs of pattern correlation (Tab. 1). It seems to suggest that surface evaporation plays a predominant role in the SST differences, because the latent heat flux holds the biggest correlation coefficient and its deviation pattern is the most obvious among the four radiation terms. However, it is more likely a manifestation of the large-scale ocean dynamical effect on the inter-model differences as suggest by Ying and Huang (2016).

… So far, the analysis has made it clear that changing the OGCM affects the SST simulation through large-scale air-sea feedback that mainly involves surface evaporation, heat conduction, atmospheric convection and cloud masking of incoming and outgoing radiative fluxes. Under the impact of net radiation surplus over western Pacific, the temperature rises with upward motion which forms a warm advection heading east and descending over eastern Pacific…..'

*4. Along the lines of pattern correlation being used to rank the importance of contributing factors. Since the region of focus is the Pacific, the pattern correlation should does be performed over the Pacific rather than globally. This should not change the concluding results, but it may provide some differences in the correlation and add robustness to your estimates.*

Reply:

Thank you for these comments. We have added the correlation analysis over the Pacific to facilitate the discussion of related physical processes in section 6. As you suggested before, the assessment of SAM index variability and AMOC have been added to the paper. The focus of the paper has thus been expanded to other oceans, and we still keep the previous correlation analysis performed at the global scale in the manuscript. Major modifications are as follows:

'… Because the analysis in previous sections is mainly focused on the Pacific, the pattern correlation over the area is also provided for comparison (Table 2).

… When the study area is narrowed down to the Pacific, the top 3 ranking variables are the same, with the net shortwave radiation takes the 1st place (Tab. 2). In both cases, the net shortwave and net longwave fluxes are ranked in top 3, which suggests that radiation budget is closely associated with SST inter-model differences. The pattern correlations of the 3 variables are higher within the Pacific than that at the global scale, indicating an increased robustness of the variable estimates.

… For simulation differences after changing the OGCM, latent heat flux holds the biggest share in pattern correlation with corresponding SST differences (Tab. 1), much bigger than that of net longwave flux at the second place and sensible heat flux at the third place. When the study area is confined within the Pacific, the three leading variables are the same but the sensible heat flux is ranked top (Tab. 2). Compared with the rankings in the AGCM case, it can be seen that the OGCM influences the simulation with different physical mechanisms than those by replacing AGCM. The effect of cloud radiation feedback is not prominent, while the physical processes associated with latent heat and sensible heat, including heat conduction, evaporation and convection, take a bigger share in the SST inter-model differences. SST differences can also lead to changes in longwave flux.'

*5. Please state significance test for pattern correlation, particularly how the effective degree of freedom is computed or decorrelation length scale used. Having a significance at 99.9% level for a correlation at 0.018 is hard to reckon with.*

Reply:

Thank you for pointing out the problem. We applied the Student-t test on the correlation coefficients, and the degree of freedom is taken as n-2 (n is the number of grid points over the globe). Because the model data has been interpolated onto the 1x1 grid, which makes the sample

points too large even without land points. The threshold value for 99.9% is 0.0174. We have added information of significance test in the table caption as follows:

'Asterisk (*) denotes the correlation coefficients passing the Student-t test above 99.9% confidence level. A larger number of grid points are involved that makes the threshold value relatively small. Numbers in boldface denotes the maximum absolute correlation value for the AGCM or OGCM replacement.'

*6. Page 6, line 13: Which year of external forcing (like CO2, greenhouse gases, aerosol, etc.) was used for the piControl run to obtain equilibrated state? Is this the same (similar) forcing as seen during observation period? If the piControl uses forcing that is quite different from observations (e.g. excess of 100ppm of CO2), then one would have to account for these differences when computing biases with respect to observations.*

Reply:

Thank you for these comments. The MPI-ESM experiment is configured based on the piControl run but not restricted to every parameter value. The external forcing is consistent with the present time climatology. The CO2 value is set to default 353.9 ppm in the user manual. Other greenhouse gases like NO2 also follows the default present time setting so that they are consistent with each other. The aerosol settings use the climatology compiled by S. Kinne without any complementation of volcanic aerosols. Model initialization is started from the climatology basic state recalculated with the AMIP run input data from 1981 to 2010. The piControl run data used in this study has the same time span as that of the reanalysis data from the year 1981 to 2010, which should be able to represent the model abilities in reproducing the climatology of the same period. Relevant information has been added to section 2.4.

'**2.4 Experimental Setup**

The control experiments conducted in this study are aimed to reproduce atmospheric and oceanic circulation characteristics of present time, and then to compare with each other in order to examine model performance improvements. The coupled control simulation is thus configured with reference to MPI-ESM piControl experiment settings. Model initialization is started from the climatology basic state recalculated with the AMIP run input data from 1981 to 2010. The CO2 value is set to default 353.9 ppm in the user manual. Other greenhouse gases like NO2 also follows the default present time setting so that they are consistent with each

other. The aerosol settings use the climatology compiled by S. Kinne without any complementation of volcanic aerosols.…'

*7. page 12, line 7-8: Rather than inferring the relationship, the authors can easily compute the pattern correlation and quantify the correlation.*

Reply:

Thanks for your suggestion. We have calculated the pattern correlation between temperature biases and easterly anomalies in the three CGCM experiments. Because the easterly biases are most remarkable below 800hPa, the levels taken into consideration are between 800hPa and 1000hPa. The pattern correlation coefficients between temperature biases and zonal wind biases are 0.315 for the ECHAM6-NEMO3.6 experiment, 0.586 for the ECHAM5-NEMO3.6 experiment, and 0.411 for the MPI-ESM experiment. All of them have passed the 99% Student-t significance test. The ECHAM5-NEMO3.6 has the most significant easterly biases and holds the largest correlation coefficient, much bigger than that of the other two CGCMs, which indicates a strong correlation between the warmer temperature and stronger easterly anomalies. The analysis result has been added to section 5.2.

'It can be seen from Figure 11 that cold temperature biases in the MPI-ESM transform into warm temperature biases in the ECHAM5-NEMO3.6 accompanied by increasing easterly anomalies in lower atmosphere, which suggests warmer temperature bias is correlated with stronger easterly anomalies. The pattern correlation between temperature biases and easterly anomalies in the three CGCM experiments has been calculated respectively. Because the easterly biases are most remarkable below 800hPa, the levels taken into consideration are between 800hPa and 1000hPa. The pattern correlation coefficients between temperature biases and zonal wind biases are 0.315 for the ECHAM6-NEMO3.6 experiment, 0.586 for the ECHAM5-NEMO3.6 experiment, and 0.411 for the MPI-ESM experiment. All of them have passed the 99% Student-t significance test. The ECHAM5-NEMO3.6 has the most significant easterly biases and holds the largest correlation coefficient, much bigger than that of the other two CGCMs, which confirms the correlation between the warmer temperature and stronger easterly anomalies.'

Minor comments

*1. page 6, line 11: please provide the actual websites*

Reply:

We have added the URLs of the websites for MPIOM and NEMO.

'…namelist settings of MPIOM and NEMO3.6 still follow their own default settings for control run provided by their respective official websites (http://www.mpimet.mpg.de/en/science/models/mpi-esm/mpiom and https://www.nemo-ocean.eu).'

2.  *page 8, line 31-32: Been staring at Fig. 4 but am not able to see "positive precipitation bias over the southern tropical Atlantic and negative bias on tropical South America". Maybe you meant "negative bias over southern tropical Atlantic and positive bias on tropical South America"? Please indicate which specific figures you are referring to.*

Reply:

Thank you for pointing out the problem. We try to describe the variational trends of inter-model differences of precipitation that are consistent with previous research finding. It refers to the model biases in Figures 4a, 4c, 4e. We have corrected some imprecise expressions in the context. The focus areas have been outlined in the figure below.

'…As suggested by Stevens et al. (2013), the extent of precipitation bias over the tropical Atlantic and South America seems related to each other, because the ECHAM5-NEMO3.6 shows less biases than those of the other two CGCMs with less blue contours over the tropical Atlantic and lighter red contours over Colombia and Venezuela (Fig. 4a, c, e) ….

[Figure]

,

3. *page 8, line 32: "… seems related to each other, because the values and scales of biases in … "Please explain, it is not obvious how this is the case.*

Reply:

As response to the above question, the simulated precipitation exhibits inter-model differences within the boxes. The ECHAM5-NEMO3.6 shows less biases than those of the other two CGCMs, with less blue contours over the tropical Atlantic and lighter red contours over Colombia and Venezuela. The simultaneous reduction of precipitation biases coincides with the research findings of Stevens et al. (2013), who point out that the precipitation biases over the two areas are possibly related. We have further modified the paper content to clarify the issue.

'…As suggested by Stevens et al. (2013), the extent of precipitation bias over the tropical Atlantic and South America seems related to each other, because the ECHAM5-NEMO3.6 shows less biases than those of the other two CGCMs with less blue contours over the tropical Atlantic and lighter red contours over Colombia and Venezuela (Fig. 4a, c, e) ….'

4.  *Fig. 5: Please provide magnitude difference in surface wind stress in colour. This would make it easier for readers to see extent of anomalies.*

Reply:

We have shaded the vectors in Figure 5 as you suggest.

'

[Figure]

**Figure 5: The same as Fig. 3 but for simulated surface wind stress (unit: N/m².).**'

5.  *page 9, line 15: Change "sea water to across" to "sea water across"*

Reply:

Thank you for pointing out the grammar mistake. We have revised the sentence.

'…which can drive the underlying sea water across the date line in the other two CGCMs.'

6.  *page 10, line 13: Change "it turns out that three CGCMs" to "it turns out that all three CGCMs"*

Reply:

Thank you for carefully reading. We have revised the sentence as you suggest.

'…and it turns out that all three CGCMs suffer from a similar bias pattern in boreal summer ….'

*7. page 11, line 11: Change "," to "."*

Reply:

Thank you for pointing out the problem. We have corrected the mistake.

'Bias in the MPI-ESM experiment manifests itself to a larger degree (Fig. 8c).'

*8. page 11, line 31: Change "In consistence" to "Consistent"*

Reply:

Thanks for your advice. We have revised the sentence as you suggest.

'Consistent with surface wind stress bias (Fig. 5), ….'

*9. page 13, line 3: Change "it can be assumed" to "it would suggest"*

Reply:

Thanks for your suggestion. We have revised the sentence as you suggest.

'Noting that latent heat flux has a correlation value close to that of net longwave radiation, it would suggest that cumulus convection changes sea surface winds and then surface evaporation, ….'

*10. page 15, line 22: None of the CGCMs used on this paper resolved mesoscale processes, so the substance "depending on mesoscale processes in the atmospheric and oceanic systems" cannot be concluded from this study and should thus be omitted.*

Reply:

Thank you for your advice. We have deleted the improper content from the sentence.

'…model errors of each CGCM are caused by different physical mechanisms.'

*11. Please state units for Fig. 5, 6, 8, 9, 10, 11.*

Reply:

We have added the units in figure captions.

'

[Figure]

Figure 5: The same as Fig. 3 but for simulated surface wind stress (unit: N/m²).

[Figure]

**Figure 6: The same as Fig. 3 but for simulated surface currents (unit: m/s).**

[Figure]

**Figure 8: Model biases of summer climatology of meridional overturning circulation simulation in North Pacific, (a) ECHAM6-NEMO3.6, (b) ECHAM5-NEMO3.6, (c) MPI-ESM, (d) MPI-ESM piControl, Units: Sv.**

[Figure]

**Figure 9: Model biases of the vertical structure of atmospheric circulation (vector) and the temperature (contour) over SEC area in summer for (a) ECHAM6-NEMO3.6, (b) ECHAM5-NEMO3.6, (c) MPI-ESM. The SEC region is marked as the blue box in (d). Vector units: m/s, contour units: °C.**

[revised manuscript text omitted]

atmosphere ocean model ECHAM5/MPI-OM (Jungclaus et al., 2006), which has been well tested and reassured of accurate surface flux transfer between the oceanic and atmospheric component models. Based on the interchange coupling structure of the ECHAM5/MPI-OM, seventeen variables are passed from ECHAM5 to NEMO-3.6st through OASIS3 coupler, including solar radiation, non-solar heat flux and its derivative with respect to temperature, zonal and meridional wind stress, evaporation minus precipitation, and sublimation. The meridional and zonal wind stress vectors are passed to the ORCA2 U and V grid of NEMO-3.6st, while other variables are passed to the T grid. Both ocean and ice regions are considered in the coupling processes. In the opposite direction, six variables comprised of the SST, sea ice temperature, sea ice fraction, sea ice albedo and surface ocean currents are transferred from the ocean model to the atmosphere model. Bilinear interpolation method is used in the exchanges of physical variables between ECHAM5 gaussian grid and NEMO-3.6st ORCA2 grid. The time steps for ocean and atmosphere models are both set to 1200 seconds, and the coupling frequency is 4 hours once (every 12-time steps). At present, the component models are integrated with the default parameter settings suggested in the user manual, model retuning will be scheduled in further research.

**2.3.2 ECHAM6-NEMO3.6**

Following the coupling framework of MPI-ESM (the update version of ECHAM5/MPI-OM, version number MPI-ESM-1.2.00p4), ECHAM6-NEMO3.6 has been developed with the same atmospheric component model ECHAM-6.3, coupled with NEMO 3.6 stable version through OASIS3-MCT (Craig et al., 2017) (Fig. 1b). Model retuning of the ECHAM-6.3 and NEMO3.6 is left for further studies due to time limitation and high computational cost. Namelist settings of the ECHAM-6.3 and the coupler OASIS3-MCT are brought into correspondence with those in ECHAM5-NEMO3.6 to the utmost, for example, with the same horizontal resolution T63 on gaussian grid and the same parameterization settings for greenhouse gases. The oceanic component model NEMO3.6 still uses the same configuration as that in ECHAM5-NEMO3.6. Coupling variables of the ECHAM6-NEMO3.6 are the same as those in ECHAM5-NEMO3.6. Details in experiment configuration are elaborated in section 2.4.

**2.3.3 MPI-ESM**

The MPI-ESM is comprised of the atmospheric general circulation model ECHAM6 and the oceanic circulation model MPIOM (Jungclaus et al., 2013; Stevens et al., 2013). It has been continuously developed at Max Planck Institute for Meteorology and has been successfully applied on a broad range of studies, including volcano studies (Zanchettin et al. 2013; Zhang et al. 2013), anthropogenic land cover change (Reick et al., 2013; Brovkin et al., 2013), circulation feedback sensitive to Intertropical Convergence Zone (ITCZ), and double ITCZ precipitation (Mobis and Stevens, 2012). MPI-ESM has been used in CMIP5 and is employed in the upcoming CMIP6. Due to high computational cost, the low-resolution version MPI-ESM-LR is used in this study (version number MPI-ESM-1.2.00p4), with ECHAM6 running at T63L47 resolution (horizontal resolution about 1.875° and 47 vertical levels). Experimental settings are migrated from piControl

default configuration, and then adjusted to being in consistence with those of ECHAM5-NEMO3.6 and ECHAM6-NEMO3.6 experiments. The major differences from default piControl experiment are the increased coupling frequency (4 hours once), and climatology recalculated from the year 1981 to 2010 to serve as the model input files (e.g. aerosol properties, ozone mole fractions and land use transitions).

**2.4 Experimental Setup**

The control experiments conducted in this study are aimed to reproduce atmospheric and oceanic circulation characteristics of present time, and then to compare with each other in order to examine model performance improvements. The coupled control simulation is thus configured with reference to MPI-ESM piControl experiment settings. The $CO_2$ value is set to default 353.9 ppm in the user manual. Other greenhouse gases like $NO_2$ also follows the default present time setting so that they are consistent with each other. The aerosol settings use the climatology compiled by S. Kinne without any complementation of volcanic aerosols. Model initialization is started from the climatology basic state recalculated with the AMIP run input data from 1981 to 2010. The atmospheric component models (ECHAM-5.4 and ECHAM-6.3) used in this study are running on T63 gaussian grid (approximately equivalent to $1.875° \times 1.875°$ on average), with the same coupling frequency of 4 hours to exchange momentum and heat fluxes with the ocean component model. Physical parameterization schemes relative to solar irradiance, aerosol optical properties, cumulus convection and strong stratospheric damping are maintained the same among ECHAM5-NEMO3.6, ECHAM6-NEMO3.6, and MPI-ESM experiments. Since the timestep length of ECHAM-6.3 in coupled mode is suggested to be 450 seconds according to its user manual, only the ECHAM5-NEMO3.6 experiment uses 1200 seconds as timestep length for the atmospheric model ECHAM-5.4. The oceanic component models (MPIOM and NEMO3.6) possess different model structures and mapping technologies, making it impossible to directly migrate physical parameterization settings from one to the other. In this regard, namelist settings of MPIOM and NEMO3.6 still follow their own default settings for control run provided by their respective official websites (http://www.mpimet.mpg.de/en/science/models/mpi-esm/mpiom and https://www.nemo-ocean.eu). Three coupled experiments, namely ECHAM5-NEMO3.6, ECHAM6-NEMO3.6, and MPI-ESM experiments, have been conducted for 200-year realizations including spin-up runs. Model results spanning the last 100 years (model year 101 to 200) should be well-equilibrated, and thus they are used to compute simulation climatology for the inter-model comparison analysis.

**3. Reanalysis Data**

The assessment of model performance with reference to each CGCM employs the monthly data from the Hadley Center (HadISST) (Rayner et al., 2003). Model precipitation evaluation uses reanalysis data of the Global Precipitation Climatology Project (GPCP) (Adler et al., 2003). Evaluation of the mean sea level pressure, zonal and meridional winds at 10m height, cloud cover and surface temperature use the ERA-Interim monthly reanalysis data (Simmons et al., 2006). Due to the limited time span of the ERA-Interim, the ERA-20c reanalysis has been used instead for the evaluation of ENSO and SAM variability. For Surface

wind stress data from the Scatterometer Climatology of Ocean Winds (SCOW) including QuikSCAT measurements (Risien & Chelton, 2008) has been chosen for its more advanced stress-measuring instrument and better sampling, which has been widely used in researches regarding oceanic circulation and dynamical processes (Kanzow et al., 2010; Roquet et al., 2011; Johnson et al., 2012) and evaluation of CGCMs and reanalysis data (Xue et al., 2011; Lee et al., 2013). To characterize the changes in ocean circulation associated with the SST bias, the SODA reanalysis data (Carton & Giese, 2008) has been used following the massive researches on ocean variability and mechanisms (Dewitte et al., 2009; Tett et al., 2014; Drenkard & Karnauskas, 2014). Finally, surface net radiation flux from CERES EBAF-Surface Ed4.0 (Kato et al., 2013) is employed for its higher accuracy by using more accurate cloud data to calculate solar radiation at the Earth's surface (Wild et al., 2013; Wild et al., 2015; Zhang et al., 2015), which is better than popular reanalysis data sets including NCEP-DOE, MERRA and ERA-Interim (Zhang et al., 2016). Because the local mesh refinements of ORCA2 grid in NEMO3.6 make meridional resolution finer in tropics (about 0.5°) than that of higher latitudes (
[revised manuscript text omitted]
 ECHAM5-NEMO3.6 shows less biases than those of the other two CGCMs with less blue contours over the tropical Atlantic and lighter red contours over Colombia and Venezuela (Fig. 4a, c, e). Inter-model simulation differences may thus imply an important role of atmospheric dynamics simulation, for example storm-track and ITCZ, in reducing precipitation biases. Rainfall biases in winter exhibit larger dry biases in tropical Pacific, also flanked by wet biases in subtropics with significantly overestimation in South Africa and South Indian Ocean.

Model improvements of the precipitation simulation can be summarized as follows: The ECHAM6-NEMO3.6 reproduces the best overall precipitation climatology with the highest pattern correlation coefficient (Fig. 2). ECHAM6-NEMO3.6 model ameliorates double ITCZ problem over SPCZ in boreal summer, while ECHAM5-NEMO3.6 best decreases the wet bias over tropical Atlantic.

**4.4 Surface Wind Stress**

[revised manuscript text omitted]

**4.7 Model variability of ENSO and SAM**

In the coupled ocean-atmosphere system, global climate variability has been driven by the El Niño-Southern Oscillation (ENSO), the southern annular mode (SAM, also called the Antarctic Oscillation) and the Indian Ocean dipole (IOD) (Philander, 1990; Wallace and Thompson, 2002; Saji et al., 1999). It is therefore necessary to examine the model variability by applying spectra analysis on relevant indices. The CGCM simulations of the three indices are generally consistent with the theoretical red noise (Markov) spectrum (figure omitted). The Niño3.4 index is defined as the SST anomalies averaged over the NINO34 region (5°N-5°S,170°W -120°W). It shows high variance in 2-7 years' period that documents the ENSO peaks in the HadISST reanalysis (Fig. 8a). All of the CGCMs reproduce similar variations of the Niño3.4 power spectra. The ECHAM6-NEMO3.6 presents weak variabilities at the interannual and interdecadal scale, whose periodic peaks are about one year less than the reanalysis counterpart. The ECHAM5-NEMO3.6 shows a better spectral distribution that best coincides with the reanalysis at the interannual scale. However,

it still suffers a weak variability at the interdecadal scale and the periodic peak is even half a year less than that of the ECHAM6-NEMO3.6. The MPI-ESM instead takes on an intensified interannual variability, which stays strong at the interdecadal scale. The Southern Oscillation Index (SOI) is calculated based on the differences in sea level pressure anomalies between Tahiti and Darwin in Australia. In comparison to the Niño3.4 spectra, the SOI exhibits similar peaks at the interannual scale in the ERA-20c reanalysis (Fig. 8b). Nevertheless, all the CGCMs reproduce weak variabilities at the interannual and interdecadal scales. The ECHAM6-NEMO3.6 presents the best simulation with a significant increase in variance around 4 years' period, while the ECHAM5-NEMO3.6 shows the weakest variability at the interannual scale. It implies that the AGCM replacement has an opposite effect on the Niño3.4 and SOI variabilities. The MPI-ESM also show reduced variance from the annual scale and above, quite the opposite to that in the Niño3.4 case. Since the model biases of ENSO variability may be attributed to thermocline feedback and zonal wind variations (Borlace et al., 2013), the reversed changes in the variabilities of Niño3.4 and SOI can be caused by the related oceanic and atmospheric processes. The SAM index is calculated following Gong and Wang (1999) by the differences of normalized monthly zonal mean sea level pressure at 40°S and 65°S. Variations of SAM tend to be more flattened than those of SOI in the ERA-20c reanalysis (Fig. 8c), with prominent fluctuations from biannual to interannual scales. Compared with the reanalysis counterpart, all CGCMs show more power at interannual time scales that represents a robust modulation of the SAM, which is possibly attributed to the semi-annual oscillation (SAO) (Hurrell and van Loon, 1994) and circulation anomalies over Antarctica (Thompson and Solomon, 2002). The ECHAM6-NEMO3.6 presents stronger decadal variability than that of the reanalysis data, while the ECHAM5-NEMO3.6 exhibits weaker low-frequency variability. Since the high variance in low-frequency band represents the upward trend of SAM index at decadal scale (Raphael and Holland, 2006), updating the AGCM can result in a drastic change of long-term climate variability in southern hemisphere. In contrast, the MPI-ESM shows the SAM variability very close to the reanalysis counterpart from 1 year and above, indicating that the OGCM feedback to the atmosphere can lead to a better representation of the inter-decadal variability.

**5. Circulation Patterns Relevant to SST Biases**

The importance of Meridional Overturning Circulation (MOC) to SST in coupled models has been proved in previous studies (Wang et al., 2014, Liu et al., 2016). Thus, the comparison of North Pacific MOC (NPMOC) (Fig. 9) between the CGCMs with the same atmospheric or oceanic component model can help to explain bias characteristics in relation to SST deviations. Model bias pattern can be influenced by the choice of ocean reanalysis dataset. Uncertainties in the ocean reanalyses (ORAs) include the estimation of sea ice thickness, interannual variability of salinity, surface heat flux and mixed layer depth (Balmaseda et al., 2015; Toyoda et al., 2017). There are substantive discrepancies in temperature, salinity and density in the deep ocean and the Southern Ocean, where observations are sparse especially before the year 2000. Ocean heat content at deep levels varies largely among the ORAs, with majority of spread originating in the Southern Hemisphere due to lack of observation for data assimilation (Palmer et al., 2017). Discrepancies among the

ORAs are evident in the strength and structures of the AMOC, with distinctive differences in the depth of equatorward return flow and the depth of the maximum AMOC in northern high latitudes (Karspeck et al., 2017). However, the uncertainties of climatology above bathypelagic zone are less than those of deeper levels. Since the SODA reanalysis data has been employed in many ocean studies, including the North Pacific MOC that is mostly within the thermocline layer, it is reliable to use the SODA reanalysis data to analyse model biases related to the MOC.

**5.1 Meridional Overturning Circulation**

The ECHAM5-NEMO3.6 possesses two prominent improvements of the SST simulation, one in the Pacific cold tongue region, and the other in North Atlantic. It also exhibits warm SST biases in the North Pacific that is the opposite to most CMIP CGCMs. These peculiar characteristics require further investigation on the meridional overturning circulation (MOC) in North Pacific and North Atlantic. For the MOC in North Pacific, the ECHAM5-NEMO3.6 and ECHAM6-NEMO3.6 possess similar bias patterns overall, with intensified tropical cell and deep tropical cell that are mentioned in Liu et al. (2011) (Fig. 9a, b). Tropical cell enhancement is more significant in the ECHAM5-NEMO3.6, so that the upwelling in the tropics and subsequent heat transport to mid-latitudes are more than those in the ECHAM6-NEMO3.6. Bias in the MPI-ESM experiment manifests itself to a larger degree (Fig. 9c). The piControl experiment result (available in http://esgf-node.llnl.gov/) is attached (Fig. 9d) to demonstrate that the prominent biases in MPI-ESM are not caused by increasing coupling frequency from one day to 4 hours. The piControl run data used in this study has the same time span as that of the reanalysis data from the year 1981 to 2010, which should be able to represent the model abilities in reproducing the climatology of the same period. More contour lines appear in NPMOC bias distribution in piControl than the MPI-ESM setting in this paper (Fig. 9c, d), suggesting obvious improvements after decreasing coupling interval. This is consistent with previous studies (Bernie et al., 2008; Ge et al., 2017). Enhancement of tropical cell in MPI-ESM is unremarkable, compared with that of ECHAM5-NEMO3.6, but deep tropical cell is significantly intensified that tropical upwelling is forced to become too strong. This may explain why tropical SST bias of MPI-ESM is more than 2℃ in boreal summer, without significant radiation errors and easterly anomalies in surface stress. Excessive upwelling in tropical Pacific, induced by intensified deep tropical cell of NPMOC, cools down the local SST. Unlike the ECHAM5-NEMO3.6 case, where the SST cooling in tropical Pacific is driven by intensified tropical cell of NPMOC that transports more heat to mid-latitudes, the MPI-ESM simulation of tropical and subtropical cells does not differ much from SODA reanalysis data. The resulting poleward heat transport carried by NPMOC is not increased in the MPI-ESM experiment. Therefore, the SST bias in North Pacific remains negative (Fig. 3e) under the impact of Northern Hemisphere annular mode (NAM) and wind-evaporation-SST (WES) feedback, when cold SST biases appear in tropical and extratropical North Atlantic (Zhang & Zhao, 2015). The cold SST bias in North Pacific and North Atlantic for ECHAM6-NEMO3.6 can also be explained with the same reason. But for the ECHAM5-NEMO3.6 case, the poleward heat transport has been enhanced by intensified tropical cell to bring up the SST in mid-latitudes (Fig. 3c).

Since the upper cell of Atlantic meridional overturning circulation (AMOC) plays a significant role in delaying warming signals from anthropogenic greenhouse gases and responding to climate change (Marshall et al., 2014; Buckley and Marshall, 2016), model bias analysis is still focused on the upper ocean levels. The overall magnitude of AMOC bias is less than that of NPMOC with significantly reduced biases near the sea surface (Fig. 10), which is consistent with those of surface currents among the three CGCMs. The ECHAM6-NEMO3.6 shows exiguous bias near the ocean surface, but presents strong biases in the mesopelagic zone of subtropical areas, bringing more heat to higher latitudes (Fig. 10a). Likewise, the ECHAM5-NEMO3.6 exhibits strong circulation biases rotating clockwise in the thermocline that intensifies poleward heat transport (Fig. 10b). With similar bias patterns of the AMOC, the ECHAM5-NEMO3.6 and the ECHAM6-NEMO3.6 have opposite SST biases in North Atlantic (Figs. 3a and 3c), which implies that the air-sea feedback including WES feedback and NAM as suggested by Zhang & Zhao (2015) takes the responsibility. The MPI-ESM experiment shows negative biases in tropical Atlantic from the sea surface to the bathypelagic zone, indicating that the overturning circulation has been restrained. There is a narrow positive bias in the subtropical Atlantic, but its strength has been limited by the negative biases nearby. One consequence of the weak AMOC is the decrease of SST in North Atlantic due to less heat supply from the tropics (Fig. 3e). The overturning circulation is enhanced in the middle latitudes with one centre located north of $35^{\circ}$ N and another centre around $55^{\circ}$ N at the depth of 1200m. It still promotes the poleward heat transport and results in warm SST biases in subpolar region (Fig. 3e). The AMOC biases in the MPI-ESM piControl experiment are similar as those in the MPI-ESM experiment, with more negative biases in tropical Atlantic. Comparing the AMOC biases between the MPI-ESM and the ECHAM5-NEMO3.6, it can be seen that the SST cold biases in North Atlantic are partially attributed to decreased MOC in the thermocline of tropical and extra-tropical oceans. However, the air-sea interaction also takes account of the SST variations in consideration of the SST differences between the ECHAM5-NEMO3.6 and the ECHAM6-NEMO3.6. Zhang & Zhao (2015) suggested that the cold SST bias in Atlantic caused the same cold bias in North Pacific through different mechanisms originating in tropical and extra-tropical Atlantic. Because the differences of NPMOC are bigger than those of AMOC between these two newly developed CGCMs, it suggests an inverse cause-and-effect relationship between the cold SST biases in North Pacific and North Atlantic where the former takes the lead.

**5.2 Vertical Structure of Atmospheric Circulation**

[revised manuscript text omitted]

**6.2 Impacts of OGCM Replacement**

For simulation differences after changing the OGCM, latent heat flux holds the biggest share in pattern correlation with corresponding SST differences (Tab. 1), much bigger than that of net longwave flux at the second place and sensible heat flux at the third place. When the study area is confined within the Pacific, the three leading variables are the same but the sensible heat flux is ranked top (Tab. 2). Compared with the rankings in the AGCM case, it can be seen that the OGCM influences the simulation with different physical mechanisms than those by replacing AGCM. The effect of cloud radiation feedback is not prominent, while the physical processes associated with latent heat and sensible heat, including heat conduction, evaporation and convection, take a bigger share in the SST inter-model differences. SST differences can also lead to changes in longwave flux. The negative sign of correlations for latent heat, sensible heat and longwave flux (Tab. 1 and 2) suggests a reverse trend to SST variations.

Contrary to the AGCM case, low-level wind deviations in tropical Pacific (Fig. 12c) are in the same direction as the background climatology (Fig. 12a), which should increase surface evaporation through stronger near surface winds and then results in colder SST through WES feedback. Nevertheless, warmer SST appears in most parts of tropical and subtropical oceans, laying waste to the assumption trying to explain SST deviation in terms of WES mechanism. Since the top 3 variables that are most relevant to SST deviations (Tab. 1) suggest changes in the surface heat budget, involving the momentum and temperature exchanges between ocean and atmosphere, it implies that the joint effects of atmosphere and ocean models lead to the model deviation patterns (Fig. 15). Simulation of ocean dynamics is different after replacing the OGCM. With changes in ocean advection, the SST and surface currents are altered which modulates the surface evaporation, convection and heat conduction. Subsequently the latent heat and sensible heat fluxes vary over the sea surface. The thermal and moisture perturbations from the ocean are passed to the atmosphere during coupling processes (Fig. 15c, d). Variations of low-level atmospheric circulation and humidity take effects

on cloud formation and cloud liquid water path that changes precipitation and cloud radiative forcing. The net shortwave and longwave radiations are influenced and make a difference to the atmospheric circulation (Fig. 12c) and the heat budget over the sea surface (Fig. 15 a, b). Then, the perturbation signal is transferred back to the ocean that changes the SST and surface currents. This air-sea feedback finally reaches a quasi-equilibrium with marked SST warming over vast maritime spaces over the globe. The associated physical processes represented by each radiation term in Figure 15 are in accordance with their signs of pattern correlation (Tab. 1). It seems to suggest that surface evaporation plays a predominant role in the SST differences, because the latent heat flux holds the biggest correlation coefficient and its deviation pattern is the most obvious among the four radiation terms. However, it is more likely a manifestation of the large-scale ocean dynamical effect on the inter-model differences as suggest by Ying and Huang (2016).

Atmospheric circulation accommodates itself to changes of radiation budget. An anomalous Walker circulation cell appears over the SEC area (Fig. 13b), with updraft over Philippine sea where positive SST deviations indicate a net radiation surplus. The upward flow rises to the upper-level troposphere and condensation becomes evident due to temperature drops. It then diverts toward east and orients downward over eastern Pacific, which enhances surface westerly anomalies in the lower atmosphere. This circulation pattern is consistent with anomalous easterlies over tropical Pacific (Fig. 12c). Warm temperature deviations (Fig.13b) can be viewed as the manifestation of surface radiation surplus and latent heat release of condensation. So far, the analysis has made it clear that changing the OGCM affects the SST simulation through large-scale air-sea feedback that mainly involves surface evaporation, heat conduction, atmospheric convection and cloud masking of incoming and outgoing radiative fluxes. Under the impact of net radiation surplus over western Pacific, the temperature rises with upward motion which forms a warm advection heading east and descending over eastern Pacific. This anomalous Walker cell drives low-level winds towards the west, leading to westerly anomalies over vast areas of tropical Pacific.

**7. Summary and Discussion**

In this study, two new CGCMs have been developed based on the coupling structure of MPI-ESM, namely ECHAM5-NEMO3.6 and ECHAM6-NEMO3.6. The new CGCMs show some improvements in the simulation of SST, precipitation and ocean currents compared with MPI-ESM. The ECHAM5-NEMO3.6 presents the best SST simulation in summer with the minimum cold tongue bias in tropical Pacific and no remarkable bias in North Atlantic, while ECHAM6-NEMO3.6 reproduces the best winter climatology with most of the biases less than 1 ºC. For precipitation simulation, ECHAM6-NEMO3.6 presents the highest pattern correlation with reanalysis counterpart and substantially ameliorates double ITCZ problem over SPCZ in boreal summer. Biases in surface currents and meridional overturning circulation are also considerably reduced in both ECHAM5-NEMO3.6 and ECHAM6-NEMO3.6. Wind stress bias patterns are alike in most areas among the three CGCMs, but MPI-ESM shows only poleward anomalies over tropical Pacific without strong easterly biases that are common in most existing coupled models. All three CGCMs can generally reproduce the ENSO and SAM variabilities. Model biases are more evident at the interannual

and interdecadal scales, suggesting a prominent effect of circulation and oscillation anomalies in both atmosphere and ocean models on the air-sea feedback (Yu and Kim, 2011; Farneti, et al. 2014). As suggested by Cai et al. (2011), the evolution of these climate drivers is affected by their interactions, which requires further investigation. The ECHAM5-NEMO3.6 has much larger biases in total radiation than MPI-ESM, whereas the latter presents the maximum deviation in SST simulation. Besides, the ECHAM5-NEMO3.6 shows warm SST bias in North Pacific, with the opposite sign to most CGCM biases at present time. These facts constitute evidence that suggests model errors of each CGCM are caused by different physical mechanisms.

Meridional overturning circulation in North Pacific and vertical structure of atmospheric circulation are analysed for a comprehensive understanding of bias genesis in each CGCM. Overestimation of tropical cell in the NPMOC transfers more heat to mid-latitudes and results in warm SST bias in North Pacific of the ECHAM5-NEMO3.6. Excessively strong deep tropical cell of the MPI-ESM intensifies the tropical upwelling that leads to the largest cold SST bias among all three CGCMs. The analysis on atmospheric vertical circulation over the SEC area has confirmed that momentum field plays a major role in the SST biases of ECHAM5-NEMO3.6 and ECHAM6-NEMO3.6, while oceanic processes and radiation budget are responsible for the same cold SST biases in tropical Pacific. Since the inter-model differences are caused by changing component models, 12 surface variables that are sensitive to coupling processes are chosen to calculate pattern correlation with the SST differences between CGCMs with the same atmospheric or oceanic component model. The top 3 variables ranked in the case of changing the AGCM are net longwave radiation, surface evaporation (latent heat flux) and net shortwave radiation. Differences in SST, radiation budget and atmospheric circulation are analysed, and it is confirmed that the AGCM replacement first militates in the alteration of cumulus convection including temperature, specific humidity and atmospheric circulation, which in turn changes SST through WES feedback and affects the radiation budget through cloud radiation feedback. For the OGCM replacement, the top 3 variables are latent heat flux, sensible heat flux and net longwave flux. Through analysis on circulation and radiation terms, it has been clear that the ocean dynamical effect plays a predominant role in the SST differences after changing the OGCM. The ocean advection initiates the perturbations of SST and surface evaporation that modulate the atmospheric humidity and low-level circulation. Consequently, the cloud masking effect on radiative fluxes are altered which influences the atmospheric circulation and surface heat budget. The resulting equilibrium of the air-sea feedback manifest itself as the inter-model differences in the related meteorological fields. In the OGCM case, the latent heat flux holding the largest correlation coefficient at the global scale is consistent with Cao et al. (2015), which points out that amplitude and meridional variability of latent heat flux over Pacific are the most diverse in CMIP5 models.

It is noteworthy that the SST deviations by changing the OGCM and AGCM are nearly the opposite (Fig. 12d, e), suggesting that reverse transformation of model bias can be realized through mechanisms unveiled in this paper. Although the top 3 coupling variables that are most relevant to SST deviations after changing the AGCM or OGCM are radiation terms, the physical mechanisms behind the opposite SST

variations are different. AGCM replacement affects cumulus convection that eventually changes momentum field and radiation budget over the sea surface, while OGCM replacement alters sea surface evaporation that results in latent heat variations and consequently leads to readjustment of radiation budget. It implies that one can pursue simulation improvement of the cold tongue from two aspects: 1. Improve the cumulus convection scheme in the AGCM, 2. Ameliorate errors of surface radiation budget in the OGCM, especially for latent heat. With better cumulus convection scheme, easterly anomalies over the tropical sea surface can be reduced significantly, which will cut down the latent heat absorption of more evaporation induced by stronger surface winds, and consequently break down the WES feedback that amplifies the cold SST bias. For the OGCM part, better treatment of surface evaporation and radiation budget can reduce the cold SST bias in relation to net radiation deficiency.

For strong easterly bias over eastern Pacific in ECHAM6-NEMO3.6, also common in the most CGCMs, the MPI-ESM instead shows poleward bias, which suggests that OGCM replacement can also diminish this bias through coupling processes. It is easy to see that an anomalous Walker circulation with counter-clockwise rotation will appear over the tropical Pacific, if net surface radiation is warmer in the east and colder in the west. With proper OGCM configuration, sea surface evaporation can initiate this radiation anomaly, and the resulting Walker circulation will decrease the easterly wind speed. Although the analysis in section 5 and 6 only focuses on the physical mechanisms behind simulation differences in summer, the atmospheric and oceanic circulations have a similar impact on radiation budget and surface heat transport in other seasons. It implies that the effects by replacing each component model may be the same, and the conclusions drawn above are also valid for SST biases in winter. The mechanisms illustrated in this study provides a new vision of model bias origin, which is heuristic for model improvement researches and practices.

**Code and data availability**

The model source code is available from the authors upon request. Restrictions on source code license will be imposed unless for academic and non-commercial use. The experimental data can be found in https://doi.org/10.5281/zenodo.1306338 (Gui et al., 2018).

The reanalysis data used in this study can be downloaded from the following websites:

1. The Hadley Centre SST data is downloaded from https://www.metoffice.gov.uk/hadobs/hadisst/.
2. The GPCP precipitation data is downloaded from  https://precip.gsfc.nasa.gov/.
3. The ERA-Interim and ERA-20c monthly reanalysis data is available at http://apps.ecmwf.int/datasets/.
4. The SCOW wind stress data is available at http://cioss.coas.oregonstate.edu/scow.
5. Surface radiation reanalysis data of CERES EBAF-Surface Ed4.0 is downloaded from https://ceres.larc.nasa.gov.
6. The SODA3.3 ocean current reanalysis data is downloaded from: http://www.atmos.umd.edu.

The MPI-ESM piControl experiment data is available at http://esgf-node.llnl.gov.

**Acknowledgements**

This work was supported by the National Key Research and Development Program of China (2016YFA0601600), and the National Natural Science Foundation of China (U1502233 and 41565002). It is also co-supported by Yunnan University's Research Innovation Fund for Graduate Students (YDY17019).

**Table 1: Pattern Correlations of surface variables and SST deviation**

| Model Replacement / Variables | AGCM | OGCM |
|---|---|---|
| Surface zonal currents | 0.006 | -0.002 |
| Surface meridional currents | 0.0158 | -0.018* |
| Sensible heat flux | -0.068* | -0.269* |
| Latent heat flux | -0.262* | **-0.393**\* |
| Mean sea level pressure | -0.136* | -0.009 |
| 10m zonal wind | 0.018* | -0.010 |
| 10m meridional wind | -0.025* | 0.045* |
| Surface albedo | -0.101* | -0.022* |
| Net shortwave flux | 0.192* | 0.138* |
| Net longwave flux | **-0.269**\* | -0.283* |
| Precipitation | 0.100* | 0.154* |

Asterisk (*) denotes the correlation coefficients passing the Student-t test above 99.9% confidence level. A larger number of grid points are involved that makes the threshold value relatively small. Numbers in boldface denotes the maximum absolute correlation value for the AGCM or OGCM replacement.

**Table 2: Pattern Correlations of surface variables and SST deviation over the Pacific**

| Model Replacement Variables | AGCM | OGCM |
|---|---|---|
| Surface zonal currents | 0.069* | -0.033* |
| Surface meridional currents | 0.055* | -0.136* |
| Sensible heat flux | -0.286* | **-0.318*** |
| Latent heat flux | -0.302* | -0.315* |
| Mean sea level pressure | -0.246* | 0.014 |
| 10m zonal wind | 0.070* | 0.039* |
| 10m meridional wind | 0.022 | -0.005 |
| Surface albedo | 0.274* | 0.034* |
| Net shortwave flux | **0.375*** | 0.102* |
| Net longwave flux | -0.303*** | -0.229* |
| Evaporation | -0.302* | -0.315* |
| Precipitation | 0.103* | 0.124* |

Asterisk (*) denotes the correlation coefficients passing the Student-t test above 99.9% confidence level. A larger number of grid points are involved that makes the threshold value relatively small. Numbers in boldface denotes the maximum absolute correlation value for the AGCM or OGCM replacement.

[Figure]

**Figure 1: Schematic structure of ECHAM5-NEMO3.6 (a), and ECHAM6-NEMO3.6 (b).**

[Figure]

[Figure]

**Figure 2: Taylor diagram that exhibits a statistical comparison between the simulations and reanalysis data of nine selected variables in summer (a) and winter (b). Each number represents one variable: (1) precipitation, (2) mean sea level pressure, (3) zonal winds at 10m height, (4) meridional winds at 10m height, (5) 2m temperature, (6) total radiation flux (net shortwave plus net longwave), (7) SST, (8) sea surface zonal currents, (9) sea surface meridional currents. Upward-pointing triangles, squares and diamonds, respectively, represent the ECHAM6-NEMO3.6, ECHAM5-NEMO3.6, and the MPI-ESM results.**

[Figure]

**Figure 3: Biases of the SST simulation in summer (left column) and winter (right column) corresponding to each CGCM: (a, b) ECHAM6-NEMO3.6, (c, d) ECHAM5-NEMO3.6, (e, f) MPI-ESM.**

[Figure]

**Figure 4: The same as Fig. 3 but for simulated precipitation climatology.**

[Figure]

**Figure 5: The same as Fig. 3 but for simulated surface wind stress (unit: N/m$^2$).**

[Figure]

**Figure 6: The same as Fig. 3 but for simulated surface currents (unit: m/s).**

[Figure]

**Figure 7: The same as Fig. 3 but for simulated total radiation.**

[Figure]

**Figure 8: Power spectra of (a) Niño3.4 index, (b) SOI, (c) SAM index. Solid line denotes the calculation results of reanalysis data, green dotted line denotes the ECHAM5-NEMO3.6 simulation, red dotted line denotes the ECHAM6-NEMO3.6 simulation, and blue dotted line denotes the MPI-ESM simulation.**

[Figure]

**Figure 9: Model biases of summer climatology of meridional overturning circulation simulation in North Pacific, (a) ECHAM6-NEMO3.6, (b) ECHAM5-NEMO3.6, (c) MPI-ESM, (d) MPI-ESM piControl, Unit: Sv.**

[Figure]

**Figure 10: Model biases of AMOC in summer, (a) ECHAM6-NEMO3.6, (b) ECHAM5-NEMO3.6, (c) MPI-ESM, (d) MPI-ESM piControl, Unit: Sv.**

[Figure]

**Figure 11: Model biases of the vertical structure of atmospheric circulation (vector) and the temperature (contour) over SEC area in summer for (a) ECHAM6-NEMO3.6, (b) ECHAM5-NEMO3.6, (c) MPI-ESM. The SEC region is marked as the blue box in (d). Vector units: m/s, contour unit: °C.**

[Figure]

**Figure 12: Summer climatology of 10m wind for (a) reanalysis data, (b) model differences between ECHAM6-NEMO3.6 and ECHAM5-NEMO3.6, (c) model differences between ECHAM6-NEMO3.6 and MPI-ESM. SST simulation differences for (d) between ECHAM6-NEMO3.6 and ECHAM5-NEMO3.6, and (e) between ECHAM6-NEMO3.6 and MPI-ESM. Vector units: m/s, contour unit: °C.**

[Figure]

**Figure 13: Simulation differences in the vertical structure of atmospheric circulation and the temperature over the SEC area: (a) between ECHAM6-NEMO3.6 and ECHAM5-NEMO3.6, (b) between ECHAM6-NEMO3.6 and MPI-ESM. Vector units: m/s, contour unit: °C.**

[Figure]

**Figure 14: Simulation differences in radiation budget between ECHAM6-NEMO3.6 and ECHAM5-NEMO3.6.**

[Figure]

**Figure 15: Simulation differences in radiation budget between ECHAM6-NEMO3.6 and MPI-ESM.**

---

## Author Comment (AC8) · 2 Oct 2018

Reply to general comments

1. *Some more information on the experimental setup would be desirable. How is the ocean initialized, e.g. are World Ocean Atlas ('Levitus') data used? This is in particular of interest since the authors claim that their model ocean is in equilibrium after only 100 years of spin up whereas other modeling groups perform multi-century (Delworth et al., 2006) or even multi-millennial (Müller et al., 2018) spin-up runs to significantly reduce the temperature drift in the ocean where clearly a drift is still visible after 300 or 500 years (Delworth et al., 2012, Fig. 1; Delworth et al., 2006, Fig. 3). Such a drift is best visible in timeseries of the global mean temperature for the surface but also deeper ocean layers, which unfortunately are not provided by the authors and should be added. It is essential for other modelling centres to provide at least a number or even better a timeseries of the TOA radiation (im)balance.*

Reply:

Thanks for these comments. Climate drift is common in the numerical models, which tends to be model dependent (Gupta et al., 2013). Since the simulation improvements of SST in the ECHAM5-NEMO3.6 are remarkable, it has been questioned whether this is caused by climate drift that happens to improve the SST simulation in the selected time period. The OGCM NEMO3.6 is initialized with the default model configuration from the present time climatology, including the World Ocean Atlas (WOA) data. The global SST time series of the ECHAM5-NEMO3.6 with five-year running average are provided below

[Figure]

The SST fluctuations displayed in the figure are confined within 0.1°C, of the same magnitude as Huang et al. (2014) who have evaluated the ICM coupled model based on ECHAM5.4 and NEMO2.3. In the selected time period (model year

101~200), the SST fluctuation shows a similar pattern as other time span. It's definitely better to integrate the three CGCMs for much longer time span, but the computational cost is high and thousands of years' realization is time-consuming. Since the most efficient way to ameliorate the unrealistic nonlinear oscillation in a weather model is through flux adjustment, which nevertheless produces undesirable results, it may be acceptable to evaluate the model performance based on the quasi-equilibrium state.

2. *For both the Atmosphere (ECHAM5 and ECHAM6) and the ocean models (NEMO, MPI-OM) very little information is provided on the technical details except for the configuration of the coupler (e.g., which parametrizations are active, which model options are switched on or off, how are the model components initialized, is nudging or restoring used in the ocean or atmosphere, model tuning)*

Reply:

Thanks for these comments. Because the models have different parameterizations and even different dynamical structures, it is less useful to provide the technical details about the parameter settings that does little help for the inter-model comparison. By emphasizing on model parameterization differences, it seems to say that the inter-model differences are resulted from the intrinsic characteristics of each component model. Since the simulation deficiencies are always model-dependent, there is no need to investigate the key mechanisms that shape the inter-model differences. This is completely opposite to the purpose of this paper. I'm afraid that the reader's attention will be diverted by those kinds of information and thus I did not provide many details.

Model initialization is started from the climatology basic state recalculated with the AMIP run input data from 1981 to 2010. In the revised manuscript, parameter settings for greenhouse gases and aerosols have been supplemented as follows:

'The $CO_2$ value is set to default 353.9 ppm in the user manual. Other greenhouse gases like $NO_2$ also follows the default present time setting so that they are consistent with each other. The aerosol settings use the climatology compiled by S. Kinne without any complementation of volcanic aerosols.'

Model retuning is left for further studies due to time limitation and high computational cost. In order to unify the model settings to the utmost, nudging hasn't been applied on any component model. After all, model integration will be much slower when nudging is used. Power spectrum of the Niño3.4 index in section 4.7 of the revised manuscript shows similar variational trends for each CGCM that coincide with the reanalysis counterpart. It can thus prove that the

improvement of the cold tongue simulation is not achieved by restoring in the OGCM, otherwise the SST variability will be heavily suppressed. Although the ECHAM5-NEMO3.6 shows weak variability especially at inter-decadal scales for SAM and SOI, it is not caused by restoring in the AGCM. In fact, I don't find a way to specify the restoring in the control run configuration of the ECHAM5. Section 4.7 that introduces system variabilities of the CGCMs has been pasted below for convenience.

**'4.7 Model variability of ENSO and SAM**

In the coupled ocean-atmosphere system, global climate variability has been driven by the El Niño-Southern Oscillation (ENSO) and the southern annular mode (SAM, also called the Antarctic Oscillation) (Philander, 1990; Wallace and Thompson, 2002). It is therefore necessary to examine the model variability by applying spectra analysis on relevant indices. The CGCM simulations of the three indices are generally consistent with the theoretical red noise (Markov) spectrum (figure omitted). The Niño3.4 index is defined as the SST anomalies averaged over the NINO34 region (5°N-5°S,170°W - 120°W). It shows high variance in 2-7 years' period that documents the ENSO peaks in the HadISST reanalysis (Fig. 8a). All of the CGCMs reproduce similar variations of the Niño3.4 power spectra. The ECHAM6-NEMO3.6 presents weak variabilities at the interannual and interdecadal scale, whose periodic peaks are about one year less than the reanalysis counterpart. The ECHAM5-NEMO3.6 shows a better spectral distribution that best coincides with the reanalysis at the interannual scale. However, it still suffers a weak variability at the interdecadal scale and the periodic peak is even half a year less than that of the ECHAM6-NEMO3.6. The MPI-ESM instead takes on an intensified interannual variability, which stays strong at the interdecadal scale. The Southern Oscillation Index (SOI) is calculated based on the differences in sea level pressure anomalies between Tahiti and Darwin in Australia. In comparison to the Niño3.4 spectra, the SOI exhibits similar peaks at the interannual scale in the ERA-20c reanalysis (Fig. 8b). Nevertheless, all the CGCMs reproduce weak variabilities at the interannual and interdecadal scales. The ECHAM6-NEMO3.6 presents the best simulation with a significant increase in variance around 4 years' period, while the ECHAM5-NEMO3.6 shows the weakest variability at the interannual scale. It implies that the AGCM replacement has an opposite effect on the Niño3.4 and SOI variabilities. The MPI-ESM also show reduced variance from the annual scale and above, quite the opposite to that in the Niño3.4 case. Since the model biases of ENSO variability may be attributed to thermocline feedback and zonal wind variations (Borlace et al., 2013), the reversed changes in the variabilities of Niño3.4 and SOI can be caused by the related oceanic and atmospheric processes. The SAM index is calculated following Gong and Wang (1999) by the differences of normalized monthly zonal mean sea level pressure at 40°S and 65°S. Variations of SAM tend to be more flattened than those of SOI in the ERA-20c reanalysis (Fig. 8c), with prominent fluctuations from biannual to

interannual scales. Compared with the reanalysis counterpart, all CGCMs show more power at interannual time scales that represents a robust modulation of the SAM, which is possibly attributed to the semi-annual oscillation (SAO) (Hurrell and van Loon, 1994) and circulation anomalies over Antarctica (Thompson and Solomon, 2002). The ECHAM6-NEMO3.6 presents stronger decadal variability than that of the reanalysis data, while the ECHAM5-NEMO3.6 exhibits weaker low-frequency variability. Since the high variance in low-frequency band represents the upward trend of SAM index at decadal scale (Raphael and Holland, 2006), updating the AGCM can result in a drastic change of long-term climate variability in southern hemisphere. In contrast, the MPI-ESM shows the SAM variability very close to the reanalysis counterpart from 1 year and above, indicating that the OGCM feedback to the atmosphere can lead to a better representation of the inter-decadal variability.

[Figure]

Figure 8: Power spectra of (a) Niño3.4 index, (b) SOI, (c) SAM index. Solid line denotes the calculation results of reanalysis data, green dotted line denotes the ECHAM5-NEMO3.6 simulation, red dotted line denotes the ECHAM6-NEMO3.6 simulation, and blue dotted line denotes the MPI-ESM simulation.'

3. *The most prominent feature of no North Atlantic cold SST bias in a 2 ocean model coupled to ECHAM5 in their ECHAM5-NEMO3.6st configuration is not properly discussed. This bias has been around for decades in coupled climate models at the given resolution, and numerous papers discuss it. None of this work is mentioned or compared to. See also our detailed comments on Figure 3 below.*

Reply:

Thanks for these comments. The ECHAM5-NEMO3.6 and ECHAM6-NEMO3.6 uses the same configuration for the OGCM, and in the ECHAM6-NEMO3.6 the cold biases are still there in North Pacific and North Atlantic. Zhang & Zhao (2015) suggested that the cold SST bias in Atlantic caused the same cold bias in North Pacific through WES feedback and NAM originating in tropical and extra-tropical Atlantic. In our study, the AMOC biases are not so much different between the two CGCMs, while the NPMOC (MOC in North Pacific) exhibits bigger inter-model differences. Therefore, the experiment results suggest an inverse cause-and-effect relationship between the cold SST biases in North Pacific and North Atlantic, through the air-sea interaction. Investigation on the mechanisms will be discussed in an ongoing research. The AMOC comparison has been pasted below:

'Since the upper cell of Atlantic meridional overturning circulation (AMOC) plays a significant role in delaying warming signals from anthropogenic greenhouse gases and responding to climate change (Marshall et al., 2014; Buckley and Marshall, 2016), model bias analysis is still focused on the upper ocean levels. The overall magnitude of AMOC bias is less than that of NPMOC with significantly reduced biases near the sea surface (Fig. 10), which is consistent with those of surface currents among the three CGCMs. The ECHAM6-NEMO3.6 shows exiguous bias near the ocean surface, but presents strong biases in the mesopelagic zone of subtropical areas, bringing more heat to higher latitudes (Fig. 10a). Likewise, the ECHAM5-NEMO3.6 exhibits strong circulation biases rotating clockwise in the thermocline that intensifies poleward heat transport (Fig. 10b). In the upper ocean levels, the AMOC poleward transport is a little more enhanced than that of the ECHAM6-NEMO3.6. With similar bias patterns of the AMOC, the ECHAM5-NEMO3.6 and the ECHAM6-NEMO3.6 have opposite SST biases in North Atlantic (Figs. 3a and 3c), which implies that the air-sea feedback including WES feedback and NAM as suggested by Zhang & Zhao (2015) takes the responsibility. The MPI-ESM experiment shows negative biases in tropical Atlantic from the sea surface to the bathypelagic zone, indicating that the overturning circulation has been restrained. There is a narrow positive bias in the subtropical Atlantic, but its strength has been limited by the negative biases nearby. One consequence of the weak AMOC is the decrease of SST in North Atlantic due to less heat supply from the tropics (Fig. 3e). The overturning circulation is enhanced in the middle latitudes with one centre located north of 35ºN and another centre around 55ºN at the depth of 1200m. It still promotes the poleward heat transport and results in

warm SST biases in subpolar region (Fig. 3e). The AMOC biases in the MPI-ESM piControl experiment are similar as those in the MPI-ESM experiment, with more negative biases in tropical Atlantic. Comparing the AMOC biases between the MPI-ESM and the ECHAM5-NEMO3.6, it can be seen that the SST cold biases in North Atlantic are partially attributed to decreased MOC in the thermocline of tropical and extra-tropical oceans. However, the air-sea interaction also takes account of the SST variations in consideration of the SST differences between the ECHAM5-NEMO3.6 and the ECHAM6-NEMO3.6. Zhang & Zhao (2015) suggested that the cold SST bias in Atlantic caused the same cold bias in North Pacific through different mechanisms originating in tropical and extra-tropical Atlantic. Because the differences of NPMOC are bigger than those of AMOC between these two newly developed CGCMs, it suggests an inverse cause-and-effect relationship between the cold SST biases in North Pacific and North Atlantic where the former takes the lead.

[Figure]

**Figure 10: Model biases of AMOC in summer, (a) ECHAM6-NEMO3.6, (b) ECHAM5-NEMO3.6, (c) MPI-ESM, (d) MPI-ESM piControl, Unit: Sv.'**

4.  *No figures or information on the stability of the control simulation (e.g., timeseries of surface air temperature, TOA radiation budget, etc.) are provided which are crucial to evaluate coupled GCM performance.*

Reply:

Thanks for these comments. As in the question No.1, we have ensured that no remarkable climate drift appears in model integration with the SST time series. It

is obviously better to provide all the information, but it also increases the length of the article which is already long enough. At the very beginning, we believed that it was necessary to integrate the CGCMs for thousands of years. But later in a LASG (The State Key Laboratory of Numerical Modeling for Atmospheric Sciences and Geophysical Fluid Dynamics) annual conference, via the personal contact with some researchers in the Institute of Atmospheric Physics (IAP), we were told that 200-300-year realization was enough for the CGCM experiment. Since the long-term integration was time-consuming and less cost-effective, we followed their advice to analyze the model results after 100-year realization and hence no time series of these quantities were provided.

5. *A new coupled model system is presented and key ocean parameters such as the Atlantic Meridional Overturning circulation (MOC) or important coupled atmosphere ocean variability patterns (e.g., ENSO, NAO), their difference amongst the different GCM configurations and their possible impact on the SST bias are not discussed and should be added to the paper.*

Reply:

Thanks for these comments. The AMOC and Model variability have been analyzed and compared among the three CGCMs in section 4.7 and section 5.2 of the revised manuscript.

**'4.7 Model variability of ENSO and SAM**

In the coupled ocean-atmosphere system, global climate variability has been driven by the El Niño-Southern Oscillation (ENSO), the southern annular mode (SAM, also called the Antarctic Oscillation) and the Indian Ocean dipole (IOD) (Philander, 1990; Wallace and Thompson, 2002; Saji et al., 1999). It is therefore necessary to examine the model variability by applying spectra analysis on relevant indices. The CGCM simulations of the three indices are generally consistent with the theoretical red noise (Markov) spectrum (figure omitted). The Niño3.4 index is defined as the SST anomalies averaged over the NINO34 region (5°N-5°S,170°W -120°W). It shows high variance in 2-7 years' period that documents the ENSO peaks in the HadISST reanalysis (Fig. 8a). All of the CGCMs reproduce similar variations of the Niño3.4 power spectra. The ECHAM6-NEMO3.6 presents weak variabilities at the interannual and interdecadal scale, whose periodic peaks are about one year less than the reanalysis counterpart. The ECHAM5-NEMO3.6 shows a better spectral distribution that best coincides with the reanalysis at the interannual scale. However, it still suffers a weak variability at the interdecadal scale and the periodic peak is even half a year less than that of the ECHAM6-NEMO3.6. The MPI-ESM instead takes on an intensified interannual variability, which stays strong at the interdecadal scale. The Southern Oscillation Index (SOI) is calculated based on the differences in sea level

pressure anomalies between Tahiti and Darwin in Australia. In comparison to the Niño3.4 spectra, the SOI exhibits similar peaks at the interannual scale in the ERA-20c reanalysis (Fig. 8b). Nevertheless, all the CGCMs reproduce weak variabilities at the interannual and interdecadal scales. The ECHAM6-NEMO3.6 presents the best simulation with a significant increase in variance around 4 years' period, while the ECHAM5-NEMO3.6 shows the weakest variability at the interannual scale. It implies that the AGCM replacement has an opposite effect on the Niño3.4 and SOI variabilities. The MPI-ESM also show reduced variance from the annual scale and above, quite the opposite to that in the Niño3.4 case. Since the model biases of ENSO variability may be attributed to thermocline feedback and zonal wind variations (Borlace et al., 2013), the reversed changes in the variabilities of Niño3.4 and SOI can be caused by the related oceanic and atmospheric processes. The SAM index is calculated following Gong and Wang (1999) by the differences of normalized monthly zonal mean sea level pressure at 40°S and 65°S. Variations of SAM tend to be more flattened than those of SOI in the ERA-20c reanalysis (Fig. 8c), with prominent fluctuations from biannual to interannual scales. Compared with the reanalysis counterpart, all CGCMs show more power at interannual time scales that represents a robust modulation of the SAM, which is possibly attributed to the semi-annual oscillation (SAO) (Hurrell and van Loon, 1994) and circulation anomalies over Antarctica (Thompson and Solomon, 2002). The ECHAM6-NEMO3.6 presents stronger decadal variability than that of the reanalysis data, while the ECHAM5-NEMO3.6 exhibits weaker low-frequency variability. Since the high variance in low-frequency band represents the upward trend of SAM index at decadal scale (Raphael and Holland, 2006), updating the AGCM can result in a drastic change of long-term climate variability in southern hemisphere. In contrast, the MPI-ESM shows the SAM variability very close to the reanalysis counterpart from 1 year and above, indicating that the OGCM feedback to the atmosphere can lead to a better representation of the inter-decadal variability.

[Figure]

Figure 8: Power spectra of (a) Niño3.4 index, (b) SOI, (c) SAM index. Solid line denotes the calculation results of reanalysis data, green dotted line denotes the ECHAM5-NEMO3.6 simulation, red dotted line denotes the ECHAM6-NEMO3.6 simulation, and blue dotted line denotes the MPI-ESM simulation.

Since the upper cell of Atlantic meridional overturning circulation (AMOC) plays a significant role in delaying warming signals from anthropogenic greenhouse gases and responding to climate change (Marshall et al., 2014; Buckley and Marshall, 2016), model bias analysis is still focused on the upper ocean levels. The overall magnitude of AMOC bias is less than that of NPMOC with significantly reduced biases near the sea surface (Fig. 10), which is consistent with those of surface currents among the three CGCMs. The ECHAM6-NEMO3.6 shows exiguous bias near the ocean surface, but presents strong biases in the mesopelagic zone of subtropical areas, bringing more heat to higher latitudes (Fig. 10a). Likewise, the ECHAM5-NEMO3.6 exhibits strong circulation biases rotating clockwise in the thermocline that intensifies poleward heat transport (Fig. 10b). In the upper ocean levels, the AMOC poleward transport is a little more enhanced than that of the ECHAM6-NEMO3.6. With similar bias patterns of the AMOC, the ECHAM5-NEMO3.6 and the ECHAM6-NEMO3.6 have opposite SST biases in North Atlantic (Figs. 3a

and 3c), which implies that the air-sea feedback including WES feedback and NAM as suggested by Zhang & Zhao (2015) takes the responsibility. The MPI-ESM experiment shows negative biases in tropical Atlantic from the sea surface to the bathypelagic zone, indicating that the overturning circulation has been restrained. There is a narrow positive bias in the subtropical Atlantic, but its strength has been limited by the negative biases nearby. One consequence of the weak AMOC is the decrease of SST in North Atlantic due to less heat supply from the tropics (Fig. 3e).  The overturning circulation is enhanced in the middle latitudes with one centre located north of 35ºN and another centre around 55ºN at the depth of 1200m. It still promotes the poleward heat transport and results in warm SST biases in subpolar region (Fig. 3e). The AMOC biases in the MPI-ESM piControl experiment are similar as those in the MPI-ESM experiment, with more negative biases in tropical Atlantic. Comparing the AMOC biases between the MPI-ESM and the ECHAM5-NEMO3.6, it can be seen that the SST cold biases in North Atlantic are partially attributed to decreased MOC in the thermocline of tropical and extra-tropical oceans. However, the air-sea interaction also takes account of the SST variations in consideration of the SST differences between the ECHAM5-NEMO3.6 and the ECHAM6-NEMO3.6. Zhang & Zhao (2015) suggested that the cold SST bias in Atlantic caused the same cold bias in North Pacific through different mechanisms originating in tropical and extra-tropical Atlantic. Because the differences of NPMOC are bigger than those of AMOC between these two newly developed CGCMs, it suggests an inverse cause-and-effect relationship between the cold SST biases in North Pacific and North Atlantic where the former takes the lead.

[Figure]

**Figure 10: Model biases of the AMOC in summer, (a) ECHAM6-NEMO3.6, (b) ECHAM5-NEMO3.6, (c) MPI-ESM, (d) MPI-ESM piControl, Unit: Sv.**

,

6.  *In our opinion, the pattern correlation method (table 1, with pattern correlations always below 0.4) cannot be used to explain the inter-model differences as it completely ignores both the physical dependencies of the parameters used in the correlation as well as the impact of ocean dynamics and coupled ocean-atmosphere feedbacks onto the SST bias in a GCM. In addition, the presented pattern correlation values are very low.*

Reply:

Thanks for these comments. Since one main purpose of this paper is to investigate the effects of changing component models on the coupled system, which inevitably involves inter-model comparison with different parameterization schemes and even dynamical structures. Although this is less rigorous than normal approaches to study the model characteristics, we take a bold step forward to study the model response with different configurations so as to overcome the

predicament of model development that tends to improve the simulation quality by blindly updating the parameterization schemes. The robustness of pattern correlation has been increased when the area of computation is narrowed down to Pacific (Tab. 2 in the revised manuscript). Low correlation values seem less convincing, but the top three ranking variables that facilitate the attribution analysis have well passed the 99.9% Student-t test. A larger number of grid points are involved that makes the threshold value relatively small. The ranking of coupling variables provides an insight into the causation of inter-model differences, which is not used in its absolute sense.

7. *Unfortunately, no information about the setup of the land component in the new coupled GCM is provided. We assume it is using JSBACH, the new land model component within ECHAM6. Is JSBACH running interactively? Why are the pattern correlations for albedo that weak? The interpretation of simulated precipitation is questionable, as differences in the extra tropics are not really visible (scale inappropriate). Additionally, the paper lacks also information about 2m temperatures (also referred to as SAT – surface air temperature) simulated over land.*

Reply:

Thank you for these comments. The land component of ECHAM6 is JSBACH, which is used with the default configuration as that of piControl run for both MPI-ESM and ECHAM6-NEMO3.6 experiments. Therefore, it should be running concurrently with the atmospheric core. The pattern correlations for albedo are calculated between the model differences of the SST and albedo, which includes the contribution of OGCMs with different model structures and parameterizations. Although low correlation values seem inconsistent with some studies, the model results are still within tolerance.

Precipitation biases are plotted with the same scales used in Huang et al. (2014), which basically emphasizes on tropical variations. In the paper, precipitation bias is just mentioned as one aspect of model evaluation. Extra-tropical biases are not closely associated with the qualitative reasoning part of the paper, which have thus been neglected.

The model SAT biases against the ERA-Interim reanalysis have been attached below. Large biases in polar areas may be attributed to model deficiencies and uncertainties in the reanalysis data.

[Figure]

**Biases of the SAT simulation in summer (left column) and winter (right column) corresponding to each CGCM: (a, b) ECHAM6-NEMO3.6, (c, d) ECHAM5-NEMO3.6, (e, f) MPI-ESM.**

8. *No information on sea ice in the different GCM configuration is provided. To be able to judge the SST differences between the different model configurations properly, some information such as sea ice extent and sea ice thickness should be added to the paper.*

Reply:

Thanks for these comments. We have added the sea-ice model description and configuration in section 2.1 and section 2.4 of the revised manuscript.

'…The Louvain-la-Neuve sea-ice model (LIM3), originally developed by Fichefet and Morales-Maqueda (1997), has been incorporated in NEMO3.6 to represent the sub-grid-scale dynamics and their impact on sea ice thickness and ice-ocean salt exchanges. Main differences between LIM3 and other ice models are related to the physical parameterization of open boundary conditions and sea-ice

interactions, with the C-grid formulation of elastic-viscous-plastic rheology (Bouillon et al., 2013).

…The sea ice model (LIM3) in NEMO3.6 is configured to compute the ice-ocean fluxes under the influence of air-sea fluxes, ocean mass and salt exchanges, with light penetration of solar radiation. Ice freezing and melting also affects the albedo in the Arctic and Antarctic regions. Likewise, the sea ice thickness and density in the MPIOM respond to wind stress and ocean currents without consideration of turning angles. Surface heat balance and the internal ice stress also affect the variations of sea ice cover with zero-layer formulation of Semtner (1976).'

9. *Some of the presented model configurations (ECHAM5 coupled to NEMO) have been developed almost 10 years ago (Park et al., 2009), and have been used extensively during the last 10 years including work on the SST bias (Wahl et al., 2009, Harlass et al., 2015). Unfortunately, none of this work is mentioned in the introduction or in the discussion.*

Reply:

Thanks for these comments. We have implicitly mentioned the some of these previous studies in the OGCM introduction part in section 2.1.1.

'Designed to serve as a flexible tool for ocean and sea ice studies, NEMO manifests good usability interacting with other ACGMs (Gualdi et al., 2003; Luo et al., 2005; Park et al., 2009; Dunlap et al., 2014; Huang et al., 2014).'

These citations have been added to the introduction part as you suggest. However, no more discussion on these previous studies is supplemented because other similar studies have been introduced.

**References**

Borlace, S., Cai, W., Santoso, A.: Multidecadal ENSO amplitude variability in a 1000-yr simulation of a coupled global climate model: implications for observed ENSO variability, J. Climate, 26, 9399–9407, 2013.

Gong, D. and Wang, S.: Definition of Antarctic oscillation index, Geophys. Res. Lett., 26, 459–462, doi:10.1029/1999GL900003, 1999.

Gupta, A.S., L.C. Muir, J.N. Brown, S.J. Phipps, P.J. Durack, D. Monselesan, and S.E. Wijffels, 2012: Climate Drift in the CMIP3 Models. *J. Climate,* **25**, 4621–4640,https://doi.org/10.1175/JCLI-D-11-00312.1

Huang P, Wang P F, Hu K M, Huang G, Zhang Z H, Liu Y, Yan B L. 2014. An Introduction to the Integrated Climate Model of the Center for Monsoon System Research and Its Simulated Influence of El Nino on East Asian–Western North Pacific Climate[J]. Advances in Atmospheric Sciences, 31: 1136–1146.

Hurrell, J. W., and Van Loon, H.: A modulation of the atmospheric annual cycle in the Southern Hemi-sphere, Tellus, 46A, 325–338, 1994.

Philander, S. G.: El Niño, La Niña, and the Southern Oscillation, Academic Press, 289, 1990.

Raphael, M. and Holland, M. M.: Twentieth Century Simulation of the Southern Hemisphere in Coupled Models. Part I: Large scale Circulation Variability, Clim. Dynam., 26, 217-228, 2006.

Semtner, A. J., 1976. A model for the thermodynamic growth of sea ice in numerical investigations of climate. J. Phys. Oceanogr., 6, 379-389.

Thompson, D. W. J., Solomon, S.: Interpretation of recent Southern Hemisphere climate change, Science, 296, 895-899, 2002.

Wallace, J. M., and Thompson D. W. J.: The Pacific center of action of the Northern Hemisphere annular mode: Real or artefact?, J. Climate, 15, 1987-1991, 2002.

Zhang, L. and Zhao, C.: Processes and mechanisms for the model SST biases in the North Atlantic and North Pacific: A link with the Atlantic meridional overturning circulation, J. Adv. Model. Earth Sy., 7, 739-758, doi:10.1002/2014MS000415, 2015.

---

## Author Comment (AC9) · 3 Oct 2018

Reply to detailed major comments

1. *Page 5, line 31: The model experiments are performed using the piControl standard scenario of the MPI-ESM (p. 5, line 31). Most of the observational or reanalysis products used to compute model biases cover more recent periods than pre-industrial (1850 or 1870). Which reference period is used to compare the model runs to?*

Reply:

Thanks for these comments. As introduced in section 2.4, the reference period is 1981~2010, the same time period of the reanalysis data. More information about the scenario setting has been added to the paper.

'The CO2 value is set to default 353.9 ppm in the user manual. Other greenhouse gases like NO2 also follows the default present time setting so that they are consistent with each other. The aerosol settings use the climatology compiled by S. Kinne without any complementation of volcanic aerosols ….'

2. *Page 6, line 1: "Model initialization is started from the climatology basic state recalculated with the AMIP run input data from 1981 to 2010." This suggests that the initial conditions in the atmosphere are based on a climatology calculated from AMIP simulations. Due to the chaotic nature of the atmospheric circulation, the choice of initial conditions of the atmosphere are not crucial for the performance of a coupled GCM. Hence we would strongly suggest to provide more information on the ocean initial conditions (see also our general comment above).*

Reply:

Thanks for these comments. As in the response to major concern No.1, the World Ocean Atlas (WOA) data has been used in the OGCM initialization. We have added some details in the ocean model initialization.

'The NEMO3.6 is initialized with temperature and salinity climatology from World Ocean Atlas (WOA) data, applying the geothermal heating at ocean bottom. The RGB formulation (Lengaigne et al., 2007) has been chosen to calculate the light penetration over the sea surface with observed time varying chlorophyll.'

3. *Figure 3c/d clearly shows the absence of a North Atlantic (NA) cold bias in a 2 ℃ ocean model coupled to a coarse resolution atmosphere (ECHAM5/6-NEMO3.6st configuration) which is a very striking result. The NA cold bias has been around for decades in coupled climate models at the given resolution, and numerous papers*

*discuss it (e.g. Zhang and Zhao, 2015). None of this work is mentioned or compared to.*

Reply:

Previous studies have been introduced in the introduction part, including Zhang and Zhao (2015) in the original manuscript. In the revised manuscript, more comparison has been added in section 5.1 about the AMOC inter-model comparison. The following picture is a snapshot of the original manuscript where Zhang and Zhao (2015) is mentioned.

[Figure]

[Figure]

amplify trade wind biases in general circulation models (GCMs) (Li and Xie, 2014), which is also responsible for excessive cold tongue simulation in equatorial Pacific. Besides traditional understanding of coupling processes that account for SST and precipitation biases, recent studies have revealed other factors in the air-sea interaction that significantly contribute to the model bias pattern. Burls et al. (2017) find a quadratic relationship between extra-tropical Pacific albedo and equatorial SST

5    bias, and Pham et al. (2017) suggested that the deep cycle of cold tongue turbulence can be affected by cloud cover and rain. The air-sea kinetic energy budget is found to be linked with surface gravity waves, where wave age and friction velocity affect in the ratio between kinetic energy from the winds and underlying surface currents (Fan and Hwang, 2017).

       The dynamic mechanisms that play a major role in the biases propagation in the CGCMs are also investigated from a wide range of perspectives. Double ITCZ precipitation problem is usually associated with biases in radiation budget and

10   surface winds (Lin, 2007). There is a close relationship between clouds and SST variation (Klein & Hartmann, 1993; Norris & Leovy, 1994), and changes of low clouds and shortwave radiation flux react on the SST and sea level pressure (SLP) (Norris et al., 1998; Mochizuki & Awaji, 2008; Bond & Cronin, 2008). Wu and Kinter III (2010) suggested that high-frequency changes in atmospheric circulation affects largely on the surface shortwave radiation, and hence its correlation with SST variability in the mid-latitude North Pacific. The weak simulation of Atlantic meridional overturning circulation (AMOC) is

15   found to be responsible for sea surface temperature (SST) cold biases in the northern hemisphere in 22 CMIP5 climate models (Wang et al., 2014), which poses a great impact on the North Pacific through Northern Hemisphere annular mode (NAM) and wind-evaporation-SST (WES) feedback, combining SST biases in extratropical North Atlantic (ENA) and tropical North Atlantic respectively (Zhang & Zhao, 2015). The temperature bias can also be attributed to underestimation of water vapor

No more comparison about the North Atlantic SST has been made in the original manuscript because the main purpose of this paper is to introduce the new CGCMs and the mechanisms behind their inter-model differences specifically in North Pacific. In the revised manuscript, the AMOC has also been analyzed in section 5.1. In section 6.1, the inter-model differences in North Atlantic SST also show that the AGCM replacement changes radiative forcing that affects surface heating. With more heat supply in subtropical and extra-tropical Atlantic, the MOC transports more heat to higher latitudes which ameliorates cold SST biases in North Atlantic in the ECHAM5-NEMO3.6 experiment. Roles of other teleconnections such as NAM

through air-sea feedback is left for a future research. Relevant content has been pasted below.

'Since the upper cell of Atlantic meridional overturning circulation (AMOC) plays a significant role in delaying warming signals from anthropogenic greenhouse gases and responding to climate change (Marshall et al., 2014; Buckley and Marshall, 2016), model bias analysis is still focused on the upper ocean levels. The overall magnitude of AMOC bias is less than that of NPMOC with significantly reduced biases near the sea surface (Fig. 10), which is consistent with those of surface currents among the three CGCMs. The ECHAM6-NEMO3.6 shows exiguous bias near the ocean surface, but presents strong biases in the mesopelagic zone of subtropical areas, bringing more heat to higher latitudes (Fig. 10a). Likewise, the ECHAM5-NEMO3.6 exhibits strong circulation biases rotating clockwise in the thermocline that intensifies poleward heat transport (Fig. 10b). In the upper ocean levels, the AMOC poleward transport is a little more enhanced than that of the ECHAM6-NEMO3.6. With similar bias patterns of the AMOC, the ECHAM5-NEMO3.6 and the ECHAM6-NEMO3.6 have opposite SST biases in North Atlantic (Figs. 3a and 3c), which implies that the air-sea feedback including WES feedback and NAM as suggested by Zhang & Zhao (2015) takes the responsibility. The MPI-ESM experiment shows negative biases in tropical Atlantic from the sea surface to the bathypelagic zone, indicating that the overturning circulation has been restrained. There is a narrow positive bias in the subtropical Atlantic, but its strength has been limited by the negative biases nearby. One consequence of the weak AMOC is the decrease of SST in North Atlantic due to less heat supply from the tropics (Fig. 3e). The overturning circulation is enhanced in the middle latitudes with one centre located north of 35ºN and another centre around 55ºN at the depth of 1200m. It still promotes the poleward heat transport and results in warm SST biases in subpolar region (Fig. 3e). The AMOC biases in the MPI-ESM piControl experiment are similar as those in the MPI-ESM experiment, with more negative biases in tropical Atlantic. Comparing the AMOC biases between the MPI-ESM and the ECHAM5-NEMO3.6, it can be seen that the SST cold biases in North Atlantic are partially attributed to decreased MOC in the thermocline of tropical and extra-tropical oceans. However, the air-sea interaction also takes account of the SST variations in consideration of the SST differences between the ECHAM5-NEMO3.6 and the ECHAM6-NEMO3.6. Zhang & Zhao (2015) suggested that the cold SST bias in Atlantic caused the same cold bias in North Pacific through different mechanisms originating in tropical and extra-tropical Atlantic. Because the differences of NPMOC are bigger than those of AMOC between these two newly developed CGCMs, it suggests an inverse cause-and-effect relationship between the cold SST biases in North Pacific and North Atlantic where the former takes the lead.'

[Figure]

**Figure 10: Model biases of the AMOC in summer, (a) ECHAM6-NEMO3.6, (b) ECHAM5-NEMO3.6, (c) MPI-ESM, (d) MPI-ESM piControl, Unit: Sv.**

'…On the contrary, the AMOC enhancement is less significant. However, with increased surface heating in subtropical and extra-tropical North Atlantic, the MOC transports more heat to higher latitudes which ameliorates cold SST biases in North Atlantic. Roles of other teleconnections between North Pacific and North Atlantic suggested by Zhang & Zhao (2015), such as NAM through air-sea feedback, is left for a future research.'

4. *Figure 3: The authors note that the SST bias is largest in the polar regions exceeding 4degC as shown on Figure 3. This large bias clearly coincides with sea-ice coverage. There, the HadISST data set, which is used as a reference here, provides temperatures near the freezing point of sea water ( -1.9degC). While the authors are right that HadISST and other reanalysis products have deficiencies in high latitudes due to the lack of observations, it is quite astonishing how the model bias can exceed 4degC where the sea water should be at or near the freezing point. Large biases can be expected at the sea-ice edge, which position may quite differ among coupled models.*

Reply:

We have compared our model results with those of Huang et al. (2014), which yields the same large bias in polar areas. Their figure has been pasted below:

[Figure]

**Fig. 3.** Distribution of the (a, b) modeled and (c, d) observed SST (°C) in JJA and DJF for the periods as defined in section 2, and (e, f) their differences.

From the above figure, it can be seen that the HadISST possesses much lower SST in polar areas, far below -4°C, which is not the same as "-1.9 degC" in this major comment. It also proves that large biases in polar areas are not my fault.

5. *Page 9, section 4.4 on ocean currents: The section on ocean currents is confusing as the differences in ocean currents are not related to the underlying ocean currents. When discussing differences in ocean currents please term the ocean currents that are enhanced/weakened, for example "enhanced/weakened Kuroshio transport is present in Model A compared to observations."*

Reply:

Thank you for these comments. We have revised the manuscript as you suggest. Major modifications made in the section have been pasted below.

'The ocean current biases are mainly located in tropical areas (Fig. 6), in predominantly zonal directions for ECHAM5-NEMO3.6 and ECHAM6-NEMO3.6, but meridionally distributed for MPI-ESM. South equatorial currents and equatorial counter currents are enhanced in ECHAM5-NEMO3.6, with more anomalous currents than those in ECHAM6-NEMO3.6 (Fig. 6a, c). Whereas the MPI-ESM features southward (northward) tilted biases south (north) of the equator to a larger degree than the other two CGCMs, which enhance the Kuroshio current, East Australian current, Brazil current and Mozambique current but weaken the Peru current, California current, Benguela current and West Australian current (Fig. 6e). The direction of ocean current biases generally agrees with that of wind stress biases, where poleward deflection can be attributed to Coriolis effects. Since the poleward motion is too strong in the MPI-ESM experiment, the ocean currents in subtropical North Pacific even turn to the east. The Kuroshio transport is enhanced for both MPI-ESM and ECHAM5-NEMO3.6, favouring more heat transport from subtropics to higher latitudes. Yet colder SST biases still exist over large maritime space in MPI-ESM experiment, which suggests an investigation on radiation budget and the meridional overturning currents that provide a full picture of most relevant oceanic processes. The SST biases are also attributed to different projection grids of NEMO3.6 and MPIOM, which however is beyond the scope of this paper. Biases in winter season are diminished to some degrees in tropical oceans, with little amplification of biases outside tropics in ECHAM5-NEMO3.6 and ECHAM6-NEMO3.6 experiments (Figs. 6b, d). But the MPI-ESM comes up with significantly enhanced Kuroshio transport in subtropical North Pacific (Fig. 6f), which may help to explain warm SST bias around Sea of Japan (Fig. 3f) and cold SST bias in the subtropical ocean.'

6. *Page 11, lines 11-15: In the first sentence you claim that the differences in NPMOC in the two MPI-ESM model simulations are not caused by the increased coupling frequency while in the second sentence you argue that "suggesting obvious improvements after decreasing coupling interval" are present. Please clarify. Additionally please provide more information on the MPI-ESM model data (e.g. MPI-ESM model version and a reference paper) cited as "The piControl experiment result (available in http://esgfnode. llnl.gov/)". To our knowledge, the publicly available MPI-ESM output available at http://esgf-node.llnl.gov is based on the CMIP5 version of MPI-ESM which implements older versions of both ECHAM6 and MPIOM. It means that the two models differ by far more than just the coupling frequency.*

Reply:

I tried to say that more MOC biases in the MPI-ESM experiment than those in the ECHAM5-NEMO3.6 and ECHAM6-NEMO3.6 are not caused by increasing coupling frequency. In fact, it helps to improve the simulation quality, which can be seen from the comparison with piControl data. This sentence has been rewritten to avoid ambiguity.

'The piControl experiment result (available in http://esgf-node.llnl.gov/) is attached (Fig. 9d) to demonstrate that the prominent biases in MPI-ESM than those in the NEMO3.6 coupled experiments are not caused by increasing coupling frequency from one day to 4 hours. The piControl run data used in this study has the same time span as that of the reanalysis data from the year 1981 to 2010, which should be able to represent the model abilities in reproducing the climatology of the same period.'

There are indeed many differences between the MPI-ESM AGCM and the counterpart in CMIP5 piControl, due to model updates of all kinds. By using the piControl data for comparison, I just try to prove that the model simulation is not degraded by changing coupling frequency, which can be used for inter-model comparison with other CGCMs. I have revised the manuscript to convey this idea more clearly.

'More contour lines appear in NPMOC biases of piControl than the MPI-ESM experiment conducted in this paper (Fig. 9c, d). Although the MPI-ESM in the CMIP5 piControl experiment is an older version, it can at least demonstrate that increasing coupling frequency from one day to 4 hours does not degrade the MPI-ESM simulation. Previous studies also suggest that model simulation can be improved by decreasing coupling interval (Bernie et al., 2008; Ge et al., 2017). Hence, the MPI-ESM experiment result and the inter-model comparison with other CGCMs are trustworthy.'

7. page 12, line 15: "The analysis on oceanic and atmospheric circulation has made it clear that the SST bias is consistent with meridional overturning circulation in North Pacific, driven by surface wind stress anomalies that are maintained by anomalous Walker circulation over the tropical Pacific. Cumulus convection process is found to be a major contributor to inter-model differences". The authors should explain in more detail why they assume that cumulus convection is the key in the chain of arguments provided.

Reply:

Thanks for these comments. We have added a more detailed explanation.

'Since cumulus convection modulates changes in temperature, specific humidity and atmospheric circulation, it is most likely to be the predominant factor that shapes the inter-model differences.'

8. *page 13, line 11: This sentence is confusing. Your statement that enhanced northerly winds (which we cannot find on Figure 10b as you indicate in the text) in ECHAM6-NEMO3.6 compared to ECHAM5-NEMO3.6 in a region dominated by easterly trade winds are responsible for stronger evaporative cooling of SSTs in ECHAM6-NEMO3.6 compared to ECHAM5-NEMO3.6 is unclear. Please clarify.*

Reply:

Thank you for pointing out the problem. The "northerly winds" in the text should be changed into "southerly winds". We are sorry for the mistake. In our opinion, the easterly winds superimposed by southerly anomalies result in a bigger wind speed that helps to increase surface evaporation. Evaporative cooling in the region decreases the SST. We have revised the statement to avoid ambiguity.

'Deviations of 10m wind exhibit southerly anomalies around the central tropical Pacific in southern hemisphere (Fig. 12b), where easterly winds prevail for summer climatology (Fig. 10a). The easterly winds superimposed by southerly anomalies result in a bigger wind speed that helps to increase surface evaporation. Hence the latent heat absorption over the sea surface are enhanced that makes SST deviation colder than 1℃ (Fig. 12d).'

9. *page 13, line 18: "Since the latent heat and surface wind differences are caused by replacing the AGCM,..." The statement is challenging, as for example the latent heat flux between ocean and atmosphere is a coupled process (see also 12. below).*

Reply:

Thank you for giving us the advice. We have corrected the statement.

'Since the latent heat and surface wind differences are caused by replacing the AGCM and the associated air-sea feedback, it is advisable to compare deviations in vertical circulation that may shed some light on corresponding physical processes.'

10. *Page 13, line 27: "It turns out that deviations in shortwave flux and latent heat are more significant than those in longwave and sensible heat fluxes." Additional information on the physics behind this statement would be helpful. In its current form it completely ignores the fact that there are large differences in the regional importance of the different fluxes.*

Reply:

Thank you for pointing out the problem. We have modified the statement as follows:

'From the general view of ocean surface energy balance, the amount of incoming and outgoing energy should be equal. Variations of shortwave flux and latent heat are more significant than those of longwave and sensible heat fluxes after replacing the AGCM.'

11. *Page 14 first lines: "...can be confirmed that the AGCM replacement first alters cumulus convection that modulates temperature, specific humidity and atmospheric circulation, which in turn accommodates cloud radiation feedback to a consistent change and affects the radiation budget". Some references that underpin the postulated process chain should be added.*

Reply:

Thank you for your suggestion. We have added citations of related publications at the end of the sentence.

'…which in turn accommodates cloud radiation feedback and changes the radiation budget (Xu and Randall, 1995; Stephens et al., 2008; Ghate et al., 2015).'

12. *Page 16, line 8: "Through analysis on circulation and radiation terms, it has been clear that latent heat of evaporation plays a predominant role in the SST differences after changing the OGCM." This statement does not take into account that LH flux is a coupled process. LH heat flux may impact SSTs in the tropics where atmospheric temperature is high and hence strong evaporation is possible, but is mainly dominated by stability of the atmospheric stratification, windspeed, moisture and temperature in the lowest atmospheric level. It's not as simple as the sentence suggests.*

Reply:

Thank you for pointing out the problem. We have addressed the problem in the response to one anonymous referee. Since the OGCM replacement changes the ocean dynamics simulation, which changes the ocean surface properties and the air-sea feedback. The atmospheric model responds to this perturbation during coupling with the OGCM. The atmosphere and ocean systems finally reach a quasi-equilibrium that exhibit variations in the SST. Although the latent heat flux holds the biggest correlation coefficient and the most obvious deviation pattern among the four radiation terms, it is more likely a manifestation of the large-scale ocean dynamical effect on the inter-model differences as suggest by Ying and Huang (2016). Major changes regarding this issue are pasted below:

'Since the top 3 variables that are most relevant to SST deviations (Tab. 1) suggest changes in the surface heat budget, involving the momentum and temperature

exchanges between ocean and atmosphere, it implies that the joint effects of atmosphere and ocean models lead to the model deviation patterns (Fig. 15). Simulation of ocean dynamics is different after replacing the OGCM. With changes in ocean advection, the SST and surface currents are altered which modulates the surface evaporation, convection and heat conduction. Subsequently the latent heat and sensible heat fluxes vary over the sea surface. The thermal and moisture perturbations from the ocean are passed to the atmosphere during coupling processes (Fig. 15c, d). Variations of low-level atmospheric circulation and humidity take effects on cloud formation and cloud liquid water path that changes precipitation and cloud radiative forcing. The net shortwave and longwave radiations are influenced and make a difference to the atmospheric circulation (Fig. 12c) and the heat budget over the sea surface (Fig. 15 a, b). Then, the perturbation signal is transferred back to the ocean that changes the SST and surface currents. This air-sea feedback finally reaches a quasi-equilibrium with marked SST warming over vast maritime spaces over the globe. The associated physical processes represented by each radiation term in Figure 15 are in accordance with their signs of pattern correlation (Tab. 1). It seems to suggest that surface evaporation plays a predominant role in the SST differences, because the latent heat flux holds the biggest correlation coefficient and the most obvious deviations among the four radiation terms. However, it is more likely a manifestation of the large-scale ocean dynamical effect on the inter-model differences as suggest by Ying and Huang (2016).'

13. *Page 16, line 10: What are "conduction processes" and how do they affect sensible heat flux? Additionally, this sentence indicates that the authors don't take into account that e.g. sensible heat flux is a coupled ocean-atmosphere process that mainly depends on the ocean atmosphere temperature difference. The importance of sensible heat flux for SST depends on the region. The authors should also provide a reference if and to what extend the surface flux parameterizations have changed in ECHAM6 with respect to ECHAM5.*

Reply:

Thanks for these comments. "Conduction processes" refer to heat conduction over the sea surface. Sensible heat flux can be affected by heat conduction and convection near the sea surface, which indeed depends on the air-sea temperature differences. We have modified the sentence as you suggest. This statement is for the OGCM replacement case, in which the AGCM is only ECHAM6. Therefore, the citations of surface flux parameterizations are provided, but changes in surface flux parameterizations are not mentioned.

'For the OGCM replacement, the top 3 variables are latent heat flux, sensible heat flux and net longwave flux. Through analysis on circulation and radiation terms, it has been clear that the ocean dynamical effect plays a predominant role in the SST differences after changing the OGCM. The ocean advection initiates the perturbations of SST and surface evaporation that modulate the atmospheric humidity and low-level circulation. Consequently, the cloud masking effect on radiative fluxes are altered which influences the atmospheric circulation and surface heat budget. The resulting equilibrium of the air-sea feedback manifest itself as the inter-model differences in the related meteorological fields. With the same surface flux parameterizations in the AGCM (Stevens et al., 2013), latent heat flux holds the largest correlation coefficient and exhibits the most prominent variations at the global scale. Since the latent heat differences are resulted from the OGCM replacement, it changes the SST and surface evaporation that also contribute to differences in the near-surface atmospheric humidity. Cao et al. (2015) point out that diversity of simulated SST and near-surface atmospheric specific humidity lead to the most diverse variability of latent heat flux over Pacific in CMIP5 models, which coincides with our research finding.'

14. *page 16: line 11: "Attributing simulation deviations to latent heat in the OGCM case is consistent with Cao et al. (2015), which points out that amplitude and meridional variability of latent heat flux over Pacific are the most diverse in CMIP5 models." From our understanding, Cao et al. summarize that LH flux is very diverse amongst coupled models due to the large differences in simulated SST but not that the LH flux differences between the models can explain the bias (From the abstract of Cao et al., 2015: "Regression analysis indicates that the inter-model diversity [in LH flux] may come from the diversity of simulated SST and near-surface atmospheric specific humidity").*

Reply:

> Thank for your comments. As we have explained in the previous question, the latent heat differences are resulted from the OGCM replacement, which changes the SST and surface evaporation. It also contributes to differences in the near-surface atmospheric specific humidity. We have modified the statement to avoid ambiguity.

'With the same surface flux parameterizations in the AGCM (Stevens et al., 2013), latent heat flux holds the largest correlation coefficient and exhibits the most prominent variations at the global scale. Since the latent heat differences are resulted from the OGCM replacement, it changes the SST and surface evaporation that also contribute to differences in the near-surface atmospheric humidity. Cao et al. (2015) point out that diversity of simulated SST and near-surface atmospheric specific humidity lead to the

most diverse variability of latent heat flux over Pacific in CMIP5 models, which coincides with our research finding.'

15. *Page 16, line 14: Please explain what the term "reverse transformation of model bias" means.*

Reply:

We would like to say the inverse variations of model SST by replacing the AGCM and OGCM. The sentence has been rewritten.

'…suggesting that inverse variations of model SST bias can be realized through ….'

16. *Page 16, line 25: More details on the "coupling processes" (line 26) that you claim to be responsible for the differences in the wind field in the different GCMs should be added.*

Reply:

Thank you for your advice. We have briefly included the physical processes that are responsible for less wind biases in zonal direction.

'… which suggests that OGCM replacement can also diminish this bias through coupling processes involving evaporative cooling and cloud radiative feedback. The associated thermal forcing drives the atmospheric circulation and changes the ocean surface wind biases. It is easy to see that an anomalous Walker circulation rotating clockwise appears over the tropical Pacific, when the surface heating is colder in the east and warmer in the west (Fig. 13b and Fig. 15).'

17. *Page 16, line 28: Please explain what you mean by "net surface radiation is warmer in the east and colder in the west."*

Reply:

We would like to say that the surface heating is enhanced over the tropical eastern Pacific but is weakened in the tropical western Pacific. We have revised the statement with proper expressions.

'It is easy to see that an anomalous Walker circulation rotating clockwise appears over the tropical Pacific, when the surface heating is weakened in the tropical eastern Pacific and is enhanced in the tropical western Pacific'

**References**

Ghate, V.P., M.A. Miller, B.A. Albrecht, and C.W. Fairall, 2015: Thermodynamic and Radiative Structure of Stratocumulus-Topped Boundary Layers. J. Atmos. Sci.,72, 430–451, https://doi.org/10.1175/JAS-D-13-0313.1

Huang, P., Wang, P.F., Hu, K.M., Huang, G., Zhang, Z.H., Liu, Y., and Yan, B.L.: An Introduction to the Integrated Climate Model of the Center for Monsoon System Research and Its Simulated Influence of El Nino on East Asian–Western North Pacific Climate, Adv. Atmos. Sci., 31, 1136–1146, 2014.

Stephens, G.L., S. van den Heever, and L. Pakula, 2008: Radiative–Convective Feedbacks in Idealized States of Radiative–Convective Equilibrium. J. Atmos. Sci.,65, 3899–3916, https://doi.org/10.1175/2008JAS2524.1

Xu, K. and D.A. Randall, 1995: Impact of Interactive Radiative Transfer on the Macroscopic Behavior of Cumulus Ensembles. Part II: Mechanisms for Cloud-Radiation Interactions. J. Atmos. Sci., 52, 800–817, https://doi.org/10.1175/1520-0469(1995)052<0800:IOIRTO>2.0.CO;2

---

## Author Comment (AC10) · 3 Oct 2018

Reply to minor comments:

1. Fig. 8: Unit of color contours missing.

Reply:

We are sorry for the mistake. The figure units have been added.

2. Fig. 8: Why does the MOC plot stop at 1600m depth?

Reply:

According to previous studies (Liu et al.,2011; Buckley and Marshall,2016), the MOC in upper levels plays an important role in heat transport that affects the air-sea interaction. The NPMOC at deep ocean levels has not been discussed in Liu et al.(2003). Without comparison with other studies, we are not sure how to correctly interpret the model biases.

3. Page 8, Line 6: Details for the reference Huang et al. (2014) are missing.

Reply:

Thank you for reminding us about the problem. We have added the reference in the bibliography.

4. Page 5, Line 29: It is not clear whether the control experiments for the ECHAM5/NEMO3.6st and ECHAM6/NEMO3.6st setup use present day or piControl external forcing. Please clarify.

Reply:

We use the present day climatology, which is also written in section 2.4.

5. Page 11, line 5: The Wang et al., 2014 paper cited focuses on the Atlantic MOC and not on the Pacific MOC, hence the citation in this context is not appropriate.

Reply:

Thank you for pointing out the problem. In the revised manuscript, section 5.1 contains inter-model comparison of both NPMOC and AMOC. Therefore, this reference is unremoved.

6. Page 11, line 15: The Ge et al., 2017 reference is not appropriate in the context, as Ge et al., 2017 focus on the impact vertical resolution on the SST bias in a ocean model (MOM5) driven by reanalysis data and not the impact of coupling frequency.

Reply:

Thank you for pointing out the problem. We have removed the citation.

7. Page 11, line 31 and Figure 9: Please provide the coordinates you use to determine the SEC region. Figure 9 does not show zonal averages (longitudes on the x axis) as indicated in the text. Please correct.

Reply:

Thank you for pointing out the problem. The statement has been corrected. The coordinates used to calculate the vertical circulation are marked within the box on subplot(d) (Fig. 11 in the revised manuscript).

8. Page 12, line 28: LH flux and evaporation describe the same physical process, so there is no need to discuss the two separately.

Reply:

Thank you for your suggestion. We have deleted the relevant content.

So far, we have addressed all the questions in the SC5 and have modified the manuscript according to your comments. Thank you for giving us so many suggestions.

Although the answer to some questions may not be satisfying for you, that is the best level we can achieve at the current stage. we will do better in future.

The final version of manuscript is attached in this post, with yellow highlights on the major modifications.

Please also note the supplement to this comment:
https://www.geosci-model-dev-discuss.net/gmd-2018-130/gmd-2018-130-AC10-supplement.pdf

[Figure]

[Figure]

**Supplement:**

**Simulation Improvements of ECHAM5-NEMO3.6 and ECHAM6-NEMO3.6 Coupled Models Compared to MPI-ESM and the Corresponding Physical Mechanisms**

Shu Gui [1], Ruowen Yang [1], and Jie Cao [1],

5   [1] Department of Atmospheric Sciences, Yunnan University, Kunming, 650091, China.

*Correspondence to*: Jie Cao (caoj@ynu.edu.cn) and Ruowen Yang (yangruowen@ynu.edu.cn)

**Abstract.** To improve the model simulation through decisive coupling mechanisms, rather than blindly updating the parameterization schemes, it is necessary to compare model performances between the CGCMs with the same atmospheric or oceanic component model. Therefore, two new CGCMs have been developed with the same oceanic component model, namely

10   ECHAM5-NEMO3.6 and ECHAM6-NEMO3.6. The MPI-ESM that consists of ECHAM6 and MPIOM has also been employed. Experiments are carried out with the same settings in coupler and individual component model if applicable, and the new models show substantial improvements in the simulation of SST, precipitation and ocean currents. Further analysis has made it clear that the primary cause of SST biases in ECHAM5-NEMO3.6 and ECHAM6-NEMO3.6 can be attributed to the momentum field, while oceanic dynamics and surface radiation budget are accountable for more SST deviations in the

15   MPI-ESM. Inter-model comparison between the coupled models with the same oceanic model suggests that cumulus convection is in the central part of simulation differences, which finally influence the SST through cloud radiative forcing and WES feedback mechanism. Whereas the OGCM replacement shows that ocean advection plays an important role in modulating the atmospheric and oceanic circulations through air-sea feedback. The mechanisms revealed in this study provide a new perspective of bias genesis during model coupling, which can be helpful for tuning other climate models towards a more

20   realistic simulation.

**1 Introduction**

The physical processes sensitive to the air-sea interaction and its impact on climate variabilities have been studied for decades. Research findings suggest that dynamic air–sea coupling is important for tropical cyclone prediction regarding its intensity and rate of intensification (Sandery et al., 2010; Chen et al., 2013; Lin et al., 2018). The air-sea interaction has a

25   substantial effect on precipitation response to ENSO teleconnections (Langenbrunner and Neelin, 2013), which is also associated with basic state climatology (Ham and Kug, 2015) in the Coupled Models Intercomparison Project phase 5 (CMIP5) models. The double intertropical convergence zone (ITCZ) problem of precipitation simulation in the coupled general circulation models (CGCMs) are closely linked with ocean-atmosphere feedbacks, including Bjerknes feedback, sea surface temperature (SST)-surface latent and surface shortwave flux feedback (Lin, 2007). The ocean-atmosphere interaction tends to

amplify trade wind biases in general circulation models (GCMs) (Li and Xie, 2014), which is also responsible for excessive cold tongue simulation in equatorial Pacific. Besides traditional understanding of coupling processes that account for SST and precipitation biases, recent studies have revealed other factors in the air-sea interaction that significantly contribute to the model bias pattern. Burls et al. (2017) find a quadratic relationship between extra-tropical Pacific albedo and equatorial SST bias, and Pham et al. (2017) suggested that the deep cycle of cold tongue turbulence can be affected by cloud cover and rain. The air-sea kinetic energy budget is found to be linked with surface gravity waves, where wave age and friction velocity affect in the ratio between kinetic energy from the winds and underlying surface currents (Fan and Hwang, 2017).

The dynamic mechanisms that play a major role in the biases propagation in the CGCMs are also investigated from a wide range of perspectives. Double ITCZ precipitation problem is usually associated with biases in radiation budget and surface winds (Lin, 2007). There is a close relationship between clouds and SST variation (Klein & Hartmann, 1993; Norris & Leovy, 1994), and changes of low clouds and shortwave radiation flux react on the SST and sea level pressure (SLP) (Norris et al., 1998; Mochizuki & Awaji, 2008; Bond & Cronin, 2008). Wu and Kinter III (2010) suggested that high-frequency changes in atmospheric circulation affects largely on the surface shortwave radiation, and hence its correlation with SST variability in the mid-latitude North Pacific. The weak simulation of Atlantic meridional overturning circulation (AMOC) is found to be responsible for sea surface temperature (SST) cold biases in the northern hemisphere in 22 CMIP5 climate models (Wang et al., 2014), which poses a great impact on the North Pacific through Northern Hemisphere annular mode (NAM) and wind-evaporation-SST (WES) feedback, combining SST biases in extratropical North Atlantic (ENA) and tropical North Atlantic respectively (Zhang & Zhao, 2015). The temperature bias can also be attributed to underestimation of water vapor amount (Liu et al., 2011), radiative and non-radiative processes (Ren et al., 2015), modelling of cloud-radiation feedback (Song et al., 2012) and gap winds (Sun & Yu, 2006). Wu and Liu (2003) suggested that the regional SST variations in central and eastern North Pacific are separately resulted from changes of Ekman advection and surface heat flux, both of which are affected by the subtropical ocean circulation, as a part of meridional overturning circulation in North Pacific (NPMOC). The NPMOC has been found to work as a bridge for mass and heat exchanges (McCreary and Yu, 1992; Liu et al., 1994), and is largely driven by sea surface wind stress and resulting in east-west sea level slope (Liu et al., 2011; Liu et al., 2013). However, there are larger discrepancies in the estimates of drag coefficients among different computational approaches, especially due to uncertainties in velocity measurements and removal of non-wind-driven currents.

Some efforts have been made to improve the SST simulation quality in a variety of CGCMs, for example, through changing zonal filtering and advection scheme (Xiao, 2006), modifying radiation and cumulus parameterization scheme (Bao et al., 2010), including frozen precipitating hydrometeors in cloud mass (Li et al., 2014), and decreasing relative humidity threshold for low cloud formation (Tang et al., 2016). However, these studies only focus on a limited range of processes that turn out to be important in the statistical analysis of model biases. Updates in the parameterization schemes bring in both improvements and setbacks in simulation. On behalf of oscillations in atmosphere-ocean coupling, the contribution of the individual component model to the simulation of some key variables, regarding ENSO variability, extreme precipitation and hurricanes, and climate response to anthropogenic greenhouse gases, have not yet been made clear.

This paper studies the simulation improvements by new combinations of component models, analyses differences by respectively changing each component model, and determines the key physical processes behind the simulation deviations. To compare relative contributions of the individual component model in the CGCM, we need 2 CGCMs based on the same oceanic model but different atmospheric models, and another 2 CGCMs with the same atmospheric model but different oceanic models. For this purpose, two new CGCMs have been developed following previous studies that establish a coupling system with the ECHAM and NEMO (Gualdi et al., 2003; Park et al., 2009; Huang et al., 2014). One uses ECHAM5.4 as the atmospheric component model and NEMO3.6st as the oceanic component model, which is referred to as ECHAM5-NEMO3.6. The other uses ECHAM6.3 as the atmospheric component model and NEMO3.6st as the oceanic component model, and thus referred to as ECHAM6-NEMO3.6. The MPI-ESM developed by Max-Plank Institute for Meteorology, based on ECHAM6.3 for atmosphere and MPIOM for ocean, is also used in this study. To minimize simulation differences caused by model configurations, the three coupled models are set to the same coupling frequency of every 4 hours, and the same horizontal resolution T63 (192 longitudes×96 latitudes) for the atmospheric component model. The ocean models, namely MPIOM and NEMO3.6st, have different model structures and thus being used with their own default configurations. The content organization of this paper is as follows: A brief description of model frameworks and experiment setup configurations are presented in section 2. The reanalysis data sets used for model assessment and bias analysis are illustrated in section 3. Model evaluation and comparison among ECHAM5-NEMO3.6, ECHAM6-NEMO3.6 and MPI-ESM are presented in section 4. Further analysis on the cold tongue bias and opposite SST bias in North Pacific is presented in Section 5. Determination of key physical processes responsible for the differences in SST simulation after changing each component model is elaborated in Section 6. Summary and discussion part is in Section 7.

**2 Model Description**

**2.1 OGCM**

**2.1.1 NEMO**

NEMO model is a well renowned modelling system with high skills in global oceanic circulation simulation, which has been widely used for scientific research, weather forecast (Storkey et al., 2014; Megann et al., 2014), and reanalysis data assimilation (Mogensen et al., 2012a, b). Designed to serve as a flexible tool for ocean and sea ice studies, NEMO manifests good usability interacting with other ACGMs (Gualdi et al., 2003; Luo et al., 2005; Park et al., 2009; Dunlap et al., 2014; Huang et al., 2014). The NEMO stable version 3.6 has been employed in this study, whose ocean component is configured to calculate primitive equations on the ORCA2 grid, a tripolar grid of 182 (longitude)×149 (latitude) curvilinear orthogonal mesh in horizontal direction and 31 vertical levels unevenly distributed on partial step z coordinate in current research. Turbulent kinetic energy (TKE) closure scheme has been chosen for vertical mixing with enhanced vertical diffusion for convective processes. The Louvain-la-Neuve sea-ice model (LIM3), originally developed by Fichefet and Morales-Maqueda (1997), has

been incorporated in NEMO3.6 to represent the sub-grid-scale dynamics and their impact on sea ice thickness and ice-ocean salt exchanges. Main differences between LIM3 and other ice models are related to the physical parameterization of open boundary conditions and sea-ice interactions, with the C-grid formulation of elastic-viscous-plastic rheology (Bouillon et al., 2013).

**2.1.2 MPIOM**

MPIOM model is formulated on Arakawa-C grid for horizontal dimension and z-grid for vertical dimension, using the hydrostatic and Boussinesq approximations in the model dynamic equations (Jungclaus et al., 2006; Jungclaus et al., 2013). Vertical mixing and diffusion are parameterized following Pacanowski and Philander (1981), with a diffusion coefficient varies with grid spacing (Redi, 1982). Sea-ice model is included in MPIOM where sea-ice thickness is modulated by turbulent atmospheric fluxes and oceanic heat transport (Wolff et al., 1997; Marsland et al., 2003; Notz et al., 2013). The model is configured with 40 unevenly spaced vertical levels on the GR1.5 grid, a conformal mapping grid of 256 (longitude) × 220 (latitude) in the horizontal making horizontal resolution approximately 1.5°.

**2.2 AGCM**

The ECHAM atmospheric model developed by the Max Planck Institute for Meteorology has been used in many studies since its first version (ECHAM1) branched from the cycle 17 operational model at Medium Range Weather Forecasts (ECMWF) (Roeckner et al., 1989; Simmons et al., 1989). Incorporation of new features along the course of model development gradually makes ECHAM capable of reproducing meticulous characteristics in the weather system, including those in cumulus convection, moisture transport, radiation and land-surface processes (Roeckner et al., 1996, 2003; Raddatz et al., 2007; Brovkin et al., 2009). ECHAM has also been employed in the coupled earth modelling system, from coupling with large‐scale geostrophic ocean model (LSG) (Maier‐Reimer et al., 1993) to the latest version of Earth system model MPI-ESM (Baehr et al., 2015). The ECHAM model consists of a dry spectral dynamic core, a set of parameterization schemes dealing with solar irradiance, moist convection, land-surface properties, etc. The versions in use for this paper are ECHAM5.4 (Roeckner et al., 2003) and ECHAM6.3 (Stevens et al., 2013). Major updates from ECHAM5 to ECHAM6 include improved representation of shortwave spectrum, a new aerosol parameterization scheme, middle atmosphere and surface albedo descriptions are also enhanced. Despite the new implementations in ECHAM6, the AGCMs are set to the same configuration if applicable, to minimize the differences caused by updates of physical parameterization schemes.

**2.3 CGCM**

**2.3.1 ECHAM5-NEMO3.6**

The schematic structure of ECHAM5-NEMO3.6 is shown in Fig. 1a. Overall, the ECHAM5-NEMO3.6 consists of the atmospheric model ECHAM5.4, oceanic and sea ice model NEMO3.6st (the stable version of NEMO3.6), and the coupler

Ocean Atmosphere Sea Ice Soil (OASIS3) (Valcke, 2013). Although ECHAM5 (Roeckner et al., 2003, 2006) is an older version of the atmospheric model developed by Max Plank Institute of Meteorology compared with ECHAM6 (Stevens et al., 2013), it was employed by the previous coupled atmosphere ocean model ECHAM5/MPI-OM (Jungclaus et al., 2006), which has been well tested and reassured of accurate surface flux transfer between the oceanic and atmospheric component models.

5    Based on the interchange coupling structure of the ECHAM5/MPI-OM, seventeen variables are passed from ECHAM5 to NEMO-3.6st through OASIS3 coupler, including solar radiation, non-solar heat flux and its derivative with respect to temperature, zonal and meridional wind stress, evaporation minus precipitation, and sublimation. The meridional and zonal wind stress vectors are passed to the ORCA2 U and V grid of NEMO-3.6st, while other variables are passed to the T grid. Both ocean and ice regions are considered in the coupling processes. In the opposite direction, six variables comprised of the

10   SST, sea ice temperature, sea ice fraction, sea ice albedo and surface ocean currents are transferred from the ocean model to the atmosphere model. Bilinear interpolation method is used in the exchanges of physical variables between ECHAM5 gaussian grid and NEMO-3.6st ORCA2 grid. The time steps for ocean and atmosphere models are both set to 1200 seconds, and the coupling frequency is 4 hours once (every 12-time steps). At present, the component models are integrated with the default parameter settings suggested in the user manual, model retuning will be scheduled in further research.

15   **2.3.2 ECHAM6-NEMO3.6**

Following the coupling framework of MPI-ESM (the update version of ECHAM5/MPI-OM, version number MPI-ESM-1.2.00p4), ECHAM6-NEMO3.6 has been developed with the same atmospheric component model ECHAM-6.3, coupled with NEMO 3.6 stable version through OASIS3-MCT (Craig et al., 2017) (Fig. 1b). Model retuning of the ECHAM-6.3 and NEMO3.6 is left for further studies due to time limitation and high computational cost. Namelist settings of the ECHAM-6.3

20   and the coupler OASIS3-MCT are brought into correspondence with those in ECHAM5-NEMO3.6 to the utmost, for example, with the same horizontal resolution T63 on gaussian grid and the same parameterization settings for greenhouse gases. The oceanic component model NEMO3.6 still uses the same configuration as that in ECHAM5-NEMO3.6. Coupling variables of the ECHAM6-NEMO3.6 are the same as those in ECHAM5-NEMO3.6. Details in experiment configuration are elaborated in section 2.4.

25   **2.3.3 MPI-ESM**

The MPI-ESM is comprised of the atmospheric general circulation model ECHAM6 and the oceanic circulation model MPIOM (Jungclaus et al., 2013; Stevens et al., 2013). It has been continuously developed at Max Planck Institute for Meteorology and has been successfully applied on a broad range of studies, including volcano studies (Zanchettin et al. 2013; Zhang et al. 2013), anthropogenic land cover change (Reick et al., 2013; Brovkin et al., 2013), circulation feedback sensitive

30   to Intertropical Convergence Zone (ITCZ), and double ITCZ precipitation (Mobis and Stevens, 2012). MPI-ESM has been used in CMIP5 and is employed in the upcoming CMIP6. Due to high computational cost, the low-resolution version MPI-ESM-LR is used in this study (version number MPI-ESM-1.2.00p4), with ECHAM6 running at T63L47 resolution (horizontal

resolution about 1.875° and 47 vertical levels). Experimental settings are migrated from piControl default configuration, and then adjusted to being in consistence with those of ECHAM5-NEMO3.6 and ECHAM6-NEMO3.6 experiments. The major differences from default piControl experiment are the increased coupling frequency (4 hours once), and climatology recalculated from the year 1981 to 2010 to serve as the model input files (e.g. aerosol properties, ozone mole fractions and

5    land use transitions).

**2.4 Experimental Setup**

The control experiments conducted in this study are aimed to reproduce atmospheric and oceanic circulation characteristics of present time, and then to compare with each other in order to examine model performance improvements. The coupled control simulation is thus configured with reference to MPI-ESM piControl experiment settings. The $CO_2$ value

10   is set to default 353.9 ppm in the user manual. Other greenhouse gases like $NO_2$ also follows the default present time setting so that they are consistent with each other. The aerosol settings use the climatology compiled by S. Kinne without any complementation of volcanic aerosols. The NEMO3.6 is initialized with temperature and salinity climatology from World Ocean Atlas (WOA) data, applying the geothermal heating at ocean bottom. The RGB formulation (Lengaigne et al., 2007) has been chosen to calculate the light penetration over the sea surface with observed time varying chlorophyll. The sea ice

15   model (LIM3) in NEMO3.6 is configured to compute the ice-ocean fluxes under the influence of air-sea fluxes, ocean mass and salt exchanges, with light penetration of solar radiation. Ice freezing and melting also affects the albedo in the Arctic and Antarctic regions. Likewise, the sea ice thickness and density in the MPIOM respond to wind stress and ocean currents without consideration of turning angles. Surface heat balance and the internal ice stress also affect the variations of sea ice cover with zero-layer formulation of Semtner (1976). Model initialization is started from the climatology basic state recalculated with the

[revised manuscript text omitted]

**4.7 Model variability of ENSO and SAM**

In the coupled ocean-atmosphere system, global climate variability has been driven by the El Niño-Southern Oscillation (ENSO) and the southern annular mode (SAM, also called the Antarctic Oscillation) (Philander, 1990; Wallace and Thompson, 2002). It is therefore necessary to examine the model variability by applying spectra analysis on relevant indices. The CGCM simulations of the three indices are generally consistent with the theoretical red noise (Markov) spectrum (figure omitted). The Niño3.4 index is defined as the SST anomalies averaged over the NINO34 region (5°N-5°S,170°W -120°W). It shows high variance in 2-7 years' period that documents the ENSO peaks in the HadISST reanalysis (Fig. 8a). All of the CGCMs reproduce similar variations of the Niño3.4 power spectra. The ECHAM6-NEMO3.6 presents weak variabilities at the interannual and

interdecadal scale, whose periodic peaks are about one year less than the reanalysis counterpart. The ECHAM5-NEMO3.6 shows a better spectral distribution that best coincides with the reanalysis at the interannual scale. However, it still suffers a weak variability at the interdecadal scale and the periodic peak is even half a year less than that of the ECHAM6-NEMO3.6. The MPI-ESM instead takes on an intensified interannual variability, which stays strong at the interdecadal scale. The Southern Oscillation Index (SOI) is calculated based on the differences in sea level pressure anomalies between Tahiti and Darwin in Australia. In comparison to the Niño3.4 spectra, the SOI exhibits similar peaks at the interannual scale in the ERA-20c reanalysis (Fig. 8b). Nevertheless, all the CGCMs reproduce weak variabilities at the interannual and interdecadal scales. The ECHAM6-NEMO3.6 presents the best simulation with a significant increase in variance around 4 years' period, while the ECHAM5-NEMO3.6 shows the weakest variability at the interannual scale. It implies that the AGCM replacement has an opposite effect on the Niño3.4 and SOI variabilities. The MPI-ESM also show reduced variance from the annual scale and above, quite the opposite to that in the Niño3.4 case. Since the model biases of ENSO variability may be attributed to thermocline feedback and zonal wind variations (Borlace et al., 2013), the reversed changes in the variabilities of Niño3.4 and SOI can be caused by the related oceanic and atmospheric processes. The SAM index is calculated following Gong and Wang (1999) by the differences of normalized monthly zonal mean sea level pressure at 40°S and 65°S. Variations of SAM tend to be more flattened than those of SOI in the ERA-20c reanalysis (Fig. 8c), with prominent fluctuations from biannual to interannual scales. Compared with the reanalysis counterpart, all CGCMs show more power at interannual time scales that represents a robust modulation of the SAM, which is possibly attributed to the semi-annual oscillation (SAO) (Hurrell and van Loon, 1994) and circulation anomalies over Antarctica (Thompson and Solomon, 2002). The ECHAM6-NEMO3.6 presents stronger decadal variability than that of the reanalysis data, while the ECHAM5-NEMO3.6 exhibits weaker low-frequency variability. Since the high variance in low-frequency band represents the upward trend of SAM index at decadal scale (Raphael and Holland, 2006), updating the AGCM can result in a drastic change of long-term climate variability in southern hemisphere. In contrast, the MPI-ESM shows the SAM variability very close to the reanalysis counterpart from 1 year and above, indicating that the OGCM feedback to the atmosphere can lead to a better representation of the inter-decadal variability.

**5. Circulation Patterns Relevant to SST Biases**

The importance of Meridional Overturning Circulation (MOC) to SST in coupled models has been proved in previous studies (Wang et al., 2014, Liu et al., 2016). Thus, the comparison of North Pacific MOC (NPMOC) (Fig. 9) between the CGCMs with the same atmospheric or oceanic component model can help to explain bias characteristics in relation to SST deviations. Model bias pattern can be influenced by the choice of ocean reanalysis dataset. Uncertainties in the ocean reanalyses (ORAs) include the estimation of sea ice thickness, interannual variability of salinity, surface heat flux and mixed layer depth (Balmaseda et al., 2015; Toyoda et al., 2017). There are substantive discrepancies in temperature, salinity and density in the deep ocean and the Southern Ocean, where observations are sparse especially before the year 2000. Ocean heat content at deep levels varies largely among the ORAs, with majority of spread originating in the Southern Hemisphere due to lack of

observation for data assimilation (Palmer et al., 2017). Discrepancies among the ORAs are evident in the strength and structures of the AMOC, with distinctive differences in the depth of equatorward return flow and the depth of the maximum AMOC in northern high latitudes (Karspeck et al., 2017). However, the uncertainties of climatology above bathypelagic zone are less than those of deeper levels. Since the SODA reanalysis data has been employed in many ocean studies, including the North Pacific MOC that is mostly within the thermocline layer, it is reliable to use the SODA reanalysis data to analyse model biases related to the MOC.

**5.1 Meridional Overturning Circulation**

The ECHAM5-NEMO3.6 possesses two prominent improvements of the SST simulation, one in the Pacific cold tongue region, and the other in North Atlantic. It also exhibits warm SST biases in the North Pacific that is the opposite to most CMIP CGCMs. These peculiar characteristics require further investigation on the meridional overturning circulation (MOC) in North Pacific and North Atlantic. For the MOC in North Pacific, the ECHAM5-NEMO3.6 and ECHAM6-NEMO3.6 possess similar bias patterns overall, with intensified tropical cell and deep tropical cell that are mentioned in Liu et al. (2011) (Fig. 9a, b). Tropical cell enhancement is more significant in the ECHAM5-NEMO3.6, so that the upwelling in the tropics and subsequent heat transport to mid-latitudes are more than those in the ECHAM6-NEMO3.6. Bias in the MPI-ESM experiment manifests itself to a larger degree (Fig. 9c). The piControl experiment result (available in http://esgf-node.llnl.gov/) is attached (Fig. 9d). The piControl run data used in this study has the same time span as that of the reanalysis data from the year 1981 to 2010, which should be able to represent the model abilities in reproducing the climatology of the same period. More contour lines appear in NPMOC biases of piControl than the MPI-ESM experiment conducted in this paper (Fig. 9c, d). Although the MPI-ESM in the CMIP5 piControl experiment is an older version, it can at least demonstrate that increasing coupling frequency from one day to 4 hours does not degrade the MPI-ESM simulation. Previous studies also suggest that model simulation can be improved by decreasing coupling interval (Bernie et al., 2008). Hence, the MPI-ESM experiment result and the inter-model comparison with other CGCMs are trustworthy. Enhancement of tropical cell in MPI-ESM is unremarkable, compared with that of ECHAM5-NEMO3.6, but deep tropical cell is significantly intensified that tropical upwelling is forced to become too strong. This may explain why tropical SST bias of MPI-ESM is more than 2℃ in boreal summer, without significant radiation errors and easterly anomalies in surface stress. Excessive upwelling in tropical Pacific, induced by intensified deep tropical cell of NPMOC, cools down the local SST. Unlike the ECHAM5-NEMO3.6 case, where the SST cooling in tropical Pacific is driven by intensified tropical cell of NPMOC that transports more heat to mid-latitudes, the MPI-ESM simulation of tropical and subtropical cells does not differ much from SODA reanalysis data. The resulting poleward heat transport carried by NPMOC is not increased in the MPI-ESM experiment. Therefore, the SST bias in North Pacific remains negative (Fig. 3e) under the impact of Northern Hemisphere annular mode (NAM) and wind-evaporation-SST (WES) feedback, when cold SST biases appear in tropical and extratropical North Atlantic (Zhang & Zhao, 2015). The cold SST bias in North Pacific and North Atlantic for ECHAM6-NEMO3.6 can also be explained with the same reason. But for the ECHAM5-NEMO3.6 case, the poleward heat transport has been enhanced by intensified tropical cell to bring up the SST in mid-latitudes (Fig. 3c).

Since the upper cell of Atlantic meridional overturning circulation (AMOC) plays a significant role in delaying warming signals from anthropogenic greenhouse gases and responding to climate change (Marshall et al., 2014; Buckley and Marshall, 2016), model bias analysis is still focused on the upper ocean levels. The overall magnitude of AMOC bias is less than that of NPMOC with significantly reduced biases near the sea surface (Fig. 10), which is consistent with those of surface currents among the three CGCMs. The ECHAM6-NEMO3.6 shows exiguous bias near the ocean surface, but presents strong biases in the mesopelagic zone of subtropical areas, bringing more heat to higher latitudes (Fig. 10a). Likewise, the ECHAM5-NEMO3.6 exhibits strong circulation biases rotating clockwise in the thermocline that intensifies poleward heat transport (Fig. 10b). In the upper ocean levels, the AMOC poleward transport is a little more enhanced than that of the ECHAM6-NEMO3.6. With similar bias patterns of the AMOC, the ECHAM5-NEMO3.6 and the ECHAM6-NEMO3.6 have opposite SST biases in North Atlantic (Figs. 3a and 3c), which implies that the air-sea feedback including WES feedback and NAM as suggested by Zhang & Zhao (2015) takes the responsibility. The MPI-ESM experiment shows negative biases in tropical Atlantic from the sea surface to the bathypelagic zone, indicating that the overturning circulation has been restrained. There is a narrow positive bias in the subtropical Atlantic, but its strength has been limited by the negative biases nearby. One consequence of the weak AMOC is the decrease of SST in North Atlantic due to less heat supply from the tropics (Fig. 3e). The overturning circulation is enhanced in the middle latitudes with one centre located north of $35^{\circ}$ N and another centre around $55^{\circ}$ N at the depth of 1200m. It still promotes the poleward heat transport and results in warm SST biases in subpolar region (Fig. 3e). The AMOC biases in the MPI-ESM piControl experiment are similar as those in the MPI-ESM experiment, with more negative biases in tropical Atlantic. Comparing the AMOC biases between the MPI-ESM and the ECHAM5-NEMO3.6, it can be seen that the SST cold biases in North Atlantic are partially attributed to decreased MOC in the thermocline of tropical and extra-tropical oceans. However, the air-sea interaction also takes account of the SST variations in consideration of the SST differences between the ECHAM5-NEMO3.6 and the ECHAM6-NEMO3.6. Zhang & Zhao (2015) suggested that the cold SST bias in Atlantic caused the same cold bias in North Pacific through different mechanisms originating in tropical and extra-tropical Atlantic. Because the differences of NPMOC are bigger than those of AMOC between these two newly developed CGCMs, it suggests an inverse cause-and-effect relationship between the cold SST biases in North Pacific and North Atlantic where the former takes the lead.

**5.2 Vertical Structure of Atmospheric Circulation**

On behalf of the atmosphere motion that accounts for cold tongue bias and strong easterly bias in surface currents, vertical circulation bias meridionally averaged over South Equatorial Current (SEC) against the ERA-Interim reanalysis is given in Fig. 9. Consistent with surface wind stress bias (Fig. 5), there are easterly anomalies in the lower atmosphere for ECHAM5-NEMO3.6 and ECHAM6-NEMO3.6 (Fig. 11a, b). With anomalous upward (downward) motion over the western (eastern) Pacific, the easterly anomalies are maintained by the anomalous Walker circulation across tropical Pacific. The easterly anomalies in lower atmosphere decrease significantly in MPI-ESM experiment (Fig. 11c), without strong downward motion

in the eastern Pacific. Inter-model differences in the easterly biases at low levels are consistent with the amplitude of NPMOC tropical cell, because the stronger surface winds collaborating with Coriolis effect push more sea water to move west and poleward.

It can be seen from Figure 11 that cold temperature biases in the MPI-ESM transform into warm temperature biases in the ECHAM5-NEMO3.6 accompanied by increasing easterly anomalies in lower atmosphere, which suggests warmer temperature bias is correlated with stronger easterly anomalies. The pattern correlation between temperature biases and easterly anomalies in the three CGCM experiments has been calculated respectively. Because the easterly biases are most remarkable below 800hPa, the levels taken into consideration are between 800hPa and 1000hPa. The pattern correlation coefficients between temperature biases and zonal wind biases are 0.315 for the ECHAM6-NEMO3.6 experiment, 0.586 for the ECHAM5-NEMO3.6 experiment, and 0.411 for the MPI-ESM experiment. All of them have passed the 99% Student-t significance test. The ECHAM5-NEMO3.6 has the most significant easterly biases and holds the largest correlation coefficient, much bigger than that of the other two CGCMs, which confirms the correlation between the warmer temperature and stronger easterly anomalies. Recalling precipitation biases over the SEC area (Fig. 4a, c, e), where ECHAM5-NEMO3.6 shows the driest situation and MPI-ESM presents the wettest circumstance, the latent heat absorption in the course of evaporation turns out to be responsible for the temperature bias. More precipitation requires higher specific humidity that favours cloud formation, in which more latent heat of vaporization is taken up when water vapor mixing ration is increased.

The analysis on oceanic and atmospheric circulation has made it clear that the SST bias is consistent with meridional overturning circulation in North Pacific, driven by surface wind stress anomalies that are maintained by anomalous Walker circulation over the tropical Pacific. Since cumulus convection modulates changes in temperature, specific humidity and atmospheric circulation, it is most likely to be the predominant factor that shapes the inter-model differences. To better understand the impact by changing each component model, it is necessary to quantitatively analyse their contributions to simulation differences.

**6. Contribution of Each Component Model**

A total of 11 variables that are sensitive to air-sea coupling are selected to calculate pattern correlation with SST simulation differences in boreal summer. Correlations between model differences in SST and each variable with ECHAM6-NEMO3.6 minus ECHAM5-NEMO3.6, denoted as "AGCM" column in Table 1, indicate the effects of changing atmospheric component model at the global scale. Similarly, correlation results with ECHAM6-NEMO3.6 minus MPI-ESM to represent the contribution of changing oceanic component model are named "OGCM" in Table 1. Because the analysis in previous sections is mainly focused on the Pacific, the pattern correlation over the area is also provided for comparison (Table 2).

**6.1 Effects of AGCM Replacement**

For the AGCM case, net longwave radiation is ranked top in the global correlation, followed by surface evaporation (latent heat flux) and net shortwave radiation (Tab. 1). When the study area is narrowed down to the Pacific, the top 3 ranking variables are the same, with the net shortwave radiation takes the 1$^{st}$ place (Tab. 2). In both cases, the net shortwave and net

5  longwave fluxes are ranked in top 3, which suggests that radiation budget is closely associated with SST inter-model differences. The pattern correlations of the 3 variables are higher within the Pacific than that at the global scale, indicating an increased robustness of the variable estimates. Changing the atmospheric component model from ECHAM-6.3 to ECHAM-5.4 affects cumulus convection processes, including temperature, specific humidity and winds, which further alter the ground radiation fluxes through cloud radiation feedback. Noting that latent heat flux has a correlation value close to that of net

10  longwave radiation, it would suggest that cumulus convection changes sea surface winds and then surface evaporation, leading to differences in latent heat flux. Consequently, the SST is altered in tropical oceans through wind-evaporation-SST (WES) feedback. The negative sign of correlations for longwave and latent heat flux (Tab. 1) suggests a contrary trend with SST variations. It is easy to understand because more latent heat and longwave dissipation cools down the surface sea water. Similarly, positive correlation for shortwave flux is due to more solar irradiance that brings up the SST.

15  To determine the physical processes responsible for simulation differences, changes in SST, surface winds, radiation budget and vertical circulation are plotted, respectively. It can be seen from Fig. 12d that negative deviations almost occupy the northern hemisphere, including tropical Pacific and Indian Ocean, North Pacific and North Atlantic. First, the mechanisms behind SST deviations in tropical and subtropical oceans are discussed here. Deviations of 10m wind exhibit southerly anomalies around the central tropical Pacific in southern hemisphere (Fig. 12b), where easterly winds prevail for summer

20  climatology (Fig. 10a). The easterly winds superimposed by southerly anomalies result in a bigger wind speed that helps to increase surface evaporation. Hence the latent heat absorption over the sea surface are enhanced that makes SST deviation colder than 1℃ (Fig. 12d). There are eastward oriented wind anomalies over subtropical Indian Ocean, subtropical Atlantic and some parts of subtropical Pacific (Fig. 12b) in the opposite direction against climatology (Fig. 12a). Their superposition results in a decreased wind speed that reduces surface evaporation and latent heat flux, which finally keeps the SST warm for

25  those sea waters. The SST deviations thus form a positive feedback (WES feedback) with changes of surface wind and evaporation, in accordance with signs of correlation in Table 1. Since the latent heat and surface wind differences are caused by replacing the AGCM and the associated air-sea feedback, it is advisable to compare deviations in vertical circulation that may shed some light on corresponding physical processes. An anti-clockwise Walker circulation accompanied by negative temperature deviations occupies central and eastern Pacific over the SEC area (Fig. 13a). Low-level easterly anomalies are

30  consistent with surface wind distribution (Fig. 12b) because northerly flows in southern tropical Pacific that contributes to more evaporation are not considered in meridional average. It can be assumed that changing the AGCM alters radiation budget, and tropospheric temperature becomes colder so that more air currents cool down and sink by insufficient heat. Downward

motion over the East Pacific connects easterly anomalies in the middle troposphere, which forms a complete Walker cell that further enhances low-level westerly anomalies.

To verify this assumption, differences in surface heat budget with respect to net shortwave and longwave fluxes, latent heat and sensible heat are drawn in Fig. 14. From the general view of ocean surface energy balance, the amount of incoming and outgoing energy should be equal. Variations of shortwave flux and latent heat are more significant than those of longwave and sensible heat fluxes after replacing the AGCM. Differences in shortwave radiation can be attributed to cloud radiation feedback, which is closely associated with cumulus convection. Whereas the deviations in latent heat flux are derived from surface wind anomalies and the SST variations. More latent heat release appears over the areas where the surface winds are intensified by local anomalies (Fig. 12a, b) or the SST is higher. The deviation patterns are congruent with the correlation signs (Tab. 1 and 2). More solar irradiance corresponds to higher SST of subtropical oceans in southern hemisphere, while less shortwave flux matches cold SST in central Pacific. Under the same cloud radiative forcing, net longwave flux shows approximately inverse variations. Differences in surface winds and heat budget also affect the latent heat flux. Therefore, it can be confirmed that the AGCM replacement first alters cumulus convection that modulates temperature, specific humidity and atmospheric circulation, which in turn accommodates cloud radiation feedback and changes the radiation budget (Xu and Randall, 1995; Stephens et al., 2008; Ghate et al., 2015). The effects cover both North Pacific and North Atlantic. It has been made clear in section 5.1 that the enhanced NPMOC in ECHAM5-NEMO3.6 is responsible for warm SST biases in North Pacific. On the contrary, the AMOC enhancement is less significant. However, with increased surface heating in subtropical and extra-tropical North Atlantic, the MOC transports more heat to higher latitudes which ameliorates cold SST biases in North Atlantic. Roles of other teleconnections between North Pacific and North Atlantic suggested by Zhang & Zhao (2015), such as NAM through air-sea feedback, is left for a future research.

**6.2 Impacts of OGCM Replacement**

For simulation differences after changing the OGCM, latent heat flux holds the biggest share in pattern correlation with corresponding SST differences (Tab. 1), much bigger than that of net longwave flux at the second place and sensible heat flux at the third place. When the study area is confined within the Pacific, the three leading variables are the same but the sensible heat flux is ranked top (Tab. 2). Compared with the rankings in the AGCM case, it can be seen that the OGCM influences the simulation with different physical mechanisms than those by replacing AGCM. The effect of cloud radiation feedback is not prominent, while the physical processes associated with latent heat and sensible heat, including heat conduction, evaporation and convection, take a bigger share in the SST inter-model differences. SST differences can also lead to changes in longwave flux. The negative sign of correlations for latent heat, sensible heat and longwave flux (Tab. 1 and 2) suggests a reverse trend to SST variations.

Contrary to the AGCM case, low-level wind deviations in tropical Pacific (Fig. 12c) are in the same direction as the background climatology (Fig. 12a), which should increase surface evaporation through stronger near surface winds and then results in colder SST through WES feedback. Nevertheless, warmer SST appears in most parts of tropical and subtropical

oceans, laying waste to the assumption trying to explain SST deviation in terms of WES mechanism. Since the top 3 variables that are most relevant to SST deviations (Tab. 1) suggest changes in the surface heat budget, involving the momentum and temperature exchanges between ocean and atmosphere, it implies that the joint effects of atmosphere and ocean models lead to the model deviation patterns (Fig. 15). Simulation of ocean dynamics is different after replacing the OGCM. With changes in ocean advection, the SST and surface currents are altered which modulates the surface evaporation, convection and heat conduction. Subsequently the latent heat and sensible heat fluxes vary over the sea surface. The thermal and moisture perturbations from the ocean are passed to the atmosphere during coupling processes (Fig. 15c, d). Variations of low-level atmospheric circulation and humidity take effects on cloud formation and cloud liquid water path that changes precipitation and cloud radiative forcing. The net shortwave and longwave radiations are influenced and make a difference to the atmospheric circulation (Fig. 12c) and the heat budget over the sea surface (Fig. 15 a, b). Then, the perturbation signal is transferred back to the ocean that changes the SST and surface currents. This air-sea feedback finally reaches a quasi-equilibrium with marked SST warming over vast maritime spaces over the globe. The associated physical processes represented by each radiation term in Figure 15 are in accordance with their signs of pattern correlation (Tab. 1). It seems to suggest that surface evaporation plays a predominant role in the SST differences, because the latent heat flux holds the biggest correlation coefficient and the most obvious deviations among the four radiation terms. However, it is more likely a manifestation of the large-scale ocean dynamical effect on the inter-model differences as suggest by Ying and Huang (2016).

Atmospheric circulation accommodates itself to changes of radiation budget. An anomalous Walker circulation cell appears over the SEC area (Fig. 13b), with updraft over Philippine sea where positive SST deviations indicate a net radiation surplus. The upward flow rises to the upper-level troposphere and condensation becomes evident due to temperature drops. It then diverts toward east and orients downward over eastern Pacific, which enhances surface westerly anomalies in the lower atmosphere. This circulation pattern is consistent with anomalous easterlies over tropical Pacific (Fig. 12c). Warm temperature deviations (Fig.13b) can be viewed as the manifestation of surface radiation surplus and latent heat release of condensation. So far, the analysis has made it clear that changing the OGCM affects the SST simulation through large-scale air-sea feedback that mainly involves surface evaporation, heat conduction, atmospheric convection and cloud masking of incoming and outgoing radiative fluxes. Under the impact of net radiation surplus over western Pacific, the temperature rises with upward motion which forms a warm advection heading east and descending over eastern Pacific. This anomalous Walker cell drives low-level winds towards the west, leading to westerly anomalies over vast areas of tropical Pacific.

**7. Summary and Discussion**

In this study, two new CGCMs have been developed based on the coupling structure of MPI-ESM, namely ECHAM5-NEMO3.6 and ECHAM6-NEMO3.6. The new CGCMs show some improvements in the simulation of SST, precipitation and ocean currents compared with MPI-ESM. The ECHAM5-NEMO3.6 presents the best SST simulation in summer with the minimum cold tongue bias in tropical Pacific and no remarkable bias in North Atlantic, while ECHAM6-NEMO3.6 reproduces

the best winter climatology with most of the biases less than 1 ºC. For precipitation simulation, ECHAM6-NEMO3.6 presents the highest pattern correlation with reanalysis counterpart and substantially ameliorates double ITCZ problem over SPCZ in boreal summer. Biases in surface currents and meridional overturning circulation are also considerably reduced in both ECHAM5-NEMO3.6 and ECHAM6-NEMO3.6. Wind stress bias patterns are alike in most areas among the three CGCMs, but MPI-ESM shows only poleward anomalies over tropical Pacific without strong easterly biases that are common in most existing coupled models. All three CGCMs can generally reproduce the ENSO and SAM variabilities. Model biases are more evident at the interannual and interdecadal scales, suggesting a prominent effect of circulation and oscillation anomalies in both atmosphere and ocean models on the air-sea feedback (Yu and Kim, 2011; Farneti, et al. 2014). As suggested by Cai et al. (2011), the evolution of these climate drivers is affected by their interactions, which requires further investigation. The ECHAM5-NEMO3.6 has much larger biases in total radiation than MPI-ESM, whereas the latter presents the maximum deviation in SST simulation. Besides, the ECHAM5-NEMO3.6 shows warm SST bias in North Pacific, with the opposite sign to most CGCM biases at present time. These facts constitute evidence that suggests model errors of each CGCM are caused by different physical mechanisms.

Meridional overturning circulation in North Pacific and vertical structure of atmospheric circulation are analysed for a comprehensive understanding of bias genesis in each CGCM. Overestimation of tropical cell in the NPMOC transfers more heat to mid-latitudes and results in warm SST bias in North Pacific of the ECHAM5-NEMO3.6. Excessively strong deep tropical cell of the MPI-ESM intensifies the tropical upwelling that leads to the largest cold SST bias among all three CGCMs. The analysis on atmospheric vertical circulation over the SEC area has confirmed that momentum field plays a major role in the SST biases of ECHAM5-NEMO3.6 and ECHAM6-NEMO3.6, while oceanic processes and radiation budget are responsible for the same cold SST biases in tropical Pacific. Since the inter-model differences are caused by changing component models, 12 surface variables that are sensitive to coupling processes are chosen to calculate pattern correlation with the SST differences between CGCMs with the same atmospheric or oceanic component model. The top 3 variables ranked in the case of changing the AGCM are net longwave radiation, surface evaporation (latent heat flux) and net shortwave radiation. Differences in SST, radiation budget and atmospheric circulation are analysed, and it is confirmed that the AGCM replacement first militates in the alteration of cumulus convection including temperature, specific humidity and atmospheric circulation, which in turn changes SST through WES feedback and affects the radiation budget through cloud radiation feedback. Although the AMOC enhancement is less significant compared with the NPMOC, the cloud radiative forcing allows more solar irradiation in subtropical and extra-tropical Atlantic. Thus, the AMOC transports more heat along its course which ameliorates the cold SST biases in North Atlantic. Teleconnections between North Pacific and North Atlantic that contribute to the simulation improvements, as suggested by Zhang and Zhao (2015), are left for an ongoing research.

For the OGCM replacement, the top 3 variables are latent heat flux, sensible heat flux and net longwave flux. Through analysis on circulation and radiation terms, it has been clear that the ocean dynamical effect plays a predominant role in the SST differences after changing the OGCM. The ocean advection initiates the perturbations of SST and surface evaporation that modulate the atmospheric humidity and low-level circulation. Consequently, the cloud masking effect on radiative fluxes

are altered which influences the atmospheric circulation and surface heat budget. The resulting equilibrium of the air-sea feedback manifest itself as the inter-model differences in the related meteorological fields. With the same surface flux parameterizations in the AGCM (Stevens et al., 2013), latent heat flux holds the largest correlation coefficient and exhibits the most prominent variations at the global scale. Since the latent heat differences are resulted from the OGCM replacement, it changes the SST and surface evaporation that also contribute to differences in the near-surface atmospheric humidity. Cao et al. (2015) point out that diversity of simulated SST and near-surface atmospheric specific humidity lead to the most diverse variability of latent heat flux over Pacific in CMIP5 models, which coincides with our research finding.

It is noteworthy that the SST deviations by changing the OGCM and AGCM are nearly the opposite (Fig. 12d, e), suggesting that inverse variations of model SST bias can be realized through mechanisms unveiled in this paper. Although the top 3 coupling variables that are most relevant to SST deviations after changing the AGCM or OGCM are radiation terms, the physical mechanisms behind the opposite SST variations are different. AGCM replacement affects cumulus convection that eventually changes momentum field and radiation budget over the sea surface, while OGCM replacement alters sea surface evaporation that results in latent heat variations and consequently leads to readjustment of radiation budget. It implies that one can pursue simulation improvement of the cold tongue from two aspects: 1. Improve the cumulus convection scheme in the AGCM, 2. Ameliorate errors of ocean advection in the OGCM. With better cumulus convection scheme, easterly anomalies over the tropical sea surface can be reduced significantly, which will cut down the latent heat absorption of more evaporation induced by stronger surface winds, and consequently break down the WES feedback that amplifies the cold SST bias. For the OGCM part, better representation of ocean dynamics can reduce the cold SST bias by modulating the atmospheric momentum, humidity and cloud radiative forcing through air-sea feedback.

For strong easterly bias over eastern Pacific in ECHAM6-NEMO3.6, also common in the most CGCMs, the MPI-ESM instead shows poleward bias, which suggests that OGCM replacement can also diminish this bias through coupling processes involving evaporative cooling and cloud radiative feedback. The associated thermal forcing drives the atmospheric circulation and changes the ocean surface wind biases. It is easy to see that an anomalous Walker circulation rotating clockwise appears over the tropical Pacific, when the surface heating is weakened in the tropical eastern Pacific and is enhanced in the tropical western Pacific (Fig. 13b and Fig. 15). Although the analysis in section 5 and 6 only focuses on the physical mechanisms behind simulation differences in summer, the atmospheric and oceanic circulations have a similar impact on radiation budget and surface heat transport in other seasons. It implies that the effects by replacing each component model may be the same, and the conclusions drawn above are also valid for SST biases in winter. The mechanisms illustrated in this study provides a new vision of model bias origin, which is heuristic for model improvement researches and practices.

**Code and data availability**

The model source code is available from the authors upon request. Restrictions on source code license will be imposed unless for academic and non-commercial use. The experimental data can be found in https://doi.org/10.5281/zenodo.1306338 (Gui et al., 2018).

5     The reanalysis data used in this study can be downloaded from the following websites:

1. The Hadley Centre SST data is downloaded from https://www.metoffice.gov.uk/hadobs/hadisst/.
2. The GPCP precipitation data is downloaded from https://precip.gsfc.nasa.gov/.
3. The ERA-Interim and ERA-20c monthly reanalysis data is available at http://apps.ecmwf.int/datasets/.
4. The SCOW wind stress data is available at http://cioss.coas.oregonstate.edu/scow.
10  5. Surface radiation reanalysis data of CERES EBAF-Surface Ed4.0 is downloaded from https://ceres.larc.nasa.gov.
6. The SODA3.3 ocean current reanalysis data is downloaded from: http://www.atmos.umd.edu.

The MPI-ESM piControl experiment data is available at http://esgf-node.llnl.gov.

**Acknowledgements**

This work was supported by the National Key Research and Development Program of China (2016YFA0601600), and the National Natural Science Foundation of China (U1502233 and 41565002). It is also co-supported by Yunnan University's Research Innovation Fund for Graduate Students (YDY17019).

**Table 1: Pattern Correlations of surface variables and SST deviation**

| Model Replacement / Variables | AGCM | OGCM |
|---|---|---|
| Surface zonal currents | 0.006 | -0.002 |
| Surface meridional currents | 0.0158 | -0.018* |
| Sensible heat flux | -0.068* | -0.269* |
| Latent heat flux | -0.262* | **-0.393*** |
| Mean sea level pressure | -0.136* | -0.009 |
| 10m zonal wind | 0.018* | -0.010 |
| 10m meridional wind | -0.025* | 0.045* |
| Surface albedo | -0.101* | -0.022* |
| Net shortwave flux | 0.192* | 0.138* |
| Net longwave flux | **-0.269*** | -0.283* |
| Precipitation | 0.100* | 0.154* |

Asterisk (*) denotes the correlation coefficients passing the Student-t test above 99.9% confidence level. A larger number of grid points are involved that makes the threshold value relatively small. Numbers in boldface denotes the maximum absolute correlation value for the AGCM or OGCM replacement.

**Table 2: Pattern Correlations of surface variables and SST deviation over the Pacific**

| Model Replacement / Variables | AGCM | OGCM |
|---|---|---|
| Surface zonal currents | 0.069* | -0.033* |
| Surface meridional currents | 0.055* | -0.136* |
| Sensible heat flux | -0.286* | **-0.318**\* |
| Latent heat flux | -0.302* | -0.315* |
| Mean sea level pressure | -0.246* | 0.014 |
| 10m zonal wind | 0.070* | 0.039* |
| 10m meridional wind | 0.022 | -0.005 |
| Surface albedo | 0.274* | 0.034* |
| Net shortwave flux | **0.375**\* | 0.102* |
| Net longwave flux | -0.303* | -0.229* |
| Precipitation | 0.103* | 0.124* |

Asterisk (*) denotes the correlation coefficients passing the Student-t test above 99.9% confidence level. A larger number of grid points are involved that makes the threshold value relatively small. Numbers in boldface denotes the maximum absolute correlation value for the AGCM or OGCM replacement.

[Figure]

**Figure 1: Schematic structure of ECHAM5-NEMO3.6 (a), and ECHAM6-NEMO3.6 (b).**

[Figure]

[Figure]

**Figure 2: Taylor diagram that exhibits a statistical comparison between the simulations and reanalysis data of nine selected variables in summer (a) and winter (b). Each number represents one variable: (1) precipitation, (2) mean sea level pressure, (3) zonal winds at 10m height, (4) meridional winds at 10m height, (5) 2m temperature, (6) total radiation flux (net shortwave plus net longwave), (7) SST, (8) sea surface zonal currents, (9) sea surface meridional currents. Upward-pointing triangles, squares and diamonds, respectively, represent the ECHAM6-NEMO3.6, ECHAM5-NEMO3.6, and the MPI-ESM results.**

[Figure]

**Figure 3: Biases of the SST simulation in summer (left column) and winter (right column) corresponding to each CGCM: (a, b) ECHAM6-NEMO3.6, (c, d) ECHAM5-NEMO3.6, (e, f) MPI-ESM.**

[Figure]

**Figure 4: The same as Fig. 3 but for simulated precipitation climatology.**

[Figure]

**Figure 5: The same as Fig. 3 but for simulated surface wind stress (unit: N/m².).**

[Figure]

**Figure 6: The same as Fig. 3 but for simulated surface currents (unit: m/s).**

[Figure]

**Figure 7: The same as Fig. 3 but for simulated total radiation.**

[Figure]

**Figure 8: Power spectra of (a) Niño3.4 index, (b) SOI, (c) SAM index. Solid line denotes the calculation results of reanalysis data, green dotted line denotes the ECHAM5-NEMO3.6 simulation, red dotted line denotes the ECHAM6-NEMO3.6 simulation, and blue dotted line denotes the MPI-ESM simulation.**

[Figure]

**Figure 9: Model biases of summer climatology of meridional overturning circulation simulation in North Pacific, (a) ECHAM6-NEMO3.6, (b) ECHAM5-NEMO3.6, (c) MPI-ESM, (d) MPI-ESM piControl, Unit: Sv.**

[Figure]

**Figure 10: Model biases of the AMOC in summer, (a) ECHAM6-NEMO3.6, (b) ECHAM5-NEMO3.6, (c) MPI-ESM, (d) MPI-ESM piControl, Unit: Sv.**

[Figure]

**Figure 11: Model biases of the vertical structure of atmospheric circulation (vector) and the temperature (contour) over SEC area in summer for (a) ECHAM6-NEMO3.6, (b) ECHAM5-NEMO3.6, (c) MPI-ESM. The SEC region is marked as the blue box in (d).**
5    **Vector units: m/s, contour unit: °C.**

[Figure]

**Figure 12: Summer climatology of 10m wind for (a) reanalysis data, (b) model differences between ECHAM6-NEMO3.6 and ECHAM5-NEMO3.6, (c) model differences between ECHAM6-NEMO3.6 and MPI-ESM. SST simulation differences for (d) between ECHAM6-NEMO3.6 and ECHAM5-NEMO3.6, and (e) between ECHAM6-NEMO3.6 and MPI-ESM. Vector units: m/s, contour unit: °C.**

[Figure]

**Figure 13: Simulation differences in the vertical structure of atmospheric circulation and the temperature over the SEC area: (a) between ECHAM6-NEMO3.6 and ECHAM5-NEMO3.6, (b) between ECHAM6-NEMO3.6 and MPI-ESM. Vector units: m/s, contour unit: °C.**

[Figure]

**Figure 14: Simulation differences in radiation budget between ECHAM6-NEMO3.6 and ECHAM5-NEMO3.6.**

[Figure]

**Figure 15: Simulation differences in radiation budget between ECHAM6-NEMO3.6 and MPI-ESM.**

---

## Author Comment (AC11) · 3 Oct 2018

Dear professor,

The manuscript has undergone other revision. Please refer to the supplement in AC10 for the latest version manuscript.

Sincerely,

Gui Shu and co-authors

---

## Author Comment (AC12) · 3 Oct 2018

Dear professor,

The manuscript has undergone other revision. Please refer to the supplement in AC10 for the latest version manuscript.

Sincerely,

Gui Shu and co-authors